# On Robust Optimal Transport: Computational Complexity and Barycenter Computation

**Khang Le**[*]

University of Texas, Austin

khanglnt@utexas.edu

**Huy Nguyen**[*]

VinAI Research

v.huynm12@vinai.io

**Quang Minh Nguyen**

Massachusetts Institute of Technology

nmquang@mit.edu

**Tung Pham**

VinAI Research

v.tungph4@vinai.io

**Hung Bui**

VinAI Research

v.hungbh1@vinai.io

**Nhat Ho**

University of Texas, Austin

minhnhat@utexas.edu

## Abstract

We consider robust variants of the standard optimal transport, named robust optimal transport, where marginal constraints are relaxed via Kullback-Leibler divergence. We show that Sinkhorn-based algorithms can approximate the optimal cost of robust optimal transport in $\widetilde{\mathcal{O}}(\frac{n^2}{\varepsilon})$ time, in which $n$ is the number of supports of the probability distributions and $\varepsilon$ is the desired error. Furthermore, we investigate a fixed-support robust barycenter problem between $m$ discrete probability distributions with at most $n$ number of supports and develop an approximating algorithm based on iterative Bregman projections (IBP). For the specific case $m = 2$, we show that this algorithm can approximate the optimal barycenter value in $\widetilde{\mathcal{O}}(\frac{mn^2}{\varepsilon})$ time, thus being better than the previous complexity $\widetilde{\mathcal{O}}(\frac{mn^2}{\varepsilon^2})$ of the IBP algorithm for approximating the Wasserstein barycenter.

## 1 Introduction

The recent advance in computation with optimal transport (OT) problem [12, 3, 13, 7, 22, 26, 20] has led to a surge of interest in using that tool in various domains of machine learning and statistics. The range of its applications is broad, including deep generative models [4, 16, 36], scalable Bayes [33, 34], mixture and hierarchical models [24], and other applications [32, 29, 10, 17, 37, 35, 8].

The goal of optimal transport is to find a minimal cost of moving masses between (supports of) probability distributions. It is known that the estimation of transport cost is not robust when there are outliers. To deal with this issue, [38] proposed a trimmed version of optimal transport. In particular, they search for truncated probability distributions such that the transport cost between them is minimized. However, their trimmed optimal transport is non-trivial to compute, which hinders its usage in practical applications. Another line of works proposed using unbalanced optimal transport (UOT) to solve the sensitivity of optimal transport to outliers [5, 31]. More specifically, their idea is to assign as small as possible masses to outliers by relaxing the marginal constraints of OT through a penalty function such as the Kullback-Leibler (KL) divergence. This direction of robust optimal transport has been shown to have good performance in generative models and domain adaptation [5]. Although this approach achieved considerable success, the full picture of its computational complexity has remained missing.

---

[*] Khang Le and Huy Nguyen contributed equally to this work.

35th Conference on Neural Information Processing Systems (NeurIPS 2021).

**Our Contribution:** In the paper, we provide a comprehensive study of the computational complexity of robust optimal transport and its corresponding barycenter problem when the probability distributions are discrete and have at most $n$ components. Our contribution is twofold and can be summarized as follows:

(1) **On robust optimal transport,** we consider two versions corresponding to two ways of relaxing marginal constraints in the standard optimal transport problem via the KL divergence. We show that two scaling algorithms computing these robust formulations have the complexities $\widetilde{\mathcal{O}}(n^2/\varepsilon)$, where $\varepsilon$ denotes the desired error for the computed cost. These complexities are lower than the complexity of the Sinkhorn algorithm for solving the optimal transport problem, which is $\widetilde{\mathcal{O}}(n^2/\varepsilon^2)$ [13], and match the complexity of the Sinkhorn algorithm that solves the UOT problem [27]. Furthermore, we show how the above complexity can be improved by utilizing the low-rank approximation method to speed up the matrix-vector computations in the loop similar to [2], and obtain the improved computing time of $\widetilde{O}(nr^2 + \frac{nr}{\varepsilon})$, where $r$ is the approximated rank.

(2) **On robust barycenter problem,** where the goal is to determine a probability measure that minimizes its robust optimal cost to a given set of $m \geq 2$ probability measures, we propose ROBUSTIBP algorithm for solving the robust barycenter problem, which is inspired by the iterative Bregman projection (IBP) algorithm for solving the traditional barycenter problem [6]. We show that when $m = 2$, the complexity of ROBUSTIBP algorithm is at the order of $\widetilde{\mathcal{O}}(mn^2/\varepsilon)$, better than that of the IBP algorithm for solving the traditional barycenter problem [19], which is $\widetilde{\mathcal{O}}(mn^2/\varepsilon^2)$. To the best of our knowledge, the ROBUSTIBP is also the first practical algorithm obtaining the near-optimal complexity $\widetilde{\mathcal{O}}(mn^2/\varepsilon)$ for solving the barycenter problem even under only the setting $m = 2$.

**Organization:** The paper is organized as follows. In Section 2, we provide the background on the optimal transport problem and some of its variants that have robust effects. In Section 3, we discuss in-depth the variant where only one marginal constraint is relaxed, study the computational complexity of a Sinkhorn-based algorithm that solves it, and then briefly introduce the fully-relaxed formulation. We also establish the complexities of these algorithms after applying Nyström method. Subsequently, we present our study of the robust barycenter problem in Section 4. In Section 5, we carry out empirical studies to illustrate the theories before concluding with a few discussions in Section 6. The proofs of our theoretical results are in the supplementary material.

**Notation:** We let $[n]$ stand for the set $\{1, 2, \ldots, n\}$ while $\mathbb{R}^n_+$ indicates the set of all vectors with non-negative entries. For a vector $x \in \mathbb{R}^n$ and $p \in [1, \infty)$, we denote $\|x\|_p$ as its $\ell_p$-norm and $\operatorname{diag}(x)$ as the diagonal matrix with $x$ on the diagonal. The natural logarithm of a vector $\mathbf{a} = (a_1, ..., a_n) \in \mathbb{R}^n_+$ is denoted by $\log \mathbf{a} = (\log a_1, ..., \log a_n)$, $\mathbf{1}_n$ stands for a vector of length $n$ that all of its entries equal to 1, and $\partial_x f$ refers to the partial differentiation of function $f$ with respect to $x$. For any given space $\mathcal{X} \subset \mathbb{R}^d$, we denote by $\mathcal{P}(\mathcal{X})$ the space of all probability measures on $\mathcal{X}$. Given an integer $n > 0$ and a real number $\varepsilon > 0$, the notation $a = \mathcal{O}(b(n, \varepsilon))$ means that $a \leq C \cdot b(n, \varepsilon)$ where $C$ is independent of $n$ and $\varepsilon$. Meanwhile, the notation $a = \widetilde{\mathcal{O}}(b(n, \varepsilon))$ indicates the previous inequality may depend on a logarithmic function of $n$ and $\varepsilon$. For any two probability measures $\mathbf{x} = (x_1, \ldots, x_n)$ and $\mathbf{y} = (y_1, \ldots, y_n)$ with the same supports, the generalized Kullback-Leibler divergence is defined as $\mathbf{KL}(\mathbf{x}\|\mathbf{y}) = \sum_{i=1}^n \left[ x_i \log \left( \frac{x_i}{y_i} \right) - x_i + y_i \right]$. Finally, the entropy of a matrix $X$ is given by $H(X) = \sum_{i,j=1}^n -X_{ij}(\log X_{ij} - 1)$.

## 2 Background on Optimal Transport

In this section, we review optimal transport and its unbalanced formulation, then from that deriving formulations for robust optimal transport. For any $P$ and $Q$ in $\mathcal{P}(\mathcal{X})$ for a space $\mathcal{X}$, the OT distance between $P$ and $Q$ takes the following form

$$\mathrm{OT}(P, Q) := \min_{\pi \in \Pi(P,Q)} \int c(x, y) d\pi(x, y), \tag{1}$$

where $\Pi(P, Q)$ is the set of joint probability distributions in $\mathcal{X} \times \mathcal{X}$ such that their marginal distributions are $P$ and $Q$, and $c : \mathcal{X} \times \mathcal{X} \to [0, \infty)$ is a cost function.

**Unbalanced Optimal Transport:** When $P$ or $Q$ is not a probability distribution, the OT formulation between $P$ and $Q$ in equation (1) is no longer valid. One solution to this issue is using the unbalanced optimal transport (UOT) [9], which is given by:

$$\text{UOT}(P,Q) := \min_{\pi \in \mathcal{M}_+(\mathcal{X} \times \mathcal{X})} \int c(x,y)d\pi(x,y) + \tau_1 \textbf{KL}(\pi_1 \| P) + \tau_2 \textbf{KL}(\pi_2 \| Q), \quad (2)$$

where $\mathcal{M}_+(\mathcal{X} \times \mathcal{X})$ denotes the set of joint non-negative measures on the space $\mathcal{X} \times \mathcal{X}$; $\pi_1, \pi_2$ are the marginal distributions of $\pi$ and respectively correspond to $P$ and $Q$; $\tau_1, \tau_2$ are regularized positive parameters. Note that, we can replace the KL divergence in equation (2) by any Csiszár-divergence [11]. However, we only consider the case of KL divergence in this work.

**Robust Optimal Transport:** Optimal transport is well-known for not being robust in the present of outliers. A way to deal with this issue is using the approach of unbalanced optimal transport (UOT), which has demonstrated favorable practical performance in generative models and domain adaptation [5]. More specifically, when $P$ and $Q$ are probability distributions in $\mathcal{X}$, the ***R**obust Unconstrained **O**ptimal **T**ransport (ROT)* admits the following form

$$\text{ROT}(P,Q) := \inf_{P_1, Q_1 \in \mathcal{P}(\mathcal{X})} \min_{\pi \in \Pi(P_1, Q_1)} \int c(x,y)d\pi(x,y) + \tau_1 \textbf{KL}(P_1 \| P) + \tau_2 \textbf{KL}(Q_1 \| Q), \quad (3)$$

where $\tau_1, \tau_2 > 0$ are some given regularized parameters. The reason to name it robust unconstrained optimal transport is that instead of looking for an optimal transport plan moving masses from $P$ to $Q$, we seek another plan that optimally transports masses between their approximations, which are probability measures $P_1$ and $Q_1$, under the KL divergence. This formulation is closely related to the ones studied in [5] and [23]: the former used $\chi^2$-divergence for the relaxation and the latter used total variation distance (note that those three divergences all together belong to the family of $f$-divergence).

By relaxing only one marginal constraint regarding (presumably) on $P$, we have another version of ROT, named ***R**obust **S**emi-constrained **O**ptimal **T**ransport (RSOT)*, which is given by

$$\text{RSOT}(P,Q) := \inf_{P_1 \in \mathcal{P}(\mathcal{X})} \min_{\pi \in \Pi(P_1, Q)} \int c(x,y)d\pi(x,y) + \tau \textbf{KL}(P_1 \| P), \quad (4)$$

where $\tau > 0$ is a regularized parameter. We could also define $\text{RSOT}(Q,P)$ similarly with a remark that although $\text{RSOT}(P,Q)$ can be different from $\text{RSOT}(Q,P)$, the techniques for obtaining the computational complexity of both are similar.

**UOT vs ROT/RSOT:** Though the formulations ROT/RSOT and UOT seem to be similar, they serve different purposes. The goal of UOT is to deal with unbalanced measures, thus there is no condition on the "transport plan". Hence, the meaning of the optimal plan $\pi$ of UOT problem is dependent on the interpreter. For example, in applications such as [30], the UOT is used to figure out the developmental trajectory of cells. Meanwhile, ROT/RSOT aim to seek an accurate transport plan between two possibly corrupted probability distributions. The toy example in Figure 1 illustrates this difference. In particular, the marginals of the "transport plan" obtained by the latter (see plots $(b), (d)$) are very different from the two original probability measures $\mathbf{a}, \mathbf{b}$. On the other hand, the solution of the former leads to good approximations of $\mathbf{a}$ and $\mathbf{b}$ (see plots $(a), (c)$) while removing some bumps in both tails which are presumably outliers.

## 3 Discrete Robust Optimal Transport and its Computational Complexity

When $P$ and $Q$ are discrete measures, the KL penalties in equations (3) and (4) suggest that the probability distributions $P_1$ and $Q_1$ need to share the same set of supports as that of $P$ and $Q$, respectively. Therefore, throughout this section, we implicitly require this condition in our formulations of RSOT and ROT and we denote the masses of $P$ and $Q$ by $\mathbf{a}$ and $\mathbf{b}$, respectively.

### 3.1 Robust Semi-constrained Optimal Transport

Assume that the marginal constraint associating with $Q$ is kept and that of $P$ is relaxed and $P_1$ and $P$ share the same set of supports, the formulation of RSOT in equation (4) can be rewritten as follows

$$\min_{X \in \mathbb{R}_+^{n \times n}, X^\top \mathbf{1}_n = \mathbf{b}} f_{\text{rsot}}(X) := \langle C, X \rangle + \tau \textbf{KL}(X\mathbf{1}_n \| \mathbf{a}), \quad (5)$$

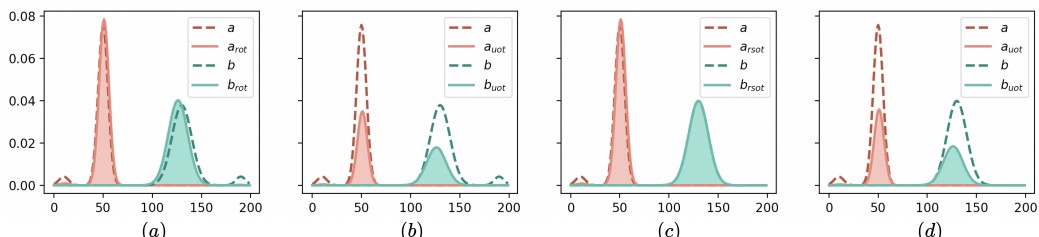

Figure 1: Comparison on two marginals induced by ROT/RSOT solutions and UOT solutions. Here $\mathbf{a}, \mathbf{b}$ are two (possibly corrupted) 1-D Gaussian distributions on which we compute the optimal transport, and $a_{[problem]}, b_{[problem]}$ represent two marginals (with respect to $a$ and $b$ respectively) of the optimal solution for the corresponding $[problem]$. In plots $(a), (b)$, we compare ROT and UOT where both $\mathbf{a}$ and $\mathbf{b}$ contain (10%) outliers from other Gaussians, while in plots $(c), (d)$ we investigate RSOT and UOT where only $\mathbf{a}$ is corrupted.

---

**Algorithm 1:** ROBUST-SEMISINKHORN

> **Input:** $C, \mathbf{a}, \mathbf{b}, \eta, \tau, n_{\text{iter}}$
> **Initialization:** $u^0 = v^0 = 0, k = 0$
> **while** $k < n_{\text{iter}}$ **do**
> $\quad a^k \leftarrow B(u^k, v^k)\mathbf{1}_n, \quad b^k \leftarrow \left(B(u^k, v^k)\right)^\top \mathbf{1}_n$
> $\quad$ **if** $k$ is even **then**
> $\quad\quad u^{k+1} \leftarrow \frac{\eta\tau}{\eta+\tau}\left[\frac{u^k}{\eta} + \log(\mathbf{a}) - \log(a^k)\right]$
> $\quad\quad v^{k+1} \leftarrow v^k$
> $\quad$ **else**
> $\quad\quad u^{k+1} \leftarrow u^k$
> $\quad\quad v^{k+1} \leftarrow \eta\left[\frac{v^k}{\eta} + \log(\mathbf{b}) - \log(b^k)\right]$
> $\quad$ **end if**
> $\quad k \leftarrow k + 1$
> **end while**
> **return** $B(u^k, v^k)$

---

where $\mathbf{a}, \mathbf{b}$ are the masses of $P$ and $Q$ respectively, and $C$ is the cost matrix whose entries are distances between the supports of these distributions. Solving directly problem (5) by traditional linear programming solvers can be expensive and not scalable in terms of $n$. Therefore, we utilize the entropic regularization approach proposed by [12] to the objective function of RSOT, leading to

$$\min_{X \in \mathbb{R}_+^{n \times n}, X^\top \mathbf{1}_n = \mathbf{b}} g_{\text{rsot}}(X) := f_{\text{rsot}}(X) - \eta H(X). \tag{6}$$

Here, $\eta > 0$ is a given regularization parameter, and we refer the problem (6) to as *entropic RSOT*. The dual problem of entropic RSOT is

$$\min_{u,v \in \mathbb{R}^n} h_{\text{rsot}}(u, v) := \eta\|B(u,v)\|_1 + \tau\langle e^{-u/\tau}, \mathbf{a}\rangle - \langle v, \mathbf{b}\rangle, \tag{7}$$

where $B(u, v)$ is defined as a matrix of size $n \times n$ with entries $[B(u,v)]_{ij} := e^{(u_i + v_j - C_{ij})/\eta}$. Since equation (7) is an unconstrained convex optimization problem, we can perform alternating minimization for $u$ and $v$ by setting $\partial h(u,v)/\partial u = 0$ and $\partial h(u,v)/\partial v = 0$, resulting in closed-form updates of a Sinkhorn-like procedure (see [12]) in Algorithm 1. This procedure is known to converge to the optimal solution $(u^*, v^*) := \arg\min h_{\text{rsot}}(u, v)$. As strong duality holds for the convex optimization problem (6), the optimal transport plan of the entropic RSOT is exactly $B(u^*, v^*)$.

Since no assumptions are made on the cost matrix, except its entries are non-negative, closed-form solutions of OT and UOT generally do not exist. Therefore, we introduce the definition of an *$\varepsilon$-approximation* solution of an optimization problem, which will be used for all the subsequent complexity analyses.

**Definition 1** ($\varepsilon$-**approximation**). *For any $\varepsilon > 0$, a transportation plan $X$ is called an $\varepsilon$-approximation of the minimizer $\widehat{X}$ of some objective function $f$ if $f(X) \le f(\widehat{X}) + \varepsilon$.*

Based on this concept, we then state our main theorem on the runtime complexity of Algorithm 1 in solving the RSOT problem (5).

**Theorem 1.** *For $U_{\mathrm{rsot}} := \max\{3\log(n), \varepsilon/\tau\}$ and $\eta = \varepsilon/U_{\mathrm{rsot}}$, Algorithm 1 returns an $\varepsilon$-approximation of the optimal solution $\widehat{X}_{\mathrm{rsot}}$ of the problem (5) in time*

$$
\mathcal{O}\left(\frac{\tau n^2}{\varepsilon}\log(n)\left[\log\left(\frac{\tau\|C\|_\infty}{\varepsilon}\right) + \log(\log(n))\right]\right).
$$

*Proof Sketch.* The full proof of Theorem 1 is in Appendix B. Note that, this result is not achieved by directly applying Theorem 2 in [27] with $\tau_2 \to \infty$ as the nature of the dual function changes in that limit, invalidating many previous results. Let $X_{\mathrm{rsot}}^k$ be the output of Algorithm 1 at the $k$-th step while $\widehat{X}_{\mathrm{rsot}}$ and $X_{\mathrm{rsot}}^*$ denotes the minimizers of equations (5) and (6), respectively. The goal is to find $k$ that guarantees $f_{\mathrm{rsot}}(X_{\mathrm{rsot}}^k) - f_{\mathrm{rsot}}(\widehat{X}_{\mathrm{rsot}}) \le \varepsilon = \eta U_{\mathrm{rsot}}$. We start by decomposing

$$
\underbrace{f_{\mathrm{rsot}}(X_{\mathrm{rsot}}^k)}_{g_{\mathrm{rsot}}(X_{\mathrm{rsot}}^k)+\eta H(X_{\mathrm{rsot}}^k)} - \underbrace{f_{\mathrm{rsot}}(\widehat{X}_{\mathrm{rsot}})}_{g_{\mathrm{rsot}}(\widehat{X}_{\mathrm{rsot}})+\eta H(\widehat{X}_{\mathrm{rsot}})} \le \left[g_{\mathrm{rsot}}(X_{\mathrm{rsot}}^k) - g_{\mathrm{rsot}}(X_{\mathrm{rsot}}^*)\right] + \eta\left[H(X_{\mathrm{rsot}}^k) - H(\widehat{X}_{\mathrm{rsot}})\right],
$$

and try to bound each term by a linear function of $\eta$. Dealing with the entropy term is simple as the $\eta$ factor is already presented, and the entropy difference can be bounded by a constant due to the fact that $1 \le H(X) \le 2\log(n) + 1$ for all $X \in \mathbb{R}_+^{n\times n}, \|X\|_1 = 1$. The non-trivial part is bounding the difference between $g_{\mathrm{rsot}}$ values, which hinges upon two results. The first one is the value of $g_{\mathrm{rsot}}$ at optimality:

$$
g_{\mathrm{rsot}}(X_{\mathrm{rsot}}^*) = -\eta - \tau(1-\alpha) + \langle v_{\mathrm{rsot}}^*, b_{\mathrm{rsot}}^* \rangle. \tag{8}
$$

The second result is the geometric convergence rate of the updates on $u$ and $v$ (Lemma 6 in Appendix B):

$$
\max\left\{\|u^{k+1} - u^*\|_\infty, \|v^{k+1} - v^*\|_\infty\right\} \le (\mathrm{const})\left(\frac{\tau}{\tau+\eta}\right)^{k/2} =: \Delta^k.
$$

The final step is using equation (8) to tailor the $g_{\mathrm{rsot}}$ difference to be bounded by a linear function of $\Delta^k$, which is an exponential function of $k$, then solving for the minimum $k$ at which this exponential function is small enough compared to $\eta$. The main technical difficulty here is to deal with the unknown term $\langle v_{\mathrm{rsot}}^*, b_{\mathrm{rsot}}^* \rangle$ in equation (8), which causes the deviation from the previous techniques. $\qquad\square$

**Remark 1.** *The result of Theorem 1 indicates that the complexity of ROBUST-SEMISINKHORN algorithm for computing RSOT is at the order of $\widetilde{\mathcal{O}}(\frac{n^2}{\varepsilon})$. This complexity is near-optimal and faster than the complexity of the standard Sinkhorn algorithm for computing the optimal transport problem [13, 22], which is at the order of $\widetilde{\mathcal{O}}(\frac{n^2}{\varepsilon^2})$.*

### 3.2 Robust Unconstrained Optimal Transport

In this section, we briefly present another version of robust optimal transport, abbreviated by ROT, when two distributions are contaminated. We first show that the approach of using the duality of the objective function of ROT problem with entropic regularizer does not produce a Sinkhorn algorithm as in the cases of RSOT and UOT. However, a second thought of the problem finds an interesting link between the optimal solutions of ROT and UOT, which results in a nice algorithm for the ROT. We also discuss some technical difficulties when analysing the complexity for the ROT problem. At the end of this section, we show that the result could be extended to the case of low-rank cost matrix, which will significantly reduce the computation.

Recall that the masses of $P$ and $Q$ are $\mathbf{a}$ and $\mathbf{b}$, respectively, the ROT problem (3) becomes

$$
\min_{X\in\mathbb{R}_+^{n\times n}, \|X\|_1=1} f_{\mathrm{rot}}(X) := \langle C, X\rangle + \tau\mathbf{KL}(X\mathbf{1}_n\|\mathbf{a}) + \tau\mathbf{KL}(X^\top\mathbf{1}_n\|\mathbf{b}). \tag{9}
$$

Here we set $\tau_1 = \tau_2 = \tau$ for the sake of simplicity, since there are no more technical difficulties to work with finite $\tau_1 \ne \tau_2$. As noted in Section 2, the formulation (9) bears some resemblance to the unbalanced optimal transport problem studied in [27], except the additional norm condition forcing $X$ to be a transportation plan (i.e., a joint probability distribution), which shows the different nature

of two problems. Following the approach of using the Sinkhorn algorithm of UOT, the duality of formulation (9) has the form

$$\eta \log \|B(u,v)\|_1 + \tau \big\{ \langle e^{u/\tau}, \mathbf{a} \rangle + \langle e^{v/\tau}, \mathbf{b} \rangle \big\}.$$

By taking derivatives of the above function with respect to $u$ and $v$ and set the derivatives to be zero, we obtain

$$\frac{B(u,v)\mathbf{1}_n}{\|B(u,v)\|_1} = e^{-u/\tau} \odot \mathbf{a}, \quad \frac{B(u,v)^\top \mathbf{1}_n}{\|B(u,v)\|_1} = e^{-v/\tau} \odot \mathbf{b},$$

where $\odot$ denotes element-wise multiplication. Unfortunately, the above equations do not have closed-form solutions to produce update as the Sinkhorn algorithms do because of the term $\|B(u,v)\|_1$ in the denominator. However, the objective function of UOT is not homogeneous with respect to $X$, but could be written as a linear function of ROT and another function of $\|X\|_1$ due to some special properties of the KL divergence. This observation leads to the interesting result summarized in the below lemma.

**Lemma 1** (**Connections with UOT**). *The optimal solution of problem* (9)*, denoted $X_{\mathrm{rot}}^*$, is the normalized version of $X_{\mathrm{uot}}^*$ which is the minimizer of UOT in entropic formulation. More specifically, we have $X_{\mathrm{rot}}^* = \frac{X_{\mathrm{uot}}^*}{\|X_{\mathrm{uot}}^*\|_1}$.*

The proof of Lemma 1 is in Appendix D. Based on this result, we can utilize the Sinkhorn algorithm that solves UOT (see [27]) with a normalizing step at the end to produce a solution for the ROT. Although the normalizing step is convenient in finding ROT's solution, it introduces new challenge in the proof compared to that of UOT since the normalizing constant does not have a lower bound. Even so, we are still able to obtain an $\varepsilon$-approximation solution for the ROT in $\widetilde{\mathcal{O}}(n^2/\varepsilon)$ time without any additional constraints on the setting. For more technical details, please refer to Appendix D.

**Further Improving Complexities by Low-Rank Approximation:** As a consequence of our complexity analysis, we can show that by using low-rank approximation method studied in [2] to the kernel matrix $K := \exp(-C/\eta)$, we could further reduce the complexities of both robust semi/unconstrained optimal transport problem to $\widetilde{O}(nr^2 + nr/\varepsilon)$ time, given the same $\varepsilon$-approximation and the approximated-rank $r$. This result is essentially different from the complexity studied in [2], where the $\varepsilon$-approximation is considered regarding the optimal value of the entropic-regularized problem, not the original one in our analysis. For a more detailed discussion, please refer to Appendix E.

## 4 The Robust Barycenter Problem

In this section, we consider the problem of computing the barycenter of a set of possibly corrupted probability measures. The semi-constrained formulation arises as a natural candidate for this goal, when potential outliers only appear in the given probability measures and the desired barycenter is the barycenter of the uncontaminated probability measures. In particular, assume that we have $m \geq 2$ discrete probability measures $P_1, \ldots, P_m$: each has at most $n$ fixed support points and the associated positive weights are given by $\omega_1, \ldots, \omega_m$ ($\sum_{i=1}^m \omega_i = 1$). The barycenter problem then aims to find the probability measure that minimizes $\sum_{i=1}^m \omega_i \mathrm{RSOT}(P_i, P)$, which is a linear combination of RSOT divergence from the barycenter to all given probability measures. We refer it as *Robust Semi-constrained Barycenter Problem (RSBP)*. In this work, we consider the fixed-support settings where all the probability measures $P_i$ share the same set of support points. This setting had been widely used in the previous works to study the computational complexity of Wasserstein barycenter problem [19, 21]. Let $\mathbf{p}_i$ be the mass of probability measure $P_i$ for $i \in [m]$, the discrete RSBP reads

$$\min_{\mathbf{p} \in \mathbb{R}_+^n, \|\mathbf{p}\|_1=1} \quad \sum_{i=1}^m \omega_i \Big[ \min_{X_i \in \mathbb{R}_+^{n \times n}, X_i^\top \mathbf{1}_n = \mathbf{p}} \langle C_i, X_i \rangle + \tau \mathbf{KL}(X_i \mathbf{1}_n \| \mathbf{p}_i) \Big],$$

which is equivalent to

$$\min_{\mathbf{X} \in \mathcal{D}_1(\mathbf{X})} \quad f_{\mathrm{rsbp}}(\mathbf{X}) := \sum_{i=1}^m \omega_i \big[ \langle C_i, X_i \rangle + \tau \mathbf{KL}(X_i \mathbf{1}_n \| \mathbf{p}_i) \big], \tag{10}$$

**Algorithm 2:** ROBUSTIBP

**Input:** $\{C_i\}_{i=1}^m, \{\mathbf{p}_i\}_{i=1}^m, \tau, \eta, n_{\text{iter}}$
**Initialization:** $u_i^0 = v_i^0 = \mathbf{0}_n$ for $i \in [m]$, $k = 0$
**while** $k < n_{\text{iter}}$ **do**
$\quad a_i^k \leftarrow B(u_i^k, v_i^k; C_i)\mathbf{1}_n; \quad b_i^k \leftarrow \left(B(u_i^k, v_i^k; C_i)\right)^\top \mathbf{1}_n \quad \forall i \in [m]$
$\quad$ **if** $k$ is even **then**
$\qquad u_i^{k+1} \leftarrow \frac{\eta\tau}{\eta+\tau}\left[\frac{u_i^k}{\eta} + \log(\mathbf{p}_i) - \log(a_i^k)\right] \quad \forall i \in [m]$
$\qquad v_i^{k+1} \leftarrow v_i^k \quad \forall i \in [m]$
$\quad$ **else**
$\qquad u_i^{k+1} \leftarrow u_i^k \quad \forall i \in [m]$
$\qquad v_i^{k+1} \leftarrow \eta\left[\frac{v_i^k}{\eta} - \log(b_i^k) - \sum_{t=1}^m \omega_t(\frac{v_t^k}{\eta} - \log(b_t^k))\right] \quad \forall i \in [m]$
$\quad$ **end if**
$\quad k \leftarrow k + 1$
**end while**
$X_i^k \leftarrow B(u_i^k, v_i^k; C_i) \quad \forall i \in [m]$
**return** $(X_1^k, \ldots, X_m^k)$ for equation (14)     or     $\left(\frac{X_1^k}{\|X_1^k\|_1}, \ldots, \frac{X_m^k}{\|X_m^k\|_1}\right)$ for equation (11).

where $\mathcal{D}_1(\mathbf{X}) := \big\{(X_1, \ldots, X_m) : X_i \in \mathbb{R}_+^{n \times n}$ and $\|X_i\|_1 = 1 \, \forall i \in [m]; X_i^\top \mathbf{1}_n = X_{i+1}^\top \mathbf{1}_n \, \forall i \in [m-1]\big\}$. Note that the objective function of RSBP is different from that of Wasserstein barycenter [19]: here we relax the marginal constraints $X_i \mathbf{1}_n = \mathbf{p}_i$ by using the KL divergence to deal with the contaminated $P_i$. Finally, the constraints $X_i^\top \mathbf{1}_n = X_{i+1}^\top \mathbf{1}_n = \mathbf{p}$ are to guarantee that the transportation plans $X_i$ have one common marginal which turns out to be a feasible barycenter $\mathbf{p}$. Similar to RSOT, we consider an entropic-regularized formulation of (10), named *entropic RSBP*:

$$\min_{\mathbf{X} \in \mathcal{D}_1(\mathbf{X})} g_{\text{rsbp}}(\mathbf{X}) := \sum_{i=1}^m \omega_i g_{\text{rsot}}(X_i; \mathbf{p}_i, C_i). \tag{11}$$

Since some functions like $g_{\text{rsot}}(X)$, depends on some parameters like $C_i$ and $\mathbf{p}_i$, we sometimes abuse the notation by including these parameters next to variables, e.g., $g_{\text{rsot}}(X_i; C_i, \mathbf{p}_i)$. A general approach to deal with (11) is to consider its dual function, which admits the following form:

$$\min_{\substack{\mathbf{u}=(u_1,\ldots,u_m), \mathbf{v}=(v_1,\ldots,v_m) \\ \sum_{i=1}^m \omega_i v_i = 0}} h_{\text{rsbp}}(\mathbf{u}, \mathbf{v}) := \sum_{i=1}^m \omega_i \left[\eta \log \|B(u_i, v_i; C_i)\|_1 + \tau\langle e^{-u_i/\tau}, \mathbf{p}_i\rangle\right]. \tag{12}$$

We could use the alternating minimization method to find the minimizer of (12). In particular, starting at an initialization $\mathbf{u}^0$ and $\mathbf{v}^0$, we update them alternatively as follows:

$$\mathbf{u}^{k+1} = \arg\min_{\mathbf{u}} h_{\text{rsbp}}(\mathbf{u}, \mathbf{v}^k), \quad \mathbf{v}^{k+1} = \arg\min_{\mathbf{v}: \sum_{i=1}^m \omega_i v_i = 0} h_{\text{rsbp}}(\mathbf{u}^{k+1}, \mathbf{v}). \tag{13}$$

In some problems (e.g., RSOT), closed-form updates can be acquired if the system of equations $\partial h_{\text{rsbp}}(\mathbf{u}, \mathbf{v}^k)/\partial\mathbf{u} = \mathbf{0}$ and $\partial h_{\text{rsbp}}(\mathbf{u}^k, \mathbf{v})/\partial\mathbf{v} = \mathbf{0}$ could be solved exactly by some simple formulas. However, this is not the case with the formulation of $h_{\text{rsbp}}$ in equation (12) because the logarithmic term leads to an intractable system of equations of the partial derivative of $h_{\text{rsbp}}$. Instead, we propose to solve the optimization problem (11) via another objective function, whose dual form can be solved effectively by alternating minimization.

## 4.1 ROBUSTIBP Algorithm

We consider a similar problem to the entropic RSBP in (11), with its feasible set $\mathcal{D}(\mathbf{X}) := \big\{(X_1, \ldots, X_m) : X_i \in \mathbb{R}_+^{n \times n}, \forall i \in [m]; X_i^\top \mathbf{1}_n = X_{i+1}^\top \mathbf{1}_n \forall i \in [m-1]\big\}$ which does not have the

norm constraint. The primal objective function and its dual are as follows:

$$\textbf{Primal:} \quad \min_{\mathbf{X} \in \mathcal{D}(\mathbf{X})} g_{\text{rsbp}}(\mathbf{X}) := \sum_{i=1}^{m} \omega_i g_{\text{rsot}}(X_i; \mathbf{p}_i, C_i), \tag{14}$$

$$\textbf{Dual:} \quad \min_{\mathbf{u}, \mathbf{v}: \sum_{i=1}^{m} \omega_i v_i = \mathbf{0}} \bar{h}_{\text{rsbp}}(\mathbf{u}, \mathbf{v}) := \sum_{i=1}^{m} \omega_i \big[ \eta \| B(u_i, v_i; C_i) \|_1 + \tau \langle e^{-u_i/\tau}, \mathbf{p}_i \rangle \big]. \tag{15}$$

The dual formulation (15) has a closed form updates for $\mathbf{u}$ and $\mathbf{v}$. Based on these, we develop Algorithm 2, namely ROBUSTIBP, since this procedure resembles the iterative Bregman projections studied in [6] and [19]. The updates of $\mathbf{u}$ and $\mathbf{v}$ are known to converge to the optimal solution $(\mathbf{u}^*, \mathbf{v}^*)$ of the problem (15), and strong duality suggests that $\mathbf{X}^* = (B(u_i^*, v_i^*; C_i))_{i=1}^m$ is the optimal solution of the problem (14). Furthermore, there is an intriguing relation between the optimal solution of the problem (14) to that of the problem (11), presented in the following lemma.

**Lemma 2.** *Let* $\bar{\mathbf{X}}^* = (\bar{X}_1^*, \ldots, \bar{X}_m^*)$ *and* $\mathbf{X}^* = (X_1^*, \ldots, X_n^*)$ *be the optimizers of* $g_{\text{rsbp}}$ *with the feasible set* $\mathcal{D}(\mathbf{X})$ *and with the feasible set* $\mathcal{D}_1(\mathbf{X})$, *respectively. Then,* $X_i^* = \dfrac{\bar{X}_i^*}{\|\bar{X}_i^*\|_1}$ *for all* $i \in [m]$.

The proof of Lemma 2 is in Appendix C. This result indicates that we can approximate the solution of equation (11) by the solution of equation (14), using the same Algorithm 2 with an additional normalizing step at the end.

## 4.2 Complexity Analysis

In this section, we provide the analysis of ROBUSTIBP algorithm for obtaining an $\varepsilon$-approximation of the robust semi-constrained barycenter problem (11) when $m = 2$. We also discuss the challenges of extending the current proof technique to $m \geq 3$ at the end of this section. First, we present the complexity of the ROBUSTIBP algorithm in the following theorem.

**Theorem 2.** *For* $m = 2$ *and* $\eta = \varepsilon U_{\text{rsbp}}^{-1}$ *where* $U_{\text{rsbp}} := \max\{2 + 2\log(n), 2\varepsilon, 3\varepsilon \log(n)/\tau\}$, *the* ROBUSTIBP *algorithm returns an* $\varepsilon$-*approximation of the optimal solution* $(\widehat{X}_1, \ldots, \widehat{X}_m)$ *of the RSBP* (10) *in time* $\mathcal{O}\Big( \dfrac{\tau n^2}{\varepsilon} \log(n) \Big[ \log \Big( \tau \sum_{i=1}^{m} \|C_i\|_\infty \Big) + \log \Big( \dfrac{\log(n)}{\varepsilon} \Big) \Big] \Big)$.

**Remark 2.** *The complexity* $\widetilde{\mathcal{O}}(n^2/\varepsilon)$ *of* ROBUSTIBP *algorithm is near-optimal and better than that of IBP algorithm for solving the Wasserstein barycenter problem, which is* $\widetilde{\mathcal{O}}(n^2/\varepsilon^2)$ *when* $m = 2$ *in [19]. It is also better than the complexity of* FASTIBP *algorithm in [21], which is* $\widetilde{\mathcal{O}}(n^{7/3}/\varepsilon^{4/3})$. *To the best of our knowledge, the* ROBUSTIBP *is also the first practical algorithm obtaining the near-optimal complexity* $\widetilde{\mathcal{O}}(n^2/\varepsilon)$ *for solving the barycenter problem under the setting* $m = 2$.

The main ingredient in the proof of Theorem 2 is the convergence rate of vectors $\mathbf{u}$ and $\mathbf{v}$ of the problem (15), which is captured as follows:

$$\max \Big\{ \sum_{i=1}^{m} \|\Delta u_i^{k+1}\|_\infty, \sum_{i=1}^{m} \|\Delta v_i^{k+1}\|_\infty \Big\} \leq (\text{constant}) \Big( \dfrac{\tau}{\tau + \eta} \Big)^{k/2}, \tag{16}$$

where $\Delta u_i^k := u_i^{k+1} - u_i^*$ and $\Delta v_i^k := v_i^{k+1} - v_i^*$. The result can be achieved by alternatively applying two following inequalities.

For the first inequality, with even $k$, from the update of $\mathbf{u}^{k+1}$ in the Algorithm 2, we obtain $\|\Delta u_i^{k+1}\|_\infty \leq \frac{\tau}{\tau+\eta} \|\Delta v_i^k\|_\infty$.

The second inequality is obtained from the update of $\mathbf{v}^k$ in Algorithm 2 as follows:

$$\sum_{i=1}^{m} \|\Delta v_i^k\|_\infty \leq \sum_{i=1}^{m} ((m-2)\omega_i + 1) \|\Delta u_i^{k-1}\|_\infty.$$

Thus, when $m = 2$, we can achieve inequality (16), though this approach is inapplicable for the case $m > 2$. For a formal statement regarding the above convergence rate, please refer to Lemma 12 in

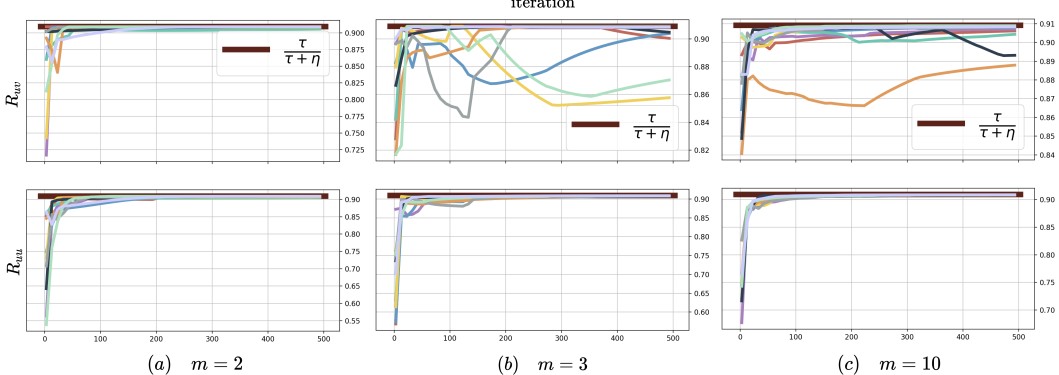

Figure 2: On the convergence rate of RSBP dual variables when $m \in \{2, 3, 10\}$. Lines with different colors present different runs (with the same values of $\tau = 0.1$ and $\eta = 0.01$). Other parameters are set as follows: $n = 10, C_i \sim \mathcal{U}[0.01, 1]^{n \times n}$.

Appendix C. Note that for $m \geq 3$, the result of Theorem 2 still holds if $\mathbf{u}^k$ and $\mathbf{v}^k$ converge at the rate of the order $(\frac{\tau}{\tau+\eta})^{k/2}$. So next we will take a closer look at this case to see whether the rate remains geometric.

**On $m \geq 3$:** In Figure 2, we plot the values of two ratios: $R_{uv} := \frac{\sum_{i=1}^{m} \|\Delta u_i^{k+1}\|_\infty}{\sum_{i=1}^{m} \|\Delta v_i^k\|_\infty}$ and $R_{uu} :=$ $\frac{\sum_{i=1}^{m} \|\Delta u_i^{k+1}\|_\infty}{\sum_{i=1}^{m} \|\Delta u_i^{k-1}\|_\infty}$. When $k$ is even, we have that $R_{uu} \leq \frac{\tau}{\tau+\eta}$ for all $m$, while the inequality $R_{uv} \leq \frac{\tau}{\tau+\eta}$ was only proved for the case $m = 2$. From this figure, both these bounds are true in all considered cases. However, while the bound on $R_{uv}$ (which is theoretically true for all $m$) is only tight when $m = 2$ and seems to be loose in several trials with larger values of $m$, the bound $R_{uu}$ (which is only showed for the case $m = 2$) appears to be tight in all reported scenarios. Thus, we conjecture that the geometric convergence rate at equation (16) may still hold for $m$ greater than 2. We leave the case $m \geq 3$ for the future work.

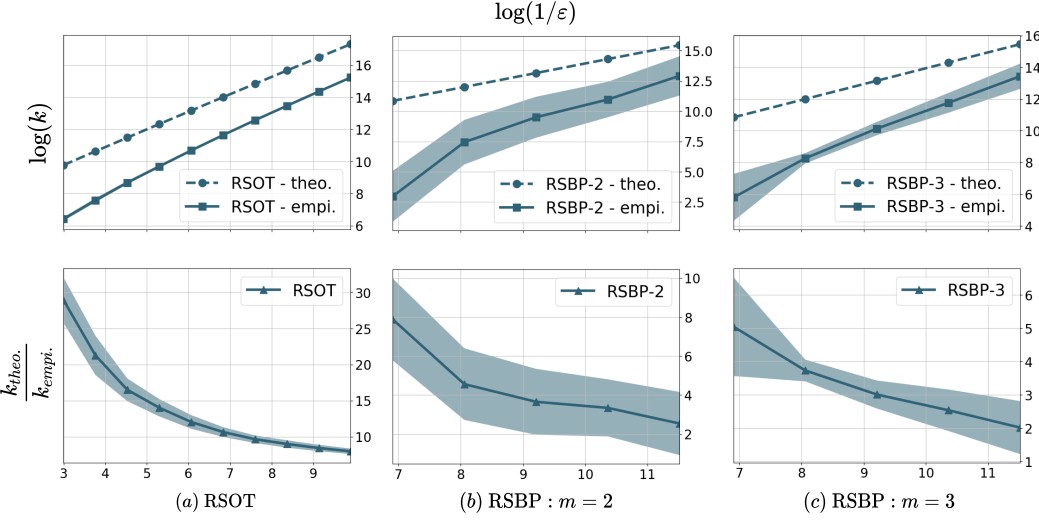

Figure 3: Runtime demonstration for $(a)$ ROBUST-SEMISINKHORN and $(b), (c)$ ROBUST-IBP algorithms. ***Top:*** The log value of the number of iterations computed in our theorems (dashed lines with circle marker) and the true number of iterations at which the algorithms achieve $\varepsilon$-approximations (solid lines with square marker). ***Bottom:*** The ratio between two values of the upper figures. Both the number of iterations (on the left) and $\varepsilon$ are plotted in the log domain, while the ratios (on the right) are computed with the original values.

# 5 Experiments

In this section, we provide numerical evidences regarding our presented complexities for ROBUST-SEMISINKHORN and ROBUST-IBP algorithms. We put additional experiments (including the runtime comparison of ROT/RSOT on synthetic and real datasets, as well as some applications for the studied robust formulations) in Appendix F. All the optimal solutions for convex problems in the following part are computed using the **cvxpy** library [1]. All the experiments are conducted on a server with 32 GB RAM, 8 cores Intel(R) Core(TM) i7-9700K and 1 GeForce RTX 2080 GPU.

**Runtime Demonstration:** For each algorithm, we investigate the number of iterations required to obtain an $\varepsilon$-approximation. We compare the theoretical values in Theorems 1 and 2 with the empirical values computed by running the corresponding algorithms to obtain the first iterations from where the algorithm always returns an $\varepsilon$-approximation.

*For RSOT*, we let $n = 100, \tau = 1$, generate entries of $C$ uniformly from the interval $[1, 50]$ and draw entries $a, b$ uniformly from $[0.1, 1]$ then normalizing them to form probability vectors. $\eta$ is set according to Theorem 1. For each $\varepsilon$ varying from $5 \times 10^{-2}$ to $5 \times 10^{-5}$, we calculate the number of theoretical and empirical iterations described above, as well as their ratio. This experiment is run 10 times and we report their mean and standard deviation values in Figure 3 $(a)$. We also carry out a similar experiment on MNIST data, which is reported in the Appendix F.

*For RSBP*, we run the ROBUSTIBP algorithm with the following setup: $n = 10; \tau = 1; \mathbf{p}_1, \ldots, \mathbf{p}_m$, $[\omega_1, \ldots, \omega_m]$ are randomly-initialized probability vectors; $\{C_i\}_{i=1}^m$ is a set of $n \times n$ matrices whose entries drawn uniformly in $[0.01, 0.1]$; five chosen values of $\varepsilon$ vary from $10^{-3}$ to $10^{-5}$ (which are relatively small compared to the optimal cost $f_{\mathrm{rsbp}}(\mathbf{X}^*)$ is about $0.019 \pm 0.001$ when $m = 2$ and is about $0.021 \pm 0.001$ when $m = 3$); and the corresponding values of $\eta$ are set according to Theorem 2. The results are shown in Figure 3 $(b)$ and $(c)$. Note that the complexity for the case $m \geq 3$ is still an open problem, and we use the formula in Theorem 2 to compute the (hypothetical) theoretical number of iterations in that case.

In all three experiments, it is noticeable that the ratios between theoretical and empirical values decrease as $\varepsilon \to 0$, indicating the our complexity bounds get tighter.

# 6 Conclusion

In the paper, we study the complexity of Sinkhorn-based algorithms for approximately solving robust versions of optimal transport between two discrete probability measures with at most $n$ components, and show that they return $\varepsilon$-approximated solutions in $\widetilde{\mathcal{O}}(n^2/\varepsilon)$ time. Low-rank approximation technique is also analysed to further reduce the dependency of these complexities on $n$, resulting in $\widetilde{\mathcal{O}}(nr^2 + nr/\varepsilon)$ complexities. Finally, we investigate a robust barycenter problem between $m$ probability measures and develop the IBP-based algorithm for solving it. When $m = 2$, the complexity of the ROBUSTIBP algorithm is proved to be at the order of $\widetilde{\mathcal{O}}(mn^2/\varepsilon)$, while in the case $m \geq 3$ we believe that a novel proof technique needs to be developed to establish the geometric convergence of the updates from the algorithm. We leave this direction for the future work.

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
