# Supplement to "On Robust Optimal Transport: Computational Complexity and Barycenter Computation"

In this supplementary material, we collect several proofs and remaining materials that are deferred from the main paper. In Appendix A, we introduce and recall necessary notations for the supplementary material. In Appendix B, we provide key lemmas and proofs for the computational complexity of robust semi-constrained optimal transport (RSOT), and those regarding ROT are in Appendix D. Appendix C is devoted to the lemmas and proofs for the computational complexity of robust semi-constrained barycenter (RSBP). We provide the proof for computational complexity of robust Sinkhorn algorithms via Nyström approximation in Appendix E. Finally, we present additional experiment studies with the proposed robust algorithms in Appendix F.

## A    Notations

This appendix aims to introduce some notations that will be used intensively in the subsequent parts of the appendix. We start with the meaning of notations for the general case, and those for remaining cases follow similarly (see the table content). First, we denote $f$ and $g$ to be the original objective and the corresponding entropic-regularized objective, respectively, and let $\widehat{X} := \arg\min f(X), X^* := \arg\min g(X)$. The sum of all elements in $X$ is $x := \|X\|_1$ (similarly, $x^* := \|X^*\|_1$). Regarding Sinkhorn algorithm, $u^k, v^k$ are the updates of the $k$-th iteration. The converged values for $u^k$ and $v^k$ (if exist) are denoted $u^*$ and $v^*$ respectively, i.e. $u^* := \lim_{k\to\infty} u^k, v^* := \lim_{k\to\infty} v^k$. Finally, for the ease of presentation, let us denote some quantites which will be frequently used in our proofs: $\Delta^k := \max\{\|u^k - u^*\|_\infty, \|v^k - v^*\|_\infty\}$, $R := \max\{\|\log(\mathbf{a})\|_\infty, \|\log(\mathbf{b})\|_\infty\} + \max\left\{\log(n), \frac{1}{\eta}\|C\|_\infty - \log(n)\right\}$, $\alpha := \|\mathbf{a}\|_1, \beta := \|\mathbf{b}\|_1$ and $\rho_i = \|\mathbf{p}_i\|_1$ for all $i \in [m]$.

| General | Robust Semi-OT | Unbalanced OT | Robust OT | Non-normalized RSBP | RSBP |
|---|---|---|---|---|---|
| $f : f(X) := \langle C, X\rangle + \tau \times \text{regularization}$ | $f_{\text{rsot}}$ | $f_{\text{rot}}$ | | $f_{\text{rsbp}}$ | |
| $g : g(X) := f(X) - \eta H(X)$ | $g_{\text{rsot}}$ | $g_{\text{rot}}$ | | $g_{\text{rsbp}}$ | |
| $\widehat{X} := \arg\min f(X)$ | $\widehat{X}_{\text{rsot}}$ | $\widehat{X}_{\text{uot}}$ | $\widehat{X}_{\text{rot}}$ | | $\widehat{\mathbf{X}}$ |
| $X^* := \arg\min g(X)$ | $X^*_{\text{rsot}}$ | $X^*_{\text{uot}}$ | $X^*_{\text{rot}}$ | $\bar{\mathbf{X}}^*$ | $\mathbf{X}^*$ |
| $x^* := \|X^*\|_1$ | 1 | $x^*_{\text{uot}}$ | 1 | $\bar{x}^*$ | 1 |
| $u^k, v^k$ ($k$-th Sinkhorn/IBP update) | $u^k_{\text{rsot}}, v^k_{\text{rsot}}$ | $u^k_{\text{uot}}, v^k_{\text{uot}}$ | | $\mathbf{u}^k = (u^k_1, \ldots, u^k_m), \mathbf{v}^k = (v^k_1, \ldots, v^k_m)$ | |
| $(u^*, v^*) := \lim_{k\to\infty}(u^k, v^k)$ | $u^*_{\text{rsot}}, v^*_{\text{rsot}}$ | $u^*_{\text{uot}}, v^*_{\text{uot}}$ | | $\mathbf{u}^* = (u^*_1, \ldots, u^*_m), \mathbf{v}^* = (v^*_1, \ldots, v^*_m)$ | |
| $\Delta^k := \max\{\|u^k - u^*\|_\infty, \|v^k - v^*\|_\infty\}$ | $\Delta^k_{\text{rsot}}$ | $\Delta^k_{\text{uot}}$ | | $\mathbf{\Delta}^k = (\Delta^k_1, \ldots, \Delta^k_m)$ | |
| $X^k := B(u^k, v^k)$ | $X^k_{\text{rsot}}$ | $X^k_{\text{uot}}$ | $X^k_{\text{rot}}$ | $\bar{\mathbf{X}}^k$ | $\mathbf{X}^k$ |
| $x^k := \|X^k\|_1$ | 1 (if $k$ is even) | $x^k_{\text{uot}}$ | 1 | $\bar{x}^k$ | 1 |

Table 1: Key notations for technical results and proofs in the supplementary material. When a term has a constant value (e.g. 1), we provide that value instead of the corresponding notation.

## B    Robust Semi-Constrained Optimal Transport: Omitted Proofs

This appendix is devoted to provide the lemmas and proofs for the computational complexity of robust semi-constrained optimal transport.

### B.1    Useful Lemmas

We first start with the following useful lemmas for the proof of Theorem 1.

**Lemma 3.** *The following inequalities are true for all positive $x_i, y_i, x, y$.*

*(a)* $\displaystyle \min_{1\le i\le n}\frac{x_i}{y_i} \le \frac{\sum_{i=1}^n x_i}{\sum_{i=1}^n y_i} \le \max_{1\le i\le n}\frac{x_i}{y_i}$,

*(b) If* $\max\left\{\dfrac{x}{y}, \dfrac{y}{x}\right\} \le 1 + \delta$, *then* $|x - y| \le \delta\min\{x, y\}$,

(c) $\left(1 + \dfrac{1}{x}\right)^{x+1} \geq e.$

*Proof of Lemma 3.*
**(a)** It follows from the assumption $x_i$ and $y_i$ are positive that

$$y_j \min_{1 \leq i \leq n}\left(\frac{x_i}{y_i}\right) \leq x_j \leq y_j \max_{1 \leq i \leq n}\left(\frac{x_i}{y_i}\right).$$

Taking the sum over $j$,

$$\sum_{j=1}^{n} y_j \min_{1 \leq i \leq n}\left(\frac{x_i}{y_i}\right) \leq \sum_{j=1}^{n} x_j \leq \sum_{j=1}^{n} y_j \max_{1 \leq i \leq n}\left(\frac{x_i}{y_i}\right).$$

This directly leads to the conclusion.
**(b)** WLOG assume that $x > y$, then

$$\frac{x}{y} \leq 1 + \delta \Rightarrow x \leq y + y\delta \Rightarrow |x - y| \leq y\delta.$$

**(c)** For the fourth inequality, taking the log of both sides, it is equivalent to

$$(x + 1)\left[\log(x + 1) - \log(x)\right] \geq 1.$$

By the mean value theorem, there exists a number $y$ between $x$ and $x + 1$ such that $\log(x + 1) - \log(x) = 1/y$, then $(x + 1)/y \geq 1$. $\qquad\square$

**Lemma 4.** *Let $a_{\mathrm{rsot}}^* = X_{\mathrm{rsot}}^* \mathbf{1}_n, a_{\mathrm{rsot}}^k = X_{\mathrm{rsot}}^k \mathbf{1}_n$ and $b_{\mathrm{rsot}}^* = (X_{\mathrm{rsot}}^*)^\top \mathbf{1}_n, b_{\mathrm{rsot}}^k = (X_{\mathrm{rsot}}^k)^\top \mathbf{1}_n$. Then,*

*(i)* $\left| \log\left(\dfrac{(a_{\mathrm{rsot}}^*)_i}{(a_{\mathrm{rsot}}^k)_i}\right) - \dfrac{(u_{\mathrm{rsot}}^*)_i - (u_{\mathrm{rsot}}^k)_i}{\eta} \right| \leq \max_{1 \leq j \leq n} \dfrac{(v_{\mathrm{rsot}}^*)_j - (v_{\mathrm{rsot}}^k)_j}{\eta},$

*(ii)* $\left| \log\left(\dfrac{(b_{\mathrm{rsot}}^*)_j}{(b_{\mathrm{rsot}}^k)_j}\right) - \dfrac{(v_{\mathrm{rsot}}^*)_j - (v_{\mathrm{rsot}}^k)_j}{\eta} \right| \leq \max_{1 \leq i \leq n} \dfrac{(u_{\mathrm{rsot}}^*)_i - (u_{\mathrm{rsot}}^k)_i}{\eta}.$

*Proof of Lemma 4.*
**(i)** From the definitions of $(a_{\mathrm{rsot}}^k)_i$ and $(a_{\mathrm{rsot}}^*)_i$, we have

$$\log\left(\frac{(a_{\mathrm{rsot}}^*)_i}{(a_{\mathrm{rsot}}^k)_i}\right) = \left(\frac{(u_{\mathrm{rsot}}^*)_i - (u_{\mathrm{rsot}}^k)_i}{\eta}\right) + \log\left(\frac{\sum_{j=1}^{n} \exp\left(\frac{(v_{\mathrm{rsot}}^*)_j - C_{ij}}{\eta}\right)}{\sum_{j=1}^{n} \exp\left(\frac{(v_{\mathrm{rsot}}^k)_j - C_{ij}}{\eta}\right)}\right).$$

The desired inequalities are equivalent to upper and lower bounds for the second term of the RHS. Applying part (a) of Lemma 3, we obtain

$$\min_{1 \leq j \leq n} \frac{(v_{\mathrm{rsot}}^*)_j - (v_{\mathrm{rsot}}^k)_j}{\eta} \leq \log\left(\frac{(a_{\mathrm{rsot}}^*)_i}{(a_{\mathrm{rsot}}^k)_i}\right) - \frac{(u_{\mathrm{rsot}}^*)_i - (u_{\mathrm{rsot}}^k)_i}{\eta} \leq \max_{1 \leq j \leq n} \frac{(v_{\mathrm{rsot}}^*)_j - (v_{\mathrm{rsot}}^k)_j}{\eta}.$$

**(ii)** Part (ii) are done similarly. $\qquad\square$

**Lemma 5.** *We have following upper bounds for the optimal solutions of RSOT's dual form, which is useful for the derivation of the convergence rate:*

$$\max\{\|u_{\mathrm{rsot}}^*\|_\infty, \|v_{\mathrm{rsot}}^*\|_\infty\} \leq (2\tau + \eta)R.$$

*Proof of Lemma 5.* First, we will show that

$$\|u_{\mathrm{rsot}}^*\|_\infty\left(\frac{1}{\tau} + \frac{1}{\eta}\right) \leq \frac{\|v_{\mathrm{rsot}}^*\|_\infty}{\eta} + R. \tag{17}$$

Since $u_{\mathrm{rsot}}^*$ is a fixed point of the update in Algorithm 1, we get

$$\frac{u_{\mathrm{rsot}}^*}{\tau} = \log(\mathbf{a}) - \log(a_{\mathrm{rsot}}^*). \tag{18}$$

Then,
$$\frac{(u^*_{\text{rsot}})_i}{\tau} = \log(\mathbf{a}_i) - \log\left[\sum_{j=1}^{n} \exp\left(\frac{(u^*_{\text{rsot}})_i + (v^*_{\text{rsot}})_j - C_{ij}}{\eta}\right)\right],$$

which is equivalent to
$$(u^*_{\text{rsot}})_i\left(\frac{1}{\tau} + \frac{1}{\eta}\right) = \log(\mathbf{a}_i) - \log\left[\sum_{j=1}^{n} \exp\left(\frac{(v^*_{\text{rsot}})_j - C_{ij}}{\eta}\right)\right].$$

The second term can be bounded as follows
$$\log\left[\sum_{j=1}^{n} \exp\left(\frac{(v^*_{\text{rsot}})_j - C_{ij}}{\eta}\right)\right] \geq \log(n) + \min_{1\leq j\leq n}\left\{\frac{(v^*_{\text{rsot}})_j - C_{ij}}{\eta}\right\} \geq \log(n) - \frac{\|v^*_{\text{rsot}}\|_\infty}{\eta} - \frac{\|C\|_\infty}{\eta},$$

and
$$\log\left[\sum_{j=1}^{n} \exp\left(\frac{(v^*_{\text{rsot}})_j - C_{ij}}{\eta}\right)\right] \leq \log(n) + \max_{1\leq j\leq n}\left\{\frac{(v^*_{\text{rsot}})_j - C_{ij}}{\eta}\right\} \leq \log(n) + \frac{\|v^*_{\text{rsot}}\|_\infty}{\eta},$$

thus leading to
$$\left|\log\left[\sum_{j=1}^{n} \exp\left(\frac{(v^*_{\text{rsot}})_j - C_{ij}}{\eta}\right)\right]\right| \leq \frac{\|v^*_{\text{rsot}}\|_\infty}{\eta} + \max\left\{\log(n), \frac{\|C\|_\infty}{\eta} - \log(n)\right\}. \tag{19}$$

Hence,
$$|(u^*_{\text{rsot}})_i|\left(\frac{1}{\eta} + \frac{1}{\tau}\right) \leq |\log(\mathbf{a}_i)| + \frac{\|v^*_{\text{rsot}}\|_\infty}{\eta} + \max\left\{\log(n), \frac{\|C\|_\infty}{\eta} - \log(n)\right\}.$$

Choosing $i$ such that $|(u^*_{\text{rsot}})_i| = \|u^*_{\text{rsot}}\|_\infty$, combining with the fact that
$$|\log(\mathbf{a}_i)| \leq \max\{\|\log(\mathbf{a})\|_\infty, \|\log(\mathbf{b})\|_\infty\},$$

we have
$$\|u^*_{\text{rsot}}\|_\infty\left(\frac{1}{\tau} + \frac{1}{\eta}\right) \leq \frac{\|v^*_{\text{rsot}}\|_\infty}{\eta} + R. \tag{20}$$

Next, we will prove that
$$\|v^*_{\text{rsot}}\|_\infty \leq \|u^*_{\text{rsot}}\|_\infty + \eta R.$$

Notice that $v^*_{\text{rsot}}$ is a fixed point of the update in Algorithm 1, we get $v^*_{\text{rsot}} = \eta\left[\frac{v^*_{\text{rsot}}}{\eta} + \log(\mathbf{b}) - \log(b^*_{\text{rsot}})\right]$, which implies that $\log(b^*_{\text{rsot}}) = \log(\mathbf{b})$. Therefore,

$$\log(\mathbf{b}_j) = \log\left[\sum_{i=1}^{n} \exp\left(\frac{(u^*_{\text{rsot}})_i + (v^*_{\text{rsot}})_j - C_{ij}}{\eta}\right)\right]$$
$$= \frac{(v^*_{\text{rsot}})_j}{\eta} + \log\left[\sum_{i=1}^{n} \exp\left(\frac{(u^*_{\text{rsot}})_i - C_{ij}}{\eta}\right)\right],$$

or equivalently,
$$\frac{(v^*_{\text{rsot}})_j}{\eta} = \log(\mathbf{b}_j) - \log\left[\sum_{i=1}^{n} \exp\left(\frac{(u^*_{\text{rsot}})_i - C_{ij}}{\eta}\right)\right].$$

Using the same arguments as for deriving equation (19), we obtain
$$\left|\log\left[\sum_{i=1}^{n} \exp\left(\frac{(u^*_{\text{rsot}})_i - C_{ij}}{\eta}\right)\right]\right| \leq \frac{\|u^*_{\text{rsot}}\|_\infty}{\eta} + \max\left\{\log(n), \frac{\|C\|_\infty}{\eta} - \log(n)\right\}.$$

It follows that
$$\frac{1}{\eta}|(v^*_{\text{rsot}})_j| \leq |\log(\mathbf{b}_j)| + \frac{\|u^*_{\text{rsot}}\|_\infty}{\eta} + \max\left\{\log(n), \frac{\|C\|_\infty}{\eta} - \log(n)\right\}.$$

Choosing $j$ such that $|(v_{\text{rsot}}^*)_j| = \|v_{\text{rsot}}^*\|_\infty$, and making use of the fact that

$$|\log(\mathbf{b}_j)| \leq \max\{\|\log(\mathbf{a})\|_\infty, \|\log(\mathbf{b})\|_\infty\},$$

we have

$$\|v_{\text{rsot}}^*\|_\infty \leq \|u_{\text{rsot}}^*\|_\infty + \eta R. \tag{21}$$

From equations (20) and (21), we get

$$\|u_{\text{rsot}}^*\|_\infty \left(\frac{1}{\tau} + \frac{1}{\eta}\right) \leq \frac{\|v_{\text{rsot}}^*\|_\infty}{\eta} + R \leq \frac{\|u_{\text{rsot}}^*\|_\infty}{\eta} + 2R,$$

which implies that

$$\|u_{\text{rsot}}^*\|_\infty \leq 2\tau R \leq (2\tau + \eta)R. \tag{22}$$

Therefore,

$$\|v_{\text{rsot}}^*\|_\infty \leq \|u_{\text{rsot}}^*\|_\infty + \eta R \leq (2\tau + \eta)R. \tag{23}$$

Combining equation (22) with equation (23), the proof is completed. $\qquad\square$

**Lemma 6.** *For any $k \geq 0$, the update $(u_{\text{rsot}}^{k+1}, v_{\text{rsot}}^{k+1})$ from Algorithm 1 satisfies the following bound*

$$\max\left\{\|u_{\text{rsot}}^{k+1} - u_{\text{rsot}}^*\|_\infty, \|v_{\text{rsot}}^{k+1} - v_{\text{rsot}}^*\|_\infty\right\} \leq \left(\frac{\tau}{\tau + \eta}\right)^{k/2} \times (2\tau + \eta)R. \tag{24}$$

*This establishes a geometric convergence rate for the dual variables in Algorithm 1.*

*Proof of Lemma 6.* We first consider the case when $k$ is even. From the update of $u_{\text{rsot}}^{k+1}$ in Algorithm 1, we have

$$(u_{\text{rsot}}^{k+1})_i = \frac{\eta\tau}{\tau + \eta}\left[\frac{(u_{\text{rsot}}^k)_i}{\eta} + \log(\mathbf{a}_i) - \log((a_{\text{rsot}}^k)_i)\right]$$

$$= \frac{\eta\tau}{\tau + \eta}\left\{\frac{(u_{\text{rsot}}^k)_i}{\eta} + \left[\log(\mathbf{a}_i) - \log((a_{\text{rsot}}^*)_i)\right] + \left[\log((a_{\text{rsot}}^*)_i) - \log((a_{\text{rsot}}^k)_i)\right]\right\}.$$

Using equation (18), the above equality is equivalent to

$$(u_{\text{rsot}}^{k+1})_i - (u_{\text{rsot}}^*)_i = \left[\eta \log\left(\frac{(a_{\text{rsot}}^*)_i}{(a_{\text{rsot}}^k)_i}\right) - ((u_{\text{rsot}}^*)_i - (u_{\text{rsot}}^k)_i)\right]\frac{\tau}{\tau + \eta}.$$

Applying Lemma 4, we get

$$|(u_{\text{rsot}}^{k+1})_i - (u_{\text{rsot}}^*)_i| \leq \max_{1 \leq j \leq n} |(v_{\text{rsot}}^k)_j - (v_{\text{rsot}}^*)_j|\frac{\tau}{\tau + \eta},$$

which implies that

$$\|u_{\text{rsot}}^{k+1} - u_{\text{rsot}}^*\|_\infty \leq \frac{\tau}{\tau + \eta}\|v_{\text{rsot}}^k - v_{\text{rsot}}^*\|_\infty. \tag{25}$$

From the update of $v_{\text{rsot}}^k$ in Algorithm 1, we have

$$(v_{\text{rsot}}^k)_j = (v_{\text{rsot}}^{k-1})_j + \eta \log\left(\frac{\mathbf{b}_j}{(b_{\text{rsot}}^{k-1})_j}\right) = (v_{\text{rsot}}^{k-1})_j + \eta \log\left(\frac{(b_{\text{rsot}}^*)_j}{(b_{\text{rsot}}^{k-1})_j}\right).$$

Subtracting $(v_{\text{rsot}}^*)_j$ from both sides and applying Lemma 4, one gets

$$|(v_{\text{rsot}}^k)_j - (v_{\text{rsot}}^*)_j| = \eta\left|\log\left(\frac{(b_{\text{rsot}}^*)_j}{(b_{\text{rsot}}^{k-1})_j}\right) - \frac{(v_{\text{rsot}}^*)_j - (v_{\text{rsot}}^{k-1})_j}{\eta}\right| \leq \|u_{\text{rsot}}^{k-1} - u_{\text{rsot}}^*\|_\infty.$$

This leads to

$$\|v_{\text{rsot}}^k - v_{\text{rsot}}^*\|_\infty \leq \|u_{\text{rsot}}^{k-1} - u_{\text{rsot}}^*\|_\infty. \tag{26}$$

Combining the two inequalities (25) and (26) yields

$$\|u_{\text{rsot}}^{k+1} - u_{\text{rsot}}^*\|_\infty \leq \frac{\tau}{\tau + \eta}\|u_{\text{rsot}}^{k-1} - u_{\text{rsot}}^*\|_\infty.$$

Repeating all the above arguments alternatively, we have

$$\|u_{\text{rsot}}^{k+1} - u_{\text{rsot}}^*\|_\infty \le \left(\frac{\tau}{\tau+\eta}\right)^{k/2} \|u_{\text{rsot}}^1 - u_{\text{rsot}}^*\|_\infty \le \left(\frac{\tau}{\tau+\eta}\right)^{k/2+1} \|v_{\text{rsot}}^0 - v_{\text{rsot}}^*\|_\infty$$

$$= \left(\frac{\tau}{\tau+\eta}\right)^{k/2+1} \|v_{\text{rsot}}^*\|_\infty.$$

Note that $v_{\text{rsot}}^{k+1} = v_{\text{rsot}}^k$ for $k$ even. Therefore, it is clear from (26) that

$$\|v_{\text{rsot}}^{k+1} - v_{\text{rsot}}^*\|_\infty \le \|u_{\text{rsot}}^{k-1} - u_{\text{rsot}}^*\|_\infty \le \left(\frac{\tau}{\tau+\eta}\right)^{k/2} \max\{\|u_{\text{rsot}}^*\|_\infty, \|v_{\text{rsot}}^*\|_\infty\}.$$

Thus,

$$\max\left\{\|u_{\text{rsot}}^{k+1} - u_{\text{rsot}}^*\|_\infty, \|v_{\text{rsot}}^{k+1} - v_{\text{rsot}}^*\|_\infty\right\} \le \left(\frac{\tau}{\tau+\eta}\right)^{k/2} \max\{\|u_{\text{rsot}}^*\|_\infty, \|v_{\text{rsot}}^*\|_\infty\}.$$

Similarly, the above result also holds for $k$ odd. Finally, applying Lemma 5, we obtain the conclusion.
□

## B.2 Detailed Proof of Theorem 1

Denoting

$$k_1 := \log\left(\frac{8R(2\tau+\eta)}{3\eta}\right) \Big/ \log\left(\frac{\tau+\eta}{\tau}\right), \qquad k_2 := \left(1+\frac{\tau}{\eta}\right)\log\left(\frac{3\tau R[2(\eta+\tau)+3R(2\tau+\eta)]}{\eta^2\log(n)}\right),$$

we will show that for all $k \ge 1 + 2\max\{k_1, k_2\}$ and $\eta = \varepsilon/U_{\text{rsot}}$, $X_{\text{rsot}}^k$ is an $\varepsilon$-approximation of the optimal solution $\widehat{X}_{\text{rsot}}$, that is

$$f_{\text{rsot}}(X_{\text{rsot}}^k) - f_{\text{rsot}}(\widehat{X}_{\text{rsot}}) \le \varepsilon = \eta U_{\text{rsot}}.$$

First, we can bound the above difference in the following way

$$\underbrace{f_{\text{rsot}}(X_{\text{rsot}}^k)}_{g_{\text{rsot}}(X_{\text{rsot}}^k)+\eta H(X_{\text{rsot}}^k)} - \underbrace{f_{\text{rsot}}(\widehat{X}_{\text{rsot}})}_{g_{\text{rsot}}(\widehat{X}_{\text{rsot}})+\eta H(\widehat{X}_{\text{rsot}})} \le \left[g_{\text{rsot}}(X_{\text{rsot}}^k) - g_{\text{rsot}}(X_{\text{rsot}}^*)\right] + \eta\left[H(X_{\text{rsot}}^k) - H(\widehat{X}_{\text{rsot}})\right],$$

where the inequality comes from $g_{\text{rsot}}(\widehat{X}_{\text{rsot}}) \ge g_{\text{rsot}}(X_{\text{rsot}}^*)$ which is the optimal value of the entropic ROT. Subsequently, the two terms in the right-hand side can be bounded separately as follows.

**Upper bound of** $H(X_{\text{rsot}}^k) - H(\widehat{X}_{\text{rsot}})$**.** The upper bound is obtained from the following inequalities for the entropy under the constraint $X \in \mathbb{R}_+^{n\times n}$ satisfying $\|X\|_1 = 1$,

$$1 \le H(X) \le 2\log(n) + 1. \tag{27}$$

Since $\widehat{X}_{\text{rsot}}$ is the optimal solution for RSOT, $\|\widehat{X}_{\text{rsot}}\|_1 = 1$. To derive the needed upper bound, we will show that $\|X_{\text{rsot}}^k\|_1 = 1$ for even $k$. Notice that when $k$ is even, at step $k-1$ of Algorithm 1 we update $v$, thus

$$v_{\text{rsot}}^k = \arg\min_v h_{\text{rsot}}(u_{\text{rsot}}^{k-1}, v) = \arg\min_v h_{\text{rsot}}(u_{\text{rsot}}^k, v), \qquad (\text{because } u_{\text{rsot}}^k = u_{\text{rsot}}^{k-1})$$

indicating that

$$X_{\text{rsot}}^k = \arg\min_{X\in\mathbb{R}_+^{n\times n}, X^\top \mathbf{1}_n = \mathbf{b}} g_{\text{rsot}}^k(X) := \langle C, X\rangle - \eta H(X) + \tau\mathbf{KL}(X\mathbf{1}_n \| \mathbf{a}^k),$$

where $\mathbf{a}^k := \exp\left(\frac{u_{\text{rsot}}^k}{\tau}\right) \odot (X\mathbf{1}_n)$ with $\odot$ denoting the element-wise multiplication. As a result, we have $\|X_{\text{rsot}}^k\|_1 = 1$ which leads to the following inequality

$$H(X_{\text{rsot}}^k) - H(\widehat{X}_{\text{rsot}}) \le 2\log(n). \tag{28}$$

**Upper bound of** $g_{\text{rsot}}(X_{\text{rsot}}^k) - g_{\text{rsot}}(X_{\text{rsot}}^*)$. The main idea for deriving this bound comes from the geometric convergence rate (i.e. Lemma 6). First, we represent the above difference by other quantities that are straightforward to bound. Reusing the definition of $g_{\text{rsot}}^k$ above, we utilize the following result regarding the optimal value of entropic RSOT

$$g_{\text{rsot}}(X_{\text{rsot}}^*) = -\eta - \tau(1-\alpha) + \langle v_{\text{rsot}}^*, b_{\text{rsot}}^* \rangle, \tag{29}$$

$$g_{\text{rsot}}^k(X_{\text{rsot}}^k) = -\eta - \tau(1-\alpha^k) + \langle v_{\text{rsot}}^k, b_{\text{rsot}}^k \rangle, \tag{30}$$

where $\alpha^k := \|\mathbf{a}^k\|_1$. We can see that these two equations have a similar form, and we can prove the first one by simple algebraic derivations as follows

$$\eta H(X_{\text{rsot}}^*) = -\eta \Big[ \sum_{i,j=1}^n (X_{\text{rsot}}^*)_{ij} \log(X_{\text{rsot}}^*)_{ij} + 1 \Big]$$

$$= -\eta \sum_{i,j=1}^n (X_{\text{rsot}}^*)_{ij} \frac{(u_{\text{rsot}}^*)_i + (v_{\text{rsot}}^*)_j - C_{ij}}{\eta} + \eta$$

$$= -\langle a_{\text{rsot}}^*, u_{\text{rsot}}^* \rangle - \langle b_{\text{rsot}}^*, v_{\text{rsot}}^* \rangle + \langle C, X_{\text{rsot}}^* \rangle + \eta.$$

The second equation comes from the fact that $(X_{\text{rsot}}^*)_{ij} = \exp\Big\{ \frac{(u_{\text{rsot}}^*)_i + (v_{\text{rsot}}^*)_j - C_{ij}}{\eta} \Big\}$. Then, we have

$$\langle C, X_{\text{rsot}}^* \rangle - \eta H(X_{\text{rsot}}^*) = -\eta + \langle a_{\text{rsot}}^*, u_{\text{rsot}}^* \rangle + \langle b_{\text{rsot}}^*, v_{\text{rsot}}^* \rangle.$$

$$\tau\mathbf{KL}(\underbrace{X_{\text{rsot}}^*\mathbf{1}_n}_{a_{\text{rsot}}^*}\|\mathbf{a}) = -\tau + \tau\alpha - \tau\Big\langle a_{\text{rsot}}^*, \log\Big(\frac{a_{\text{rsot}}^*}{\mathbf{a}}\Big)\Big\rangle$$

$$= -\tau(1-\alpha) - \langle a_{\text{rsot}}^*, u_{\text{rsot}}^* \rangle,$$

because $u_{\text{rsot}}^*$ satisfies the fixed-point equation: $\frac{u_{\text{rsot}}^*}{\tau} = \log\Big(\frac{a_{\text{rsot}}^*}{\mathbf{a}}\Big)$. The equation for $g_{\text{rsot}}(X_{\text{rsot}}^*)$ comes straight from adding the above two equations. Then the difference of interest can be written as

$$g(X_{\text{rsot}}^k) - g(X_{\text{rsot}}^*) = \big[ g(X_{\text{rsot}}^k) - g^k(X_{\text{rsot}}^k) \big] + \big[ g^k(X_{\text{rsot}}^k) - g(X_{\text{rsot}}^*) \big]$$

$$= \tau\Big\langle a_{\text{rsot}}^k, \log\Big(\frac{\mathbf{a}^k}{\mathbf{a}}\Big)\Big\rangle + \big( \langle v_{\text{rsot}}^k, b_{\text{rsot}}^k \rangle - \langle v_{\text{rsot}}^*, b_{\text{rsot}}^* \rangle \big). \tag{31}$$

Both terms above can be bounded with regards to $\Delta_{\text{rsot}}^k := \max\big\{ \|u_{\text{rsot}}^k - u_{\text{rsot}}^*\|_\infty, \|v_{\text{rsot}}^k - v_{\text{rsot}}^*\|_\infty \big\}$.
*On the first term in equation (31).* From the fixed-point result for $u$-updates, we have

$$\Big\| \log\Big(\frac{\mathbf{a}^k}{\mathbf{a}}\Big) \Big\|_\infty = \Big\| \frac{u_{\text{rsot}}^k - u_{\text{rsot}}^*}{\tau} - \log\Big(\frac{a_{\text{rsot}}^*}{a_{\text{rsot}}^k}\Big) \Big\|_\infty$$

$$\le \frac{1}{\tau}\|u_{\text{rsot}}^k - u_{\text{rsot}}^*\|_\infty + \Big\| \log\Big(\frac{a_{\text{rsot}}^*}{a_{\text{rsot}}^k}\Big) \Big\|_\infty$$

$$\le \frac{1}{\tau}\|u_{\text{rsot}}^k - u_{\text{rsot}}^*\|_\infty + \frac{1}{\eta}(\|u_{\text{rsot}}^k - u_{\text{rsot}}^*\|_\infty + \|v_{\text{rsot}}^k - v_{\text{rsot}}^*\|_\infty)$$

$$\le \Big(\frac{1}{\tau} + \frac{2}{\eta}\Big)\Delta_{\text{rsot}}^k,$$

$$\tau\Big\langle a_{\text{rsot}}^k, \log\Big(\frac{\mathbf{a}^k}{\mathbf{a}}\Big)\Big\rangle \le \tau\underbrace{\|a_{\text{rsot}}^k\|_1}_{=1}\Big\| \log\Big(\frac{\mathbf{a}^k}{\mathbf{a}}\Big) \Big\|_\infty \le \frac{2\tau+\eta}{\eta}\Delta_{\text{rsot}}^k. \tag{32}$$

*On the second term in equation (31).* We find that

$$\langle v_{\text{rsot}}^k, b_{\text{rsot}}^k \rangle - \langle v_{\text{rsot}}^*, b_{\text{rsot}}^* \rangle = \langle v_{\text{rsot}}^k - v_{\text{rsot}}^*, b_{\text{rsot}}^k \rangle - \langle v_{\text{rsot}}^*, b_{\text{rsot}}^* - b_{\text{rsot}}^k \rangle$$

$$\le \underbrace{\|b_{\text{rsot}}^k\|_1}_{=1}\underbrace{\|v_{\text{rsot}}^k - v_{\text{rsot}}^*\|_\infty}_{\le\Delta_{\text{rsot}}^k} + \underbrace{\|v_{\text{rsot}}^*\|_\infty}_{\le(2\tau+\eta)R}\|b_{\text{rsot}}^* - b_{\text{rsot}}^k\|_1.$$

Thus, we need an upper bound for $\|b_{\mathrm{rsot}}^* - b_{\mathrm{rsot}}^k\|_1$, i.e., $\ell_1$-norm of the difference between $b_{\mathrm{rsot}}^*$ and $b_{\mathrm{rsot}}^k$. Note that we have the following bound on their ratio (which is a direct result of Lemma 4)

$$\max_j \left\{ \frac{(b_{\mathrm{rsot}}^*)_j}{(b_{\mathrm{rsot}}^k)_j}, \frac{(b_{\mathrm{rsot}}^k)_j}{(b_{\mathrm{rsot}}^*)_j} \right\} \leq \exp\left( \frac{\|u_{\mathrm{rsot}}^k - u_{\mathrm{rsot}}^*\|_\infty + \|v_{\mathrm{rsot}}^k - v_{\mathrm{rsot}}^*\|_\infty}{\eta} \right) \leq \exp\left( \frac{2\Delta_{\mathrm{rsot}}^k}{\eta} \right).$$

Applying part (b) of Lemma 3, we obtain

$$\left| (b_{\mathrm{rsot}}^*)_j - (b_{\mathrm{rsot}}^k)_j \right| \leq \left[ \exp\left( \frac{2\Delta_{\mathrm{rsot}}^k}{\eta} \right) - 1 \right] \min_j \left\{ (b_{\mathrm{rsot}}^k)_j, (b_{\mathrm{rsot}}^*)_j \right\}.$$

$$\sum_{j=1}^n \left| (b_{\mathrm{rsot}}^k)_j - (b_{\mathrm{rsot}}^*)_j \right| \leq \left[ \exp\left( \frac{2\Delta_{\mathrm{rsot}}^k}{\eta} \right) - 1 \right] \underbrace{\sum_{j=1}^n \min\left\{ (b_{\mathrm{rsot}}^k)_j, (b_{\mathrm{rsot}}^*)_j \right\}}_{\leq \|b_{\mathrm{rsot}}^*\|_1 = 1} \leq \exp\left( \frac{2\Delta_{\mathrm{rsot}}^k}{\eta} \right) - 1.$$

Hence,

$$\|b_{\mathrm{rsot}}^* - b_{\mathrm{rsot}}^k\|_1 \leq \sum_{j=1}^n \left| (b_{\mathrm{rsot}}^k)_j - (b_{\mathrm{rsot}}^*)_j \right| \leq \exp\left( \frac{2\Delta_{\mathrm{rsot}}^k}{\eta} \right) - 1.$$

To remove the exponential operator, noting that for $k \geq 1 + 2k_1$, we have $\frac{\Delta_{\mathrm{rsot}}^k}{\eta} \leq \frac{3}{8}$. Thus, $\exp\left( \frac{2\Delta_{\mathrm{rsot}}^k}{\eta} \right) - 1 \leq \frac{3\Delta_{\mathrm{rsot}}^k}{\eta}$, and consequently $\|b_{\mathrm{rsot}}^* - b_{\mathrm{rsot}}^k\|_1 \leq \frac{3\Delta_{\mathrm{rsot}}^k}{\eta}$. Having this bound on $\|b_{\mathrm{rsot}}^* - b_{\mathrm{rsot}}^k\|_1$, we can completely bound the second term of interest as follows

$$\langle v_{\mathrm{rsot}}^k, b_{\mathrm{rsot}}^k \rangle - \langle v_{\mathrm{rsot}}^*, b_{\mathrm{rsot}}^* \rangle \leq \left[ 1 + \frac{3}{\eta}(2\tau + \eta)R \right] \Delta_{\mathrm{rsot}}^k. \tag{33}$$

Plugging the bounds (32) and (33) to equation (31), we obtain

$$g_{\mathrm{rsot}}(X_{\mathrm{rsot}}^k) - g_{\mathrm{rsot}}(X_{\mathrm{rsot}}^*) \leq \left[ 1 + \frac{2\tau + \eta}{\eta} + \frac{3}{\eta}(2\tau + \eta)R \right] \Delta_{\mathrm{rsot}}^k.$$

From this bound, we will show that

$$g_{\mathrm{rsot}}(X_{\mathrm{rsot}}^k) - g_{\mathrm{rsot}}(X_{\mathrm{rsot}}^*) \leq \eta \log(n). \tag{34}$$

From Lemma 6 we have $\Delta_{\mathrm{rsot}}^k \leq 3\tau \left( \frac{\tau}{\tau + \eta} \right)^{(k-1)/2} R$. Thus, we only need to prove that for $k \geq 2k_2 + 1$,

$$3\tau \left( \frac{\tau}{\tau + \eta} \right)^{(k-1)/2} \cdot R \cdot \left[ 1 + \frac{2\tau + \eta}{\eta} + \frac{3}{\eta}(2\tau + \eta)R \right] \leq \eta \log(n).$$

This form of inequality can be represented through the following lemma.

**Lemma 7.** *For $0 < s < 1$, if $D \geq s^2$ and $\kappa \geq \left( 1 + \frac{1}{s} \right) \log\left( \frac{D}{s^2} \right)$, then $D \leq s^2(1 + s)^\kappa$.*

*Proof of Lemma 7.* The statement comes directly from a chain of inequalities using Lemma 3c for $x = \frac{1}{s}$:

$$s^2(1 + s)^\kappa \geq s^2(1 + s)^{(1 + \frac{1}{s}) \log\left( \frac{D}{s^2} \right)}$$

$$\geq s^2 \exp\left\{ \log\left( \frac{D}{s^2} \right) \right\} = D.$$

$\square$

Applying Lemma 7 for $s = \frac{\eta}{\tau} \in (0, 1)$, $D = \frac{3R}{\tau \log(n)} \left[ 2(\tau + \eta) + 3R(2\tau + \eta) \right]$ and $\kappa = \frac{k-1}{2}$, we get the inequality (34). Combining the bounds (28) and (34), we obtain

$$f_{\mathrm{rsot}}(X_{\mathrm{rsot}}^k) - f_{\mathrm{rsot}}(\widehat{X}_{\mathrm{rsot}}) \leq \eta \log(n) + 2\eta \log(n) = 3\eta \log(n) \leq \eta U_{\mathrm{rsot}} = \varepsilon.$$

**The complexity of Algorithm 1.** By definition, $U_{\text{rsot}} = \mathcal{O}(\log(n))$. Applying part (c) of Lemma 3 with $x = \frac{\tau}{\eta}$, we have

$$\log\left(\frac{\tau + \eta}{\tau}\right) \geq \frac{1}{1 + \frac{\tau}{\eta}}.$$

Then, $k_1$ can be bounded as follows

$$k_1 = \frac{\log\left(\frac{8R(2\tau+\eta)}{3\eta}\right)}{\log\left(\frac{\tau+\eta}{\tau}\right)} \leq \log\left(\frac{8R(2\tau+\eta)}{3\eta}\right)\left(1 + \frac{\tau}{\eta}\right)$$

$$= \left(1 + \frac{\tau U_{\text{rsot}}}{\varepsilon}\right)\left[\log\left(\frac{U_{\text{rsot}}}{\varepsilon}\right) + \log\left(\frac{8}{3}\eta R\right) + \log\left(2\frac{\tau U_{\text{rsot}}}{\varepsilon} + 1\right)\right].$$

Assume that $R = \mathcal{O}\left(\frac{1}{\eta}\|C\|_\infty\right)$, we obtain

$$k_1 = \mathcal{O}\left(\frac{\tau\log(n)}{\varepsilon}\left[\log\left(\frac{\log(n)}{\varepsilon}\right) + \log(\|C\|_\infty) + \log\left(\frac{\tau\log(n)}{\varepsilon}\right)\right]\right)$$

$$= \mathcal{O}\left(\tau\frac{\log(n)}{\varepsilon}\left[\log(\|C\|_\infty) + \log(\log(n)) + \log(\tau) + \log\left(\frac{1}{\varepsilon}\right)\right]\right). \tag{35}$$

Next, let us consider

$$k_2 = \left(1 + \frac{\tau}{\eta}\right)\left[\log(3\tau R) + \log(2(\tau + \eta) + 3R(2\tau + \eta)) + 2\log\left(\frac{1}{\eta}\right) - \log(\log(n))\right]$$

$$\leq \left(1 + \frac{\tau}{\eta}\right)\left[\log(3R) + \log(4 + 9R) + 2\log(\tau) + 2\log\left(\frac{1}{\eta}\right) - \log(\log(n))\right]$$

$$\leq \left(1 + \frac{\tau U_{\text{rsot}}}{\varepsilon}\right)\left[\log(3\eta R) + 2\log(9\eta R) + 2\log(\tau) + 5\log\left(\frac{U_{\text{rsot}}}{\varepsilon}\right) - \log(\log(n))\right].$$

Thus,

$$k_2 = \mathcal{O}\left(\tau\frac{\log(n)}{\varepsilon}\left[\log(\|C\|_\infty) + \log(\tau) + 5\log\left(\frac{\log(n)}{\varepsilon}\right) - \log(\log(n))\right]\right)$$

$$= \mathcal{O}\left(\tau\frac{\log(n)}{\varepsilon}\left[\log(\|C\|_\infty) + \log(\tau) + \log(\log(n)) + \log\left(\frac{1}{\varepsilon}\right)\right]\right). \tag{36}$$

Equations (35) and (36) imply that

$$k = \mathcal{O}\left(\tau\left[\frac{\log(n)}{\varepsilon}\right]\left[\log(\|C\|_\infty) + \log(\tau) + \log(\log(n)) + \log\left(\frac{1}{\varepsilon}\right)\right]\right).$$

Multiplying the above quantity with $\mathcal{O}(n^2)$ arithmetic operations per iteration, we obtain the final complexity. As a consequence, we reach the conclusion of Theorem 1.

## C Robust Semi-Constrained Barycenter: Omitted Proofs

In this appendix, we provide some useful lemmas and proofs for deriving the computational complexity of the robust semi-constrained barycenter problem.

### C.1 Useful Lemmas

**Lemma 8.** *The dual form of entropic RSBP in* (11) *without constraints* $\|X_i\|_1 = 1$ *for all* $i \in [m]$ *is given by*

$$\min_{\mathbf{u},\mathbf{v}:\sum_{i=1}^m \omega_i v_i = \mathbf{0}_n} \bar{h}_{rsbp}(\mathbf{u},\mathbf{v}) := \sum_{i=1}^m \omega_i\left(\eta\|B(u_i, v_i; C_i)\|_1 + \tau\langle e^{-u_i/\tau}, \mathbf{p}_i\rangle\right).$$

*Proof of Lemma 8.* First, we rewrite the objective function (11) as follows

$$\min_{\substack{X_i \in \mathbb{R}_+^{n \times n}, X_i \mathbf{1}_n = \mathbf{y}_i, \forall i \in [m]; \\ X_i^\top \mathbf{1}_n = X_{i+1}^\top \mathbf{1}_n, \forall i \in [m-1]}} \sum_{i=1}^m \omega_i \left[ \langle C_i, X_i \rangle - \eta H(X_i) + \tau \mathbf{KL}(\mathbf{y}_i \| \mathbf{p}_i) \right]. \tag{37}$$

The Lagrangian function for the above problem is equal to

$$\sum_{i=1}^m \left( \omega_i [\langle C_i, X_i \rangle - \eta H(X_i) + \tau \mathbf{KL}(\mathbf{y}_i \| \mathbf{p}_i)] - \lambda_i^\top (X_i \mathbf{1}_n - \mathbf{y}_i) - \mu_i^\top (X_{i+1}^\top \mathbf{1}_n - X_i^\top \mathbf{1}_n) \right)$$

$$= \sum_{i=1}^m \left( \omega_i [\langle C_i, X_i \rangle - \eta H(X_i) + \tau \mathbf{KL}(\mathbf{y}_i \| \mathbf{p}_i)] - \lambda_i^\top (X_i \mathbf{1}_n - \mathbf{y}_i) - (\mu_{i-1} - \mu_i)^\top X_i^\top \mathbf{1}_n \right),$$

where $\lambda_i, \mu_i \in \mathbb{R}^n$ for all $i \in [m]$ with convention $\mu_0 = \mu_m = \mathbf{0}_n$. Using the change of variables $u_i = \lambda_i / \omega_i$ and $v_i = (\mu_{i-1} - \mu_i)/\omega_i$, we have $\sum_{i=1}^m \omega_i v_i = \mathbf{0}_n$ which allows to uniquely reconstruct $\mu_1, \ldots, \mu_m$. Then, the problem (37) is equivalent to

$$\max_{\substack{\mathbf{u}, \mathbf{v} \\ \sum_{i=1}^m \omega_i v_i = \mathbf{0}_n}} \min_{\substack{X_i \in \mathbb{R}^{n \times n}, \forall i \in [m] \\ \mathbf{y}_i \in \mathbb{R}^n, \forall i \in [m]}} \sum_{i=1}^m \omega_i [\langle C_i, X_i \rangle - \eta H(X_i) + \tau \mathbf{KL}(\mathbf{y}_i \| \mathbf{p}_i)$$

$$- u_i^\top (X_i \mathbf{1}_n - \mathbf{y}_i) - v_i^\top X_i^\top \mathbf{1}_n] \tag{38}$$

It can be verified that for all $i \in [m]$,

$$\min_{\mathbf{y}_i \in \mathbb{R}^n} \tau \mathbf{KL}(\mathbf{y}_i \| \mathbf{p}_i) + u_i^\top \mathbf{y}_i = -\tau \left\langle e^{-u_i/\tau}, \mathbf{p}_i \right\rangle + \mathbf{p}_i^\top \mathbf{1}_n.$$

Moreover, the objective function of the optimization problem

$$\min_{X_i \in \mathbb{R}^{n \times n}} \langle C_i, X_i \rangle - u_i^\top X_i \mathbf{1}_n - v_i^\top X_i^\top \mathbf{1}_n - \eta H(X_i)$$

is strongly convex. Thus, it has an unique optimal solution which could be directly calculated as $\bar{X}_i = B(u_i, v_i; C_i)$. Therefore,

$$\min_{X_i \in \mathbb{R}^{n \times n}} \langle C_i, X_i \rangle - u_i^\top X_i \mathbf{1}_n - v_i^\top X_i^\top \mathbf{1}_n - \eta H(X_i) = -\eta \| B(u_i, v_i; C_i) \|_1.$$

Collecting all of the above results, the optimization problem (38) turns into

$$\max_{\substack{\mathbf{u}, \mathbf{v} \\ \sum_{i=1}^m \omega_i v_i = \mathbf{0}_n}} \sum_{i=1}^m \omega_i \left( -\eta \| B(u_i, v_i; C_i) \|_1 - \tau \left\langle e^{-u_i/\tau}, \mathbf{p}_i \right\rangle + \mathbf{p}_i^\top \mathbf{1}_n \right)$$

$$= \min_{\substack{\mathbf{u}, \mathbf{v} \\ \sum_{i=1}^m \omega_i v_i = \mathbf{0}_n}} \sum_{i=1}^m \omega_i \left( \eta \| B(u_i, v_i; C_i) \|_1 + \tau \left\langle e^{-u_i/\tau}, \mathbf{p}_i \right\rangle \right).$$

We have thus proved our claim. □

Next, we will derive formulas for the updates $(\mathbf{u}^k, \mathbf{v}^k)$ of Algorithm 2 in the following lemma. Assume that at iteration $k$ where $k$ is even, $\mathbf{u}^{k+1}$ was found by minimizing the function $\bar{h}_{\text{rsbp}}$ given $\mathbf{v}^k$ and simply keep $\mathbf{v}^{k+1} = \mathbf{v}^k$ while for odd $k$, we do vice versa. In particular,

$$\mathbf{u}^{k+1} = \arg\min_{\mathbf{u}} \bar{h}_{\text{rsbp}}(\mathbf{u}, \mathbf{v}^k), \qquad \mathbf{v}^{k+1} = \mathbf{v}^k \quad \text{if k is even;}$$

$$\mathbf{v}^{k+1} = \arg\min_{\mathbf{v}: \sum_{i=1}^m \omega_i v_i = \mathbf{0}_n} \bar{h}_{\text{rsbp}}(\mathbf{u}^k, \mathbf{v}), \qquad \mathbf{u}^{k+1} = \mathbf{u}^k \quad \text{if k is odd.}$$

Let $\bar{\mathbf{X}}^k = (\bar{X}_1^k, \ldots, \bar{X}_m^k)$ be the non-normalized output at $k$-th iteration of Algorithm 2. For the ease of presentation, let us denote $a_i^k = \bar{X}_i^k \mathbf{1}_n$ and $b_i^k = (\bar{X}_i^k)^\top \mathbf{1}_n$ for all $i \in [m]$.

**Lemma 9.** *In Algorithm 2, the updates $(\mathbf{u}^k, \mathbf{v}^k)$ admit the following form*

$$u_i^{k+1} = \frac{\eta\tau}{\eta+\tau}\left[\frac{u_i^k}{\eta} + \log(\mathbf{p}_i) - \log(a_i^k)\right] \qquad \text{if } k \text{ is even;} \tag{39}$$

$$v_i^{k+1} = \eta\left[\frac{v_i^k}{\eta} - \log(b_i^k) - \sum_{t=1}^m \omega_t\big(\frac{v_t^k}{\eta} - \log(b_t^k)\big)\right] \quad \text{if } k \text{ is odd,} \tag{40}$$

*for all $i \in [m]$.*

*Proof of Lemma 9.* For $k$ even, by setting the gradients of $\bar{h}_{\text{rsbp}}$ with respect to $u_i$ to 0 given fixed $\mathbf{v}^k$, the update $u_i^k$ satisfies

$$\exp\left(\frac{(u_i^{k+1})_j}{\eta}\right)\sum_{l=1}^n \exp\left(\frac{(v_i^k)_l - (C_i)_{jl}}{\eta}\right) = \exp\left(-\frac{(u_i^{k+1})_j}{\tau}\right)\mathbf{p}_i \quad \text{for all } j \in [n].$$

Multiplying both sides by $\exp\left(\frac{(u_i^k)_j}{\eta}\right)$, we get

$$\exp\left(\frac{(u_i^{k+1})_j}{\eta}\right)(a_i^k)_j = \exp\left(\frac{(u_i^k)_j}{\eta}\right)\exp\left(-\frac{(u_i^{k+1})_j}{\tau}\right)\mathbf{p}_i \quad \text{for all } j \in [n].$$

Taking logarithm of the above equation and simplifying the result lead to the equality (39). For $k$ odd, recall that $\mathbf{v}^{k+1} = \arg\min_{\mathbf{v}:\sum_{i=1}^m \omega_i v_i = \mathbf{0}_n} \bar{h}_{\text{rsbp}}(\mathbf{u}^k, \mathbf{v})$, which also means that

$$\mathbf{v}^{k+1} = \arg\min_{\mathbf{v}} \sum_{i=1}^m \omega_i\left(\eta\|B(u_i^k, v_i; C_i)\|_1 + \tau\big\langle e^{-u_i^k/\tau}, \mathbf{p}_i\big\rangle\right) + \gamma^\top\left(\sum_{i=1}^m \omega_i v_i\right),$$

where $\gamma \in \mathbb{R}^n$ is a vector of Lagrange multipliers. Taking the derivatives of the above objective function with respect to $v_i$,

$$\exp\left(\frac{v_i^{k+1}}{\eta}\right) \odot A_i^k + \gamma = \mathbf{0}_n$$

$$\Leftrightarrow \frac{v_i^{k+1}}{\eta} + \log(A_i^k) = \log(-\gamma), \tag{41}$$

where $A_i^k = \left(\sum_{j=1}^n \exp\left\{\frac{(u_i^k)_j - (C_i)_{jl}}{\eta}\right\}\right)_{l=1}^n$. Subsequently, taking sum over $i$ and using the fact that $\sum_{i=1}^m \omega_i v_i^{k+1} = 0$, we obtain $\log(-\gamma) = \sum_{i=1}^m \omega_i \log(A_i^k)$. Plugging this result in equation (41), we obtain

$$\frac{v_i^{k+1}}{\eta} = \sum_{t=1}^m \omega_t \log(A_t^k) - \log(A_i^k)$$

$$= \frac{v_i^k}{\eta} - \log\left(A_i^k \odot \exp\left(\frac{v_i^k}{\eta}\right)\right) + \sum_{t=1}^m \omega_t\left[\log\left(A_t^k \odot \exp\left(\frac{v_t^k}{\eta}\right)\right) - \frac{v_t^k}{\eta}\right]$$

$$= \frac{v_i^k}{\eta} - \log(b_i^k) - \sum_{t=1}^m \omega_t\left(\frac{v_t^k}{\eta} - \log(b_t^k)\right).$$

Hence, the proof is completed. $\qquad\square$

**Lemma 10.** *Reusing the definition of the function $g_{\text{rsot}}$ in equation (6), we have the following property which is useful for the proofs of subsequent lemmas*

$$g_{\text{rsot}}(tX) = t g_{\text{rsot}}(X) + \tau(1-t)\alpha + (\tau+\eta)xt\log(t),$$

*for any $X \in \mathbb{R}_+^{n\times n}$ and $t \in \mathbb{R}^+$ where $x = \|X\|_1$.*

*Proof of Lemma 10.* By the definition of $g_{\text{rsot}}$, one has

$$g_{\text{rsot}}(tX) = \langle C, tX \rangle + \tau \mathbf{KL}(tX\mathbf{1}_n || \mathbf{a}) - \eta H(tX).$$

For the KL term of $g_{\text{rsot}}(tX)$, by denoting $a := X\mathbf{1}_n$, we get

$$
\begin{aligned}
\mathbf{KL}(tX\mathbf{1}_n || \mathbf{a}) &= \sum_{i=1}^n ta_i \log\left(\frac{a_i}{\mathbf{a}_i}\right) - \sum_{i=1}^n ta_i + \sum_{i=1}^n \mathbf{a}_i \\
&= \sum_{i=1}^n ta_i \left[\log\left(\frac{a_i}{\mathbf{a}_i}\right) + \log(t)\right] - tx + \alpha \\
&= t \sum_{i=1}^n \left[a_i \log\left(\frac{a_i}{\mathbf{a}_i}\right) - a_i + \mathbf{a}_i\right] + (1-t)\alpha + xt\log(t) \\
&= tg_{\text{rsot}}(X) + \tau(1-t)\alpha + (\tau + \eta)xt\log(t).
\end{aligned}
$$

For the entropic term, it can be verified that

$$-H(tX) = \sum_{i,j=1}^n tX_{ij}(\log(tX_{ij}) - 1) = \sum_{i,j=1}^n tX_{ij}(\log(X_{ij}) - 1) + xt\log(t) = -tH(X) + xt\log(t).$$

Collecting all of the above results, we obtain the conclusion. $\qquad\square$

**Remark 3.** *Notice that when $k$ is even, at step $(k-1)$-th of Algorithm 2, $\{v_i^k\}_{i=1}^m$ is found by minimizing the dual function* (15) *given $\{\mathbf{p}_i\}_{i=1}^m$ and fixed $\{u_i^{k-1}\}_{i=1}^m$, and remain $\{u_i^k\}_{i=1}^m = \{u_i^{k-1}\}_{i=1}^m$. Thus, $\bar{\mathbf{X}}^k$ is the optimal solution of*

$$\min_{X_1,\ldots,X_m \in \mathbb{R}_+^{n \times n}} g_{\text{rsbp}}^k(X_1,\ldots,X_m) := \sum_{i=1}^m \omega_i \left[\langle C_i, X_i \rangle + \tau \mathbf{KL}(X_i\mathbf{1}_n || \mathbf{p}_i^k) - \eta H(X_i)\right]$$

$$\text{s.t. } X_i^\top \mathbf{1}_n = X_{i+1}^\top \mathbf{1}_n \text{ for all } i \in [m-1],$$

*where $\mathbf{p}_i^k = \exp\left(\frac{u_i^k}{\tau}\right) \odot (\bar{X}_i^k \mathbf{1}_n)$ with $\odot$ denoting element-wise multiplication. The constraints $X_i^\top \mathbf{1}_n = X_{i+1}^\top \mathbf{1}_n$ for all $i \in [m-1]$ imply that $\|\bar{X}_i^k\|_1 = \|\bar{X}_{i+1}^k\|_1$ for any $i \in [m-1]$. Recall that $\bar{\mathbf{X}}^*$ is the optimizer of $g_{\text{rsbp}}$ with the feasible set $\mathcal{D}(\mathbf{X})$. By using similar arguments, we also have $\|\bar{X}_i^*\|_1 = \|\bar{X}_{i+1}^*\|_1$ for all $i \in [m-1]$. Denote $\bar{x}^k = \|\bar{X}_1^k\|_1$ for $k$ even and $\bar{x}^* = \|\bar{X}_1^*\|_1$, we will derive the upper bound of these quantities in the following lemma.*

**Lemma 11.** *The upper bounds of $\bar{x}^k$ and $\bar{x}^*$ are derived as follows*

(i) $\bar{x}^* \leq 3 + \dfrac{1}{\log(n)}$;

(ii) $\bar{x}^k \leq \dfrac{3}{2}\left(3 + \dfrac{1}{\log(n)}\right), \quad$ *for all even* $k \geq 2 + 2\left(\frac{\tau}{\eta} + 1\right)\log\left(\frac{4R_{\text{rsbp}}\tau^2}{\eta^2}\right)$.

*Proof of Lemma 11.*
**(i)** Consider the function $g_{\text{rsbp}}(t\bar{\mathbf{X}}^*)$ where $t \in \mathbb{R}^+$,

$$
\begin{aligned}
g_{\text{rsbp}}(t\bar{\mathbf{X}}^*) &= \sum_{i=1}^m \omega_i g_{\text{rsot}}(t\bar{X}_i^k; \mathbf{p}_i, C_i) \\
&= \sum_{i=1}^m \omega_i \left[tg_{\text{rsot}}(\bar{X}_i^k; \mathbf{p}_i, C_i) + \tau(1-t) + (\tau + \eta)\bar{x}^k t\log(t)\right] \\
&= tg_{\text{rsbp}}(\bar{\mathbf{X}}^*) + \tau(1-t) + (\tau + \eta)t\log(t)\bar{x}^*. \qquad\qquad (42)
\end{aligned}
$$

The second equality is due to Lemma 10. Taking the derivative of $g_{\text{rsbp}}(t\bar{\mathbf{X}}^*)$ with respect to $t$,

$$\partial_t g_{\text{rsbp}}(t\bar{\mathbf{X}}^k) = g_{\text{rsbp}}(\bar{\mathbf{X}}^*) - \tau + (\tau + \eta)(1 + \log(t))\bar{x}^*.$$

Since $g_{\mathrm{rsbp}}(t\bar{\mathbf{X}}^*)$ attains its minimum at $t = 1$, we obtain

$$g_{\mathrm{rsbp}}(\bar{\mathbf{X}}^*) + (\tau + \eta)\bar{x}^* = \tau. \tag{43}$$

By using the facts $g_{\mathrm{rsbp}}(\bar{\mathbf{X}}^*) \geq -\eta \sum_{i=1}^{m} \omega_i H(\bar{X}_i^*)$ and $H(\bar{X}_i^*) \leq 2\bar{x}^* \log(n) + \bar{x}^* - \bar{x}^* \log(\bar{x}^*)$, we have

$$
\begin{aligned}
\tau - (\tau + \eta)\bar{x}^* &\geq -\eta \sum_{i=1}^{m} \omega_i H(\bar{X}_i^*) \\
&\geq \eta \sum_{i=1}^{m} \omega_i \left[ -2\bar{x}^* \log(n) - \bar{x}^* + \bar{x}^* \log(\bar{x}^*) \right] \\
&= \eta \left[ -2\bar{x}^* \log(n) - \bar{x}^* + \bar{x}^* \log(\bar{x}^*) \right].
\end{aligned}
$$

It follows from the inequalities $z \log(z) \geq z - 1$ that

$$\tau \geq \eta \bar{x}^* \log(\bar{x}^*) + (\tau - 2\eta \log(n))\bar{x}^* \geq \eta \bar{x}^* - \eta + (\tau - 2\eta \log(n))\bar{x}^*.$$

Then, combining the above result and the inequality $3\eta \log(n) \leq \tau$, we get

$$\bar{x}^* \leq \frac{\tau + \eta}{\eta + \tau - 2\eta \log(n)} \leq 3 + \frac{1}{\log(n)}.$$

**(ii)** First, let us denote

$$\Delta_i^k = \max \left\{ \|u_i^k - u_i^*\|_\infty, \|v_i^k - v_i^*\|_\infty \right\}.$$

From Lemma 12, we have

$$\Delta_i^{k+1} \leq \tau \left( \frac{\tau}{\tau + \eta} \right)^{k/2} R_{\mathrm{rsbp}}.$$

Next, we will prove that $\Delta_i^{k+1} \leq \frac{\eta^2}{4\tau}$ for all even $k \geq 2 \left( \frac{\tau}{\eta} + 1 \right) \log \left( \frac{4R_{\mathrm{rsbp}}\tau^2}{\eta^2} \right)$ $i \in [m]$, which is equivalent to

$$
\begin{aligned}
\tau \left( \frac{\tau}{\tau + \eta} \right)^{k/2} R_{\mathrm{rsbp}} &\leq \frac{\eta^2}{4\tau} \\
\Leftrightarrow \left( \frac{\tau + \eta}{\tau} \right)^{k/2} \frac{\eta^2}{\tau^2} &\geq 4R_{\mathrm{rsbp}} \\
\Leftrightarrow (1 + s)^{k/2} s &\geq 4R_{\mathrm{rsbp}},
\end{aligned}
$$

where $s = \frac{\eta}{\tau}$. Let $t = 1 + \frac{\log(4R_{\mathrm{rsbp}})}{2 \log(\frac{1}{s})}$. Since $4R_{\mathrm{rsbp}} \geq 8 \log(n) \geq \frac{\eta^2}{\tau^2} = s^2$, therefore, $t > 1 + \frac{2 \log(s)}{2 \log(\frac{1}{s})} = 0$. Due to the fact that $\frac{k}{2} \geq \left( \frac{\tau}{\eta} + 1 \right) \log \left( \frac{4R_{\mathrm{rsbp}}\tau^2}{\eta^2} \right) = \left( 1 + \frac{1}{s} \right)(2t) \log \left( \frac{1}{s} \right) > 0$, we obtain

$$
\begin{aligned}
s^2 (1 + s)^{k/2} &\geq s^2 (1 + s)^{(\frac{1}{s} + 1) 2 \log(\frac{1}{s}) t} \\
&\geq s^2 \exp \left\{ 2 \log(1/s) t \right\} \\
&= \frac{1}{s^{2t-2}} = \frac{1}{s^{\log(4R_{\mathrm{rsbp}}) / \log(1/s)}} = \frac{1}{s^{- \log_s(4R_{\mathrm{rsbp}})}} = 4R_{\mathrm{rsbp}}.
\end{aligned}
$$

Therefore, $\max_{1 \leq i \leq m} \Delta_i^{k+1} \leq \frac{\eta^2}{4\tau} \leq \frac{1}{8}$. Then, by using the same arguments as part (b) of Lemma 5 in [27], we get

$$|\bar{x}^k - \bar{x}^*| \leq \frac{3}{\eta} \Delta_1^k \min \left\{ \bar{x}^k, \bar{x}^* \right\}. \tag{44}$$

Note that $u_1^k = u_1^{k-1}$ and $v_1^{k+1} = v_1^k$ for even $k$, hence, $\Delta_1^k \leq \max\{\Delta_1^{k-1}, \Delta_1^{k+1}\} \leq \frac{\eta^2}{4\tau}$. As a result,

$$\bar{x}^k \leq \left( 1 + \frac{3}{\eta} \Delta_1^k \right) \bar{x}^* \leq \frac{3}{2} \bar{x}^* \leq \frac{3}{2} \left( 3 + \frac{1}{\log(n)} \right).$$

We have thus proved our claim. $\qquad \square$

## C.2 Proof of Lemma 2

From the constraints $X_i^\top \mathbf{1}_n = X_{i+1}^\top \mathbf{1}_n$ for all $i \in [m-1]$ in $\mathcal{D}(\mathbf{X})$, we have that $\|\bar{X}_i^*\|_1$ is equal to each other for all $i \in [m]$ and denote $\bar{x}^* = \|\bar{X}_1^*\|_1$. Applying Lemma 10, we get

$$
\begin{aligned}
g_{\text{rsbp}}(\bar{\mathbf{X}}^*) &= \sum_{i=1}^m \omega_i g_{\text{rsot}}\left(\bar{X}_i^*; \mathbf{p}_i, C_i\right) \\
&= \sum_{i=1}^m \omega_i g_{\text{rsot}}\left(\bar{x}^* \frac{\bar{X}_i^*}{\bar{x}^*}; \mathbf{p}_i, C_i\right) \\
&= \sum_{i=1}^m \omega_i \left[\bar{x}^* g_{\text{rsot}}\left(\frac{\bar{X}_i^*}{\bar{x}^*}; \mathbf{p}_i, C_i\right) + \tau(1-\bar{x}^*)\rho_i + (\tau+\eta)\bar{x}^* \log(\bar{x}^*)\right] \\
&= \bar{x}^* g_{\text{rsbp}}\left(\frac{\bar{\mathbf{X}}^*}{\bar{x}^*}\right) + \tau(1-\bar{x}^*)\sum_{i=1}^m \omega_i \rho_i + (\tau+\eta)\bar{x}^* \log(\bar{x}^*).
\end{aligned}
$$

Similarly, applying Lemma 10, we obtain

$$
\begin{aligned}
g_{\text{rsbp}}(x^* \mathbf{X}^*) &= \sum_{i=1}^m \omega_i g_{\text{rsot}}\left(\bar{x}^* X_i^*; \mathbf{p}_i, C_i\right) \\
&= \sum_{i=1}^m \omega_i \left[\bar{x}^* g_{\text{rsot}}(X_i^*; \mathbf{p}_i, C_i) + \tau(1-\bar{x}^*)\rho_i + (\tau+\eta)\bar{x}^* \log(\bar{x}^*)\right] \\
&= \bar{x}^* g_{\text{rsbp}}(\mathbf{X}^*) + \tau(1-\bar{x}^*)\sum_{i=1}^m \omega_i \rho_i + (\tau+\eta)\bar{x}^* \log(\bar{x}^*).
\end{aligned}
$$

It follows from $\bar{x}^* \mathbf{X}^* \in \mathcal{D}(\mathbf{X})$ and the definition of $\bar{\mathbf{X}}^*$ that $g_{\text{rsbp}}(\bar{\mathbf{X}}^*) \le g_{\text{rsbp}}(\bar{x}^* \mathbf{X}^*)$. Therefore, we have $g_{\text{rsbp}}\left(\dfrac{\bar{\mathbf{X}}^*}{\bar{x}^*}\right) \le g_{\text{rsbp}}(\mathbf{X}^*)$. Since $\dfrac{\bar{\mathbf{X}}^*}{\bar{x}^*} \in \mathcal{D}_1(\mathbf{X})$ and the minimizer $\mathbf{X}^*$ of function $g_{\text{rsbp}}$ is unique, we obtain $X_i^* = \dfrac{\bar{X}_i^*}{\bar{x}^*} = \dfrac{\bar{X}_i^*}{\|\bar{X}_i^*\|_1}$ for all $i \in [m]$.

## C.3 Proof of Lemma 12

**Lemma 12.** *Let $(\mathbf{u}^k, \mathbf{v}^k)$ be the updates of* ROBUSTIBP *algorithm at the $k$-th step and $\mathbf{u}^* = (u_1^*, \dots, u_m^*)$ and $\mathbf{v}^* = (v_1^*, \dots, v_m^*)$ be the optimal solution of the dual problem (15). Let $\Delta u_i^k := u_i^k - u_i^*$ and $\Delta v_i^k := v_i^k - v_i^*$ for $i \in [m]$. When $m = 2$ and $k$ is even, we obtain that*

$$
\max\left\{\sum_{i=1}^m \|\Delta u_i^{k+1}\|_\infty, \sum_{i=1}^m \|\Delta v_i^{k+1}\|_\infty\right\} \le \tau\left(\frac{\tau}{\tau+\eta}\right)^{k/2} R_{\text{rsbp}},
$$

*where $R_{\text{rsbp}} := \sum_{i=1}^m \left(\max\left\{\log(n), \dfrac{\|C_i\|_\infty}{\eta} - \log(n)\right\} + \|\log(\mathbf{p}_i)\|_\infty + \dfrac{\eta+\tau}{\eta\tau}\|C_i\|_\infty\right)$.*

*Proof.* Firstly, we will show that when $k$ is even, $k \ge 1$ and $m = 2$,

$$
\max\left\{\sum_{i=1}^m \|\Delta u_i^{k+1}\|_\infty, \sum_{i=1}^m \|\Delta v_i^{k+1}\|_\infty\right\} \le \left(\frac{\tau}{\tau+\eta}\right)^{k/2} \sum_{i=1}^m \|v_i^*\|_\infty. \tag{45}
$$

Using the same arguments as deriving inequality (25), we have $\|\Delta u_i^{k+1}\|_\infty \le \frac{\tau}{\tau+\eta}\|\Delta v_i^k\|_\infty$. Since $\{v_i^*\}_{i=1}^m$ are the fixed points of the update in Algorithm 2,

$$
\frac{v_i^*}{\eta} = \left[\frac{v_i^*}{\eta} - \log(b_i^*)\right] - \sum_{t=1}^m \omega_t \left[\frac{v_t^*}{\eta} - \log(b_t^*)\right].
$$

Combining the above equality with the update of $v_i^k$ in Algorithm 2 and the fact $\sum_{t=1}^m \omega_t = 1$, we find that

$$\frac{\Delta v_i^k}{\eta} = \Delta V_i^{k-1} - \sum_{t=1}^m \omega_t \Delta V_t^{k-1} = \sum_{t \neq i} \omega_t (\Delta V_i^{k-1} - \Delta V_t^{k-1}).$$

where

$$\Delta V_i^k := \left( \frac{v_i^k}{\eta} - \log(b_i^k) \right) - \left( \frac{v_i^*}{\eta} - \log(b_i^*) \right) \quad \text{for all } i \in [m].$$

Notice that Lemma 4 can also be applied for this section, therefore, $\|\Delta V_i^k\|_\infty \leq \frac{\|\Delta u_i^k\|_\infty}{\eta}$ for all $i \in [m]$. Collecting these results, we have

$$\|\Delta v_i^k\|_\infty \leq \sum_{t \neq i} \omega_t (\|\Delta u_t^{k-1}\|_\infty + \|\Delta u_i^{k-1}\|_\infty).$$

When $m = 2$, these bounds show that

$$\sum_{i=1}^m \|\Delta v_i^k\|_\infty \leq \sum_{i=1}^m \|\Delta u_i^{k-1}\|_\infty.$$

Thus,

$$\sum_{i=1}^m \|\Delta u_i^{k+1}\|_\infty \leq \frac{\tau}{\tau+\eta} \sum_{i=1}^m \|\Delta u_i^{k-1}\|_\infty \leq \cdots \leq \left( \frac{\tau}{\tau+\eta} \right)^{k/2} \sum_{i=1}^m \|\Delta u_i^1\|_\infty$$

$$\leq \left( \frac{\tau}{\tau+\eta} \right)^{(k+2)/2} \sum_{i=1}^m \|\Delta v_i^0\|_\infty = \left( \frac{\tau}{\tau+\eta} \right)^{(k+2)/2} \sum_{i=1}^m \|v_i^*\|_\infty,$$

which leads to

$$\sum_{i=1}^m \|\Delta v_i^k\|_\infty \leq \sum_{i=1}^m \|\Delta u_i^{k-1}\|_\infty \leq \left( \frac{\tau}{\tau+\eta} \right)^{k/2} \sum_{i=1}^m \|v_i^*\|_\infty.$$

Recall that $v_i^{k+1} = v_i^k$ for all $i \in [m]$ when $k$ is even. Then, putting all of the above results, we obtain equation (45).

Next, we will prove that

$$\sum_{i=1}^m \|v_i^*\|_\infty \leq \tau R_{\mathrm{rsbp}}. \tag{46}$$

Since $\mathbf{u}^*$ is the fixed point of the update in Algorithm 2 , we have

$$\frac{(u_i^*)_j}{\tau} = \log((\mathbf{p}_i)_j) - \log \left( \sum_{l=1}^n \exp \left\{ \frac{(u_i^*)_j + (v_i^*)_l - (C_i)_{jl}}{\eta} \right\} \right),$$

which is equivalent to,

$$\left( \frac{1}{\tau} + \frac{1}{\eta} \right) (u_i^*)_j = \log((\mathbf{p}_i)_j) - \log \left( \sum_{l=1}^n \exp \left\{ \frac{(v_i^*)_l - (C_i)_{jl}}{\eta} \right\} \right).$$

Therefore,

$$\left( \frac{1}{\tau} + \frac{1}{\eta} \right) \sum_{i=1}^m \|u_i^*\|_\infty \leq \sum_{i=1}^m \left[ \|\log(\mathbf{p}_i)\|_\infty + \frac{\|v_i^*\|_\infty}{\eta} + \max \left\{ \log(n), \frac{\|C_i\|_\infty}{\eta} - \log(n) \right\} \right]. \tag{47}$$

For fixed $\mathbf{u}^*$, we have that

$$\mathbf{v}^* = \operatorname*{arg\,min}_{\mathbf{v} : \sum_{i=1}^m \omega_i v_i = \mathbf{0}_n} \bar{h}_{\mathrm{rsbp}}(\mathbf{u}^*, \mathbf{v}),$$

or equivalently,

$$\mathbf{v}^* = \arg\min \sum_{i=1}^m \omega_i \left[ \eta \sum_{j,l=1}^n \exp\left\{ \frac{(u_i^*)_j + (v_i)_l - (C_i)_{jl}}{\eta} \right\} + \tau \langle e^{-u_i^*/\tau}, \mathbf{p}_i \rangle \right] + \lambda^\top \sum_{i=1}^m \omega_i v_i,$$

where $\lambda \in \mathbb{R}^n$ is a vector of Lagrange multipliers. For each $i \in [m]$, taking derivatives of the RHS with respect to $v_i$,

$$\exp\left(\frac{v_i^*}{\eta}\right) \odot A_i + \lambda = \mathbf{0}_n$$

$$\Leftrightarrow \frac{v_i^*}{\eta} + \log(A_i) = \log(-\lambda). \tag{48}$$

where $A_i = \left( \sum_{j=1}^n \exp\left\{ \frac{(u_i^*)_j - (C_i)_{jl}}{\eta} \right\} \right)_{l=1}^n$.

Next, taking sum over $i$ and utilizing the fact that $\sum_{i=1}^m \omega_i v_i^* = 0$, we obtain $\sum_{i=1}^m \omega_i \log(A_i) = \log(-\lambda)$. Putting this result together with equation (48) leads to

$$\frac{v_i^*}{\eta} = \sum_{t=1}^m \omega_t \log(A_t) - \log(A_i) = \sum_{t=1}^m \omega_t \left[ \log(A_t) - \log(A_i) \right].$$

Since $m = 2$, the above equality indicates that $\frac{1}{\eta} \sum_{i=1}^m \|v_i^*\|_\infty \le \|\log(A_2) - \log(A_1)\|_\infty$. Furthermore, for all $l \in [n]$, applying part (a) of Lemma 3,

$$|\log(A_2)_l - \log(A_1)_l| = \left| \log\left( \frac{\sum_{j=1}^n \exp\left\{ \frac{(u_2^*)_j - (C_2)_{jl}}{\eta} \right\}}{\sum_{j=1}^n \exp\left\{ \frac{(u_1^*)_j - (C_1)_{jl}}{\eta} \right\}} \right) \right|$$

$$\le \frac{1}{\eta} \max_{1 \le j \le n} |(u_2^*)_j - (C_2)_{jl} - (u_1^*)_j + (C_1)_{jl}|$$

$$\le \frac{1}{\eta} \sum_{i=1}^m (\|u_i^*\|_\infty + \|C_i\|_\infty),$$

which implies that

$$\sum_{i=1}^m \|v_i^*\|_\infty \le \eta \|\log(A_2) - \log(A_1)\|_\infty \le \sum_{i=1}^m (\|u_i^*\|_\infty + \|C_i\|_\infty). \tag{49}$$

Combining equation (47) with equation (49), we obtain

$$\sum_{i=1}^m \|u_i^*\|_\infty \le \tau \sum_{i=1}^m \left[ \|\log(\mathbf{p}_i)\|_\infty + \frac{\|C_i\|_\infty}{\eta} + \max\left\{ \log(n), \frac{\|C_i\|_\infty}{\eta} - \log(n) \right\} \right].$$

Hence,

$$\sum_{i=1}^m \|v_i^*\|_\infty \le \sum_{i=1}^m \left[ \tau \|\log(\mathbf{p}_i)\|_\infty + \left(1 + \frac{\tau}{\eta}\right) \|C_i\|_\infty + \tau \max\left\{ \log(n), \frac{\|C_i\|_\infty}{\eta} - \log(n) \right\} \right] = \tau R_{\text{rsbp}}.$$

From equations (45) and (46), we get the conclusion of this lemma. □

### C.4 Proof of Theorem 2

Let $\mathbf{X}^k = (X_1^k, \ldots, X_m^k)$ be the normalized output at $k$-th iteration of Algorithm 2. We will firstly show that $\mathbf{X}^k$ is an $\varepsilon$-approximation of $\widehat{\mathbf{X}}$ for all even $k \ge 2 + 2\left(\frac{\tau}{\eta} + 1\right) \log\left(\frac{4R_{\text{rsbp}}\tau^2}{\eta^2}\right)$. By definition of $f_{\text{rsbp}}$ and $g_{\text{rsbp}}$,

$$f_{\text{rsbp}}(\mathbf{X}^k) - f_{\text{rsbp}}(\widehat{\mathbf{X}}) = g_{\text{rsbp}}(\mathbf{X}^k) - g_{\text{rsbp}}(\widehat{\mathbf{X}}) + \eta \sum_{i=1}^m \omega_i \left[ H(X_i^k) - H(\widehat{X}_i) \right]$$

$$\le g_{\text{rsbp}}(\mathbf{X}^k) - g_{\text{rsbp}}(\mathbf{X}^*) + \eta \sum_{i=1}^m \omega_i \left[ H(X_i^k) - H(\widehat{X}_i) \right]$$

The above two terms can be bounded as follows.

**Upper bound of $\sum_{i=1}^{m} \omega_i \left[ H(X_i^k) - H(\widehat{X}_i) \right]$.**

Applying the inequalities (27) for the entropy function, we have

$$\sum_{i=1}^{m} \omega_i \left[ H(X_i^k) - H(\widehat{X}_i) \right] \leq \sum_{i=1}^{m} \omega_i [2 \log(n) + 1 - 1] = 2 \log(n). \tag{50}$$

**Upper bound of $g_{\text{rsbp}}(\mathbf{X}^k) - g_{\text{rsbp}}(\mathbf{X}^*)$.**

Firstly, we consider the quantity $g_{\text{rsbp}}(\mathbf{X}^*)$.

$$
\begin{aligned}
g_{\text{rsbp}}(\mathbf{X}^*) &= g_{\text{rsbp}}\left(\frac{1}{\bar{x}^*}\bar{\mathbf{X}}^*\right) = \frac{1}{\bar{x}^*} g_{\text{rsbp}}(\bar{\mathbf{X}}^*) + \tau\left(1 - \frac{1}{\bar{x}^*}\right) \sum_{i=1}^{m} \omega_i \rho_i + (\tau + \eta) \log\left(\frac{1}{\bar{x}^*}\right) \\
&= \frac{1}{\bar{x}^*}\left[\tau \sum_{i=1}^{m} \omega_i \rho_i - (\tau + \eta)\bar{x}^*\right] + \tau\left(1 - \frac{1}{\bar{x}^*}\right) \sum_{i=1}^{m} \omega_i \rho_i - (\tau + \eta)\log(\bar{x}^*) \\
&= -(\eta + \tau) - (\eta + \tau)\log(\bar{x}^*) + \tau \sum_{i=1}^{m} \omega_i \rho_i.
\end{aligned}
$$

The second equality is due to equation (42) and the third one results from equation (43).

Based on Remark 3 and the fact that $\mathbf{X}^k = \dfrac{\bar{\mathbf{X}}^k}{\bar{x}^k}$, it is clear that $\mathbf{X}^k$ is the optimal solution of

$$\min_{X_1,\ldots,X_m \in \mathbb{R}_+^{n\times n}} g_{\text{rsbp}}^k(X_1,\ldots,X_m) := \sum_{i=1}^{m} \omega_i \left[ \langle C_i, X_i \rangle + \tau \mathbf{KL}(X_i \mathbf{1}_n \| \mathbf{p}_i^k) - \eta H(X_i) \right]$$

s.t. $X_i^\top \mathbf{1}_n = X_{i+1}^\top \mathbf{1}_n$ for all $i \in [m-1]$,

$\quad \|X_i\|_1 = 1$ for all $i \in [m]$.

Therefore, using the same arguments as for deriving for the quantity $g_{\text{rsbp}}(\mathbf{X}^*)$, we have

$$g_{\text{rsbp}}^k(\mathbf{X}^k) = -(\eta + \tau) - (\eta + \tau)\log(\bar{x}^k) + \tau \sum_{i=1}^{m} \omega_i \rho_i^k.$$

where $\rho_i^k := \|\mathbf{p}_i^k\|_1$. Denote $a_i^k = \bar{X}_i^k \mathbf{1}_n$ for all $i \in [m]$. Writing $g_{\text{rsbp}}(\mathbf{X}^k) - g_{\text{rsbp}}(\mathbf{X}^*) = \left[ g_{\text{rsbp}}(\mathbf{X}^k) - g_{\text{rsbp}}^k(\mathbf{X}^k) \right] + \left[ g_{\text{rsbp}}^k(\mathbf{X}^k) - g_{\text{rsbp}}(\mathbf{X}^*) \right]$, using the above equations of $g_{\text{rsbp}}^k(\mathbf{X}^k)$ and $g_{\text{rsbp}}(\mathbf{X}^*)$, and the definitions of $g_{\text{rsbp}}(\mathbf{X}^k)$ and $g_{\text{rsbp}}^k(\mathbf{X}^k)$, we get

$$g_{\text{rsbp}}(\mathbf{X}^k) - g_{\text{rsbp}}(\mathbf{X}^*) = (\eta + \tau)\log\left(\frac{\bar{x}^*}{\bar{x}^k}\right) + \frac{\tau}{\bar{x}^k} \sum_{i=1}^{m} \omega_i \sum_{j=1}^{n} (a_i^k)_j \log\left(\frac{(\mathbf{p}_i^k)_j}{(\mathbf{p}_i)_j}\right).$$

It follows from equation (44) that

$$\frac{1}{1 + \frac{3}{\eta}\Delta_1^k} \leq \frac{\bar{x}^*}{\bar{x}^k} \leq 1 + \frac{3}{\eta}\Delta_1^k,$$

or equivalently,

$$\left|\log\left(\frac{\bar{x}^*}{\bar{x}^k}\right)\right| \leq \log\left(1 + \frac{3}{\eta}\Delta_1^k\right) \leq \frac{3}{\eta}\Delta_1^k \leq \frac{3}{4}\frac{\eta}{\tau}.$$

Note that $(\mathbf{p}_i^k)_j = \exp\left(\frac{(u_i^k)_j}{\tau}\right)(a_i^k)_j$ and $(\mathbf{p}_i)_j = \exp\left(\frac{(u_i^*)_j}{\tau}\right)(a_i^*)_j$, the second term can be bounded as follows

$$
\begin{aligned}
\tau\left|\log\left(\frac{(\mathbf{p}_i^k)_j}{(\mathbf{p}_i)_j}\right)\right| &= \tau\left|\frac{1}{\tau}((u_i^k)_j - (u_i^*)_j) - \log\left(\frac{(a_i^*)_j}{(a_i^k)_j}\right)\right| \\
&\leq |(u_i^k)_j - (u_i^*)_j| + \tau\left|\log\left(\frac{(a_i^*)_j}{(a_i^k)_j}\right)\right| \\
&\leq \|u_i^k - u_i^*\|_\infty + \frac{\tau}{\eta}\left(\|u_i^k - u_i^*\|_\infty + \|v_i^k - v_i^*\|_\infty\right) \\
&\leq \left(\frac{2\tau + \eta}{\eta}\right)\Delta_i^k \\
&\leq \left(\frac{2\tau + \eta}{\eta}\right)\left(\frac{\eta^2}{4\tau}\right) \leq \eta\left(\frac{1}{2} + \frac{1}{12\log(n)}\right).
\end{aligned}
$$

Therefore,

$$
\left|\frac{\tau}{\bar{x}^k}\sum_{i=1}^m \omega_i \sum_{j=1}^n (a_i^k)_j \log\left(\frac{(\mathbf{p}_i^k)_j}{(\mathbf{p}_i)_j}\right)\right| \leq \eta\left(\frac{1}{2} + \frac{1}{12\log(n)}\right)\left[\frac{1}{\bar{x}^k}\sum_{i=1}^m \omega_i \sum_{j=1}^n (a_i^k)_j\right]
$$

$$
= \eta\left(\frac{1}{2} + \frac{1}{12\log(n)}\right).
$$

Combining the above bounds of the two terms leads to

$$
g_{\text{rsbp}}(\mathbf{X}^k) - g_{\text{rsbp}}(\mathbf{X}^*) \leq \eta\left(\frac{5}{4} + \frac{1}{3\log(n)}\right) \leq 2\eta. \tag{51}
$$

Finally, from equations (50) and (51), we obtain

$$
f_{\text{rsbp}}(\mathbf{X}^k) - f_{\text{rsbp}}(\widehat{\mathbf{X}}) \leq \eta\left(2 + 2\log(n)\right) \leq \eta U_{\text{rsbp}} = \varepsilon.
$$

**The complexity of Algorithm 2.** Next, we will derive the computational complexity of Algorithm 2. By definition of $U_{\text{rsbp}}$, the order of this quantity is $\mathcal{O}(\log(n))$. Rewriting the sufficient number of iterations for obtaining an $\varepsilon$-approximation as below

$$
2 + 2\left(\frac{\tau U_{\text{rsbp}}}{\varepsilon}\left[\log(4) + 2\log(\tau) + \log(\eta R_{\text{rsbp}}) + \log\left(\frac{U_{\text{rsbp}}}{\varepsilon}\right)\right]\right),
$$

which leads to

$$
k = \mathcal{O}\left(\frac{\tau\log(n)}{\varepsilon}\left[\log(\tau) + \log(\|C_1\|_\infty + \|C_2\|_\infty) + \log\left(\frac{\log(n)}{\varepsilon}\right)\right]\right).
$$

Multiplying with $\mathcal{O}(n^2)$ arithmetic operations per iteration, we get the final complexity.

## D Robust Unconstrained Optimal Transport: Useful Lemmas and Omitted Proofs

In this appendix, we continue to discuss in-depth the ROT problem, which is briefly introduced in Section 3.2. Similar to RSOT, solving directly the optimization problem (9) would be computationally expensive, particularly when $n$ is large. This encourages us to work on the entropic version of the problem (9), which admits the following form:

$$
\min_{X \in \mathbb{R}_+^{n \times n}; \|X\|_1 = 1} g_{\text{rot}}(X) := f_{\text{rot}}(X) - \eta H(X), \tag{52}
$$

for some $\eta > 0$. We name this objective *entropic ROT*. A general approach to solve this optimization problem is to derive its Fenchel duality, then performing alternating minimization on dual variables.

**Lemma 13.** *The dual form of the entropic ROT problem in equation* (52) *admits the following form*

$$
\min_{u,v \in \mathbb{R}^n} h(u,v) := \eta\log\|B(u,v)\|_1 + \tau\langle e^{-u/\tau}, \mathbf{a}\rangle + \tau\langle e^{-v/\tau}, \mathbf{b}\rangle. \tag{53}
$$

*Proof of Lemma 13.* The objective function (52) can be rewritten as follows

$$\min_{\substack{X\in\mathbb{R}^{n\times n},\|X\|_1=1;\\ X\mathbf{1}_n=y,X^\top\mathbf{1}_n=z}} \langle C,X\rangle - \eta H(X) + \tau\mathbf{KL}(y\|\mathbf{a}) + \tau\mathbf{KL}(z\|\mathbf{b}).$$

By introducing the dual variables $u\in\mathbb{R}^n$ and $v\in\mathbb{R}^n$, the Lagrangian duality of the above objective function takes the following form

$$\max_{u,v\in\mathbb{R}^n}\min_{\substack{X\in\mathbb{R}^{n\times n},\|X\|_1=1;\\ y,z\in\mathbb{R}^n}} \langle C,X\rangle - \eta H(X) + \tau\mathbf{KL}(y\|\mathbf{a}) + \tau\mathbf{KL}(z\|\mathbf{b})$$

$$- u^\top(X\mathbf{1}_n - y) - v^\top(X^\top\mathbf{1}_n - z).$$

We can check that

$$\min_{y\in\mathbb{R}^n}\tau\mathbf{KL}(y\|\mathbf{a}) + u^\top y = -\tau\left\langle e^{-u/\tau},\mathbf{a}\right\rangle + \mathbf{a}^\top\mathbf{1}_n,$$

$$\min_{z\in\mathbb{R}^n}\tau\mathbf{KL}(z\|\mathbf{b}) + v^\top z = -\tau\left\langle e^{-v/\tau},\mathbf{b}\right\rangle + \mathbf{b}^\top\mathbf{1}_n.$$

Furthermore, for the minimization problem

$$\min_{X\in\mathbb{R}^{n\times n},\|X\|_1=1}\langle C,X\rangle - u^\top X\mathbf{1}_n - v^\top X^\top\mathbf{1}_n - \eta H(X),$$

the objective function is strongly convex. Therefore, it has an unique global minima. Direct calculations demonstrate that the optimal solution of that objective function takes the following form

$$\bar{X} = \frac{B(u,v)}{\|B(u,v)\|_1},\quad\text{where } B(u,v)_{ij} := \exp\left(\frac{u_i + v_j - C_{ij}}{\eta}\right).$$

Based on the above argument, we can check that

$$\min_{X\in\mathbb{R}^{n\times n},\|X\|_1=1}\langle C,X\rangle - u^\top X\mathbf{1}_n - v^\top X^\top\mathbf{1}_n - \eta H(X) = -\eta\log\|B(u,v)\|_1.$$

Combining all the above results, we obtain the conclusion. □

Strong duality holds for the problem (52), and its optimal solution can be obtained via the optimal solution of the problem (53), i.e., $X^* = B(u^*,v^*)$. To solve the latter, we can set the partial derivatives of its objective with respect to $u$ and $v$ to zero, resulting in

$$\frac{B(u,v)\mathbf{1}_n}{\|B(u,v)\|_1} = e^{-u/\tau}\odot\mathbf{a},\quad \frac{B(u,v)^T\mathbf{1}_n}{\|B(u,v)\|_1} = e^{-v/\tau}\odot\mathbf{b},$$

where $\odot$ denoting element-wise multiplication. It is challenging to derive closed-form solutions for each coordinate $u_i$ and $v_j$ for $i,j\in[n]$ from this system of equations. Consequently, we do not get a direct update for $u_i$ and $v_j$ in the coordinate descent algorithm. Therefore, developing directly Sinkhorn algorithm for solving entropic ROT like the RSOT case could be non-trivial.

---

**Algorithm 3:** ROBUST-SINKHORN

> **Input:** $C,\mathbf{a},\mathbf{b},\tau,\eta,k_{iter}$
> **Output:** $X$
> **Initialization:** $u^0 = v^0 = \mathbf{0}_n, k = 0$
> **while** $k < k_{iter}$ **do**
> $\quad a^k = B(u^k,v^k)\mathbf{1}_n$
> $\quad b^k = (B(u^k,v^k))^\top\mathbf{1}_n$
> $\quad$**if** $k$ is even **then**
> $\quad\quad u^{k+1} \leftarrow \frac{\eta\tau}{\eta+\tau}\left[\frac{u^k}{\eta} + \log(\mathbf{a}) - \log(a^k)\right]$
> $\quad\quad v^{k+1} \leftarrow v^k$
> $\quad$**else**
> $\quad\quad u^{k+1} \leftarrow u^k$
> $\quad\quad v^{k+1} \leftarrow \frac{\eta\tau}{\eta+\tau}\left[\frac{v^k}{\eta} + \log(\mathbf{b}) - \log(b^k)\right]$
> $\quad$**end if**
> $\quad k = k+1$
> **end while**
> **return** $X^k = B(u^k,v^k)/\|B(u^k,v^k)\|_1$

It is worth noting that the required iteration to reach an $\varepsilon$-approximation of UOT is *not* identical to that of ROT, or in a broader sense, it is not trivial to derive one from the other. Hence, in the following theorem, we present one of our main results regarding the complexity of ROBUST-SINKHORN algorithm in reaching an $\varepsilon$-approximation of ROT.

**Theorem 3.** *For* $\eta = \varepsilon U_{\mathrm{rot}}^{-1}$ *where*

$$U_{\mathrm{rot}} = \max\left\{ \frac{3(\tau+2)}{4(\tau+1)} + 2\log(n), 2\varepsilon, \frac{5\varepsilon\log(n)}{\tau} \right\},$$

*Algorithm 3 returns an $\varepsilon$-approximation of the optimal solution $\widehat{X}_{\mathrm{rot}}$ for the problem* (9) *in time*

$$\mathcal{O}\left( \frac{\tau n^2}{\varepsilon} \log(n) \left[ \log\left( \frac{\tau \|C\|_\infty}{\varepsilon} \right) + \log(\log(n)) \right] \right).$$

The result of Theorem 3 shows that the complexity of ROBUST-SINKHORN algorithm for computing ROT is at the order of $\widetilde{\mathcal{O}}(\frac{n^2}{\varepsilon})$, which is near-optimal and at the same order as that of the Sinkhorn algorithm for solving UOT [27]. Furthermore, similar to the RSOT case, the complexity of ROBUST-SINKHORN algorithm is also better than that of the Sinkhorn algorithm for computing the standard optimal transport problem.

### D.1 Useful Lemmas

Prior to presenting the proof of Theorem 3, in this section, we provide the proof of Lemma 1 as well as several useful properties of ROT and UOT that will be used later on.

*Proof of Lemma 1.* Using the equation for $g_{\mathrm{rot}}(tX)$ in (54), we have that

$$g_{\mathrm{rot}}(X_{\mathrm{uot}}^*) = g_{\mathrm{rot}}\left( (x_{\mathrm{uot}}^*)\left( \frac{X_{\mathrm{uot}}^*}{x_{\mathrm{uot}}^*} \right) \right)$$

$$= x_{\mathrm{uot}}^* g_{\mathrm{rot}}\left( \frac{X_{\mathrm{uot}}^*}{x_{\mathrm{uot}}^*} \right) + \tau\left( 1 - x_{\mathrm{uot}}^* \right)(\alpha + \beta) + (2\tau + \eta)x_{\mathrm{uot}}^* \log(x_{\mathrm{uot}}^*)$$

$$g_{\mathrm{rot}}(x_{\mathrm{uot}}^* X_{\mathrm{rot}}^*) = x_{\mathrm{uot}}^* g_{\mathrm{rot}}(X_{\mathrm{rot}}^*) + \tau\left( 1 - x_{\mathrm{uot}}^* \right)(\alpha + \beta) + (2\tau + \eta)x_{\mathrm{uot}}^* \log(x_{\mathrm{uot}}^*).$$

In terms of the left-handed sides, $g_{\mathrm{rot}}(X_{\mathrm{uot}}^*) \le g_{\mathrm{rot}}(x_{\mathrm{uot}}^* X_{\mathrm{rot}}^*)$ by definition of $X_{\mathrm{uot}}^*$. On the right-handed sides, the second and third are the same. Thus, from the above two equations we obtain

$$g_{\mathrm{rot}}\left( \frac{X_{\mathrm{uot}}^*}{x_{\mathrm{uot}}^*} \right) \le g_{\mathrm{rot}}(X_{\mathrm{rot}}^*).$$

As the optimization problem of ROT has an unique solution, $X_{\mathrm{rot}}^* = \frac{X_{\mathrm{uot}}^*}{x_{\mathrm{uot}}^*}$. $\qquad\square$

**Lemma 14 (Convergence rate for $u_{\mathrm{uot}}^k$ and $v_{\mathrm{uot}}^k$).** *For any* $k \ge 1 + \left( \frac{\tau}{\eta} + 1 \right)\log\left( \frac{8R\tau(\tau+1)}{\eta^2} \right)$, *the updates* $(u_{\mathrm{uot}}^k, v_{\mathrm{uot}}^k)$ *from Algorithm 3 can be bounded as follows,*

$$\Delta_{\mathrm{uot}}^k := \max\{\|u_{\mathrm{uot}}^k - u_{\mathrm{uot}}^*\|_\infty, \|v_{\mathrm{uot}}^k - v_{\mathrm{uot}}^*\|_\infty\} \le \frac{\eta^2}{8(\tau+1)}.$$

*Proof of Lemma 14.* This lemma is the combination of Theorem 1 and Lemma 5 part (a) in [27]. $\quad\square$

**Lemma 15.** *Let* $x_{\mathrm{uot}}^* := \|X_{\mathrm{uot}}^*\|_1$, *then the quantity* $g_{\mathrm{rot}}(X_{\mathrm{uot}}^*)$ *is presented as*
$$g_{\mathrm{rot}}(X_{\mathrm{uot}}^*) + 2(\tau + \eta)x_{\mathrm{uot}}^* = \tau(\alpha + \beta).$$

*Proof of Lemma 15.* The proof of this lemma can be found in Lemma 4 of [27]. $\qquad\square$

**Lemma 16.** *We have the following relation between the optimal value of entropic ROT and other parameters*
$$g_{\mathrm{rot}}(X_{\mathrm{rot}}^*) = \tau(\alpha + \beta - 2) - \eta - (2\tau + \eta)\log(x_{\mathrm{uot}}^*).$$
*Furthermore, let* $g_{\mathrm{rot}}^k(X) := \langle C, X \rangle - \eta H(X) + \tau \mathbf{KL}\left( X\mathbf{1}_n \| \mathbf{a} \right) + \tau \mathbf{KL}\left( X^\top \mathbf{1}_n \| \mathbf{b}_{\mathrm{uot}}^k \right)$, *with* $\mathbf{b}_{\mathrm{uot}}^k := \exp\left( \frac{v_{\mathrm{uot}}^k}{\tau} \right) \odot \left[ \left( X_{\mathrm{uot}}^k \right)^T \mathbf{1}_n \right]$ *and* $\beta_{\mathrm{uot}}^k := \|\mathbf{b}_{\mathrm{uot}}^k\|_1$. *If $k$ is odd, we have that*
$$g_{\mathrm{rot}}^k(X_{\mathrm{rot}}^k) = \tau(\alpha + \beta_{\mathrm{uot}}^k - 2) - \eta - (2\tau + \eta)\log(x_{\mathrm{uot}}^k).$$

*Proof of Lemma 16.* First, we recall from Lemma 4 [27] that, for $t \in \mathbb{R}_+$ and $X \in \mathbb{R}_+^{n \times n}$,

$$g_{\text{rot}}(tX) = tg_{\text{rot}}(X) + \tau(1-t)(\alpha+\beta) + (2\tau+\eta)xt\log(t). \tag{54}$$

Applying this equation with $X = X_{\text{rot}}^*$ and $t = x_{\text{uot}}^*$, we obtain

$$g_{\text{rot}}(X_{\text{uot}}^*) = x_{\text{uot}}^* g_{\text{rot}}(X_{\text{rot}}^*) + \tau(1 - x_{\text{uot}}^*)(\alpha+\beta) + (2\tau+\eta)x_{\text{uot}}^* \log(x_{\text{uot}}^*).$$

Combining with the fact that $g_{\text{rot}}(X_{\text{uot}}^*) + (2\tau+\eta)x_{\text{uot}}^* = \tau(\alpha+\beta)$ stated in Lemma 15, we get the final equality for $g_{\text{rot}}(X_{\text{rot}}^*)$. Finally, note that $X_{\text{uot}}^k = \arg\min g_{\text{rot}}^k(X)$, the same argument thus can be applied, and we obtain the equality for $g_{\text{rot}}^k(X_{\text{rot}}^k)$. □

## D.2 Proof of Theorem 3

First, we will show that $X_{\text{rot}}^k$ is an $\varepsilon$-approximation of $\widehat{X}_{\text{rot}}$ for all $k \geq 1 + \left(\frac{\tau}{\eta}+1\right)\log\left(\frac{8R\tau(\tau+1)}{\eta^2}\right)$. By definitions of $f_{\text{rot}}$ and $g_{\text{rot}}$, we have

$$\begin{aligned}
f_{\text{rot}}(X_{\text{rot}}^k) - f_{\text{rot}}(\widehat{X}_{\text{rot}}) &= g_{\text{rot}}(X_{\text{rot}}^k) + \eta H(X_{\text{rot}}^k) - g_{\text{rot}}(\widehat{X}_{\text{rot}}) - \eta H(\widehat{X}_{\text{rot}}) \\
&\leq \left[g_{\text{rot}}(X_{\text{rot}}^k) - g_{\text{rot}}(X_{\text{rot}}^*)\right] + \eta\left[H(X_{\text{rot}}^k) - H(\widehat{X}_{\text{rot}})\right],
\end{aligned} \tag{55}$$

**Upper bound of $H(X_{\text{rot}}^k) - H(\widehat{X}_{\text{rot}})$.** Since $\|X_{\text{rot}}^k\|_1 = \|\widehat{X}_{\text{rot}}\|_1 = 1$, applying the lower and upper bounds for the entropy in (27), we have

$$H(X_{\text{rot}}^k) - H(\widehat{X}_{\text{rot}}) \leq 2\log(n). \tag{56}$$

**Upper bound of $g_{\text{rot}}(X_{\text{rot}}^k) - g_{\text{rot}}(X_{\text{rot}}^*)$.** WLOG, we consider the case where $k$ is odd. By Lemma 16,

$$g_{\text{rot}}(X_{\text{rot}}^*) = \tau(\alpha+\beta-2) - \eta - (2\tau+\eta)\log(x_{\text{uot}}^*) \tag{57}$$

$$g_{\text{rot}}^k(X_{\text{rot}}^k) = \tau(\alpha+\beta_{\text{uot}}^k-2) - \eta - (2\tau+\eta)\log(x_{\text{uot}}^k). \tag{58}$$

Writing $g_{\text{rot}}(X_{\text{rot}}^k) - g_{\text{rot}}(X_{\text{rot}}^*) = \left[g_{\text{rot}}(X_{\text{rot}}^k) - g_{\text{rot}}^k(X_{\text{rot}}^k)\right] + \left[g_{\text{rot}}^k(X_{\text{rot}}^k) - g_{\text{rot}}(X_{\text{rot}}^*)\right]$. For the first term, we have

$$g_{\text{rot}}(X_{\text{rot}}^k) = \langle C, X_{\text{rot}}^k \rangle + \tau\mathbf{KL}(X_{\text{rot}}^k \mathbf{1}_n \| \mathbf{a}) + \tau\mathbf{KL}((X_{\text{rot}}^k)^T \mathbf{1}_n \| \mathbf{b}) - \eta H(X_{\text{rot}}^k)$$

$$g_{\text{rot}}^k(X_{\text{rot}}^k) = \langle C, X_{\text{rot}}^k \rangle + \tau\mathbf{KL}(X_{\text{rot}}^k \mathbf{1}_n \| \mathbf{a}) + \tau\mathbf{KL}((X_{\text{rot}}^k)^T \mathbf{1}_n \| \mathbf{b}_{\text{uot}}^k) - \eta H(X_{\text{rot}}^k).$$

Then, we find that

$$g_{\text{rot}}(X_{\text{rot}}^k) - g_{\text{rot}}^k(X_{\text{rot}}^k) = \tau\left[\mathbf{KL}(\underbrace{(X_{\text{rot}}^k)^T \mathbf{1}_n}_{:=b_{rot}^k} \| \mathbf{b}) - \mathbf{KL}((X_{\text{rot}}^k)^T \mathbf{1}_n \| \mathbf{b}_{\text{uot}}^k)\right]$$

$$= \tau\left[\sum_{j=1}^n (b_{\text{rot}}^k)_j \log\left(\frac{(\mathbf{b}_{\text{uot}}^k)_j}{\mathbf{b}_j}\right) + (\beta - \beta_{\text{uot}}^k)\right]. \tag{59}$$

Combining equations (57), (58) and (59), we obtain

$$g_{\text{rot}}(X_{\text{rot}}^k) - g_{\text{rot}}(X_{\text{rot}}^*) = (2\tau+\eta)\log\left(\frac{x_{\text{uot}}^*}{x_{\text{uot}}^k}\right) + \tau\left[\sum_{j=1}^n (b_{\text{rot}}^k)_j \log\left(\frac{(\mathbf{b}_{\text{uot}}^k)_j}{\mathbf{b}_j}\right)\right]. \tag{60}$$

Using the following result

$$\max\left\{\frac{x_{\text{uot}}^*}{x_{\text{uot}}^k}, \frac{x_{\text{uot}}^k}{x_{\text{uot}}^*}\right\} \leq \left(\frac{\|u_{\text{uot}}^k - u_{\text{uot}}^*\|_\infty}{\eta}\right)\left(\frac{\|v_{\text{uot}}^k - v_{\text{uot}}^*\|_\infty}{\eta}\right)$$

in the proof of Lemma 5 part (b) in [27], the first term is bounded by $\frac{2(2\tau+\eta)}{\eta}\Delta_{\text{uot}}^k$.

Let $b_{\text{uot}}^k := (X_{\text{uot}}^k)^\top \mathbf{1}_n$ and $b_{\text{uot}}^* := (X_{\text{uot}}^*)^\top \mathbf{1}_n$. Note that $(\mathbf{b}_{\text{uot}}^k)_j = \exp\left(\frac{(v_{\text{uot}}^k)_j}{\eta}\right) (b_{\text{uot}}^k)_j$ and $\mathbf{b}_j = \exp\left(\frac{(v_{\text{uot}}^*)_j}{\eta}\right) (b_{\text{uot}}^*)_j$. Applying part (b) of Lemma 4, we find that

$$\left|\log\left(\frac{(\mathbf{b}_{\text{uot}}^k)_j}{\mathbf{b}_j}\right)\right| = \left|-\log\left(\frac{(b_{\text{uot}}^*)_j}{(b_{\text{uot}}^k)_j}\right) + \frac{1}{\tau}[(v_{\text{uot}}^k)_j - (v_{\text{uot}}^*)_j]\right|$$

$$\leq \frac{2}{\eta}\Delta_{\text{uot}}^k + \frac{1}{\tau}\Delta_{\text{uot}}^k = \left(\frac{2}{\eta} + \frac{1}{\tau}\right)\Delta_{\text{uot}}^k,$$

which leads to

$$\left|\sum_{j=1}^n (b_{\text{rot}}^k)_j \log\left(\frac{(\mathbf{b}_{\text{uot}}^k)_j}{\mathbf{b}_j}\right)\right| \leq \underbrace{\left(\sum_{j=1}^n (b_{\text{rot}}^k)_j\right)}_{=\|X_{\text{rot}}^k\|_1 = 1} \max_{1 \leq j \leq n}\left|\log\left(\frac{(\mathbf{b}_{\text{uot}}^k)_j}{\mathbf{b}_j}\right)\right| \leq \left(\frac{2}{\eta} + \frac{1}{\tau}\right)\Delta_{\text{uot}}^k.$$

Collecting all the inequalities for each term in (60), we obtain

$$g_{\text{rot}}(X_{\text{rot}}^k) - g_{\text{rot}}(X_{\text{rot}}^*) \leq \frac{3}{\eta}(2\tau + \eta)\Delta_{\text{uot}}^k.$$

Furthermore, from Lemma 14, we get $\Delta_{\text{uot}}^k \leq \frac{\eta^2}{8(\tau+1)}$. Then,

$$g_{\text{rot}}(X_{\text{rot}}^k) - g_{\text{rot}}(X_{\text{rot}}^*) \leq \frac{3\eta(2\tau + 4)}{8(\tau + 1)} = \eta\left[\frac{3(\tau + 2)}{4(\tau + 1)}\right]. \tag{61}$$

Putting the results from equations (56) and (61) leads to

$$f_{\text{rot}}(X_{\text{rot}}^k) - f_{\text{rot}}(\widehat{X}_{\text{rot}}) \leq \eta\left[\frac{3(\tau + 2)}{4(\tau + 1)} + 2\log(n)\right] \leq \eta U_{\text{rot}} = \varepsilon.$$

**The complexity of Algorithm 3.** Next, we will compute the complexity of Algorithm 3 under the assumption that $R = \mathcal{O}\left(\frac{1}{\eta}\|C\|_\infty\right)$. The sufficient number of iterates to obtain an $\varepsilon$-approximation of $\widehat{X}_{\text{rot}}$ can be rewritten as

$$\left(\frac{\tau U_{\text{rot}}}{\varepsilon} + 1\right)\left[\log(\eta R) + \log(\tau(\tau + 1)) + \log\left(\frac{U_{\text{rot}}}{\varepsilon}\right)\right].$$

By the definition of $U_{\text{rot}}$, we find that $U_{\text{rot}} = \mathcal{O}(\log(n))$. Overall,

$$k = \mathcal{O}\left(\frac{\tau \log(n)}{\varepsilon}\left[\log(\|C\|_\infty) + \log(\tau) + \log(\log(n)) + \log\left(\frac{1}{\varepsilon}\right)\right]\right).$$

By multiplying the above bound of $k$ with $\mathcal{O}(n^2)$ arithmetic operations per iteration, we get the desired complexity.

## E   Details on Low-Rank Approximation

Though previous complexity analyses of standard Sinkhorn algorithms are favorable in terms of $\varepsilon$, they exhibit quadratic growth with regards to $n$ in both time and space complexity. Therefore, they are unscalable when $n$ is huge in practice. As the robust Sinkhorn algorithms mainly involve matrix-vector multiplications, the computational cost can be reduced by utilizing special structures of some factors, such as the Gaussian kernel matrix $K := \exp\left(\frac{-C}{\eta}\right)$. By approximating $K$ with a low-rank matrix, we show that the proposed robust Sinkhorn algorithms can be sped up considerably with a high probability while still reaching a nearly-optimal solution. A similar approach based on Nyström method had been studied in the optimal transport problem [2]. In this section, building on these analyses, we provide some novel results for scaling up the robust algorithms developed in previous sections. The idea of Nyström approximation is that given a kernel matrix $K$ where $K_{ij} = k(x_i, x_j)$ are constructed from $n$ data points $\mathcal{X} = \{x_1, \ldots, x_n\} \subset \mathbb{R}^d$, with $k : \mathcal{X} \times \mathcal{X} \to \mathbb{R}$ being a kernel function, we select $r$ points $\{x_{p_1}, \ldots, x_{p_r}\} \subset \mathcal{X}$ to construct two matrices: $V \in \mathbb{R}^{n \times r}$

where $V_{ij} = k(x_i, x_{p_j})$ and $A \in \mathbb{R}^{r \times r}$ where $A_{ij} = k(x_{p_i}, x_{p_j})$. An approximation of $K$ is given by $\widetilde{K} = VA^{-1}V^\top$, which is the kernel matrix of the dataset after being projected onto the space of the chosen subset. Whether $\widetilde{K}$ is a good approximation of $K$ depends on $r$ and the art of selecting $r$ data points. In Algorithm 5, we make use of the adaptive procedure namely ADAPTIVENYSTRÖM from [2] to obtain $\widetilde{K}$, which subsequently is used in the ROBUST-SEMISINKHORN (or ROBUST-SINKHORN) algorithm. We show in Theorem 4 that, with some specific choices of parameters, we could obtain matrix $\widetilde{K}$ such that an $\varepsilon$-approximation is achievable in almost linear time.

---

**Algorithm 4:** ADAPTIVENYSTRÖM

> **Input:** $\mathcal{X} = \{x_1, x_2, ..., x_n\}, \eta > 0, \tau > 0$
> **Output:** $\widetilde{K} \in \mathbb{R}^{n \times n}, r \in \mathbb{N}$
> err $\leftarrow +\infty, r \leftarrow 1$
> **while** err $> \tau$ **do**
>     $r \rightarrow 2r$
>     $\widetilde{K} \leftarrow$ NYSTRÖM $(\mathcal{X}, \eta, r)$
>     err $\leftarrow 1 - \min_{i \in [n]} \widetilde{K}_{ii}$
> **end while**
> **return** $(\widetilde{K}, \text{rank}(\widetilde{K}))$

---

**Algorithm 5:** ROBUST-NYSSINK

> **Input:** $\mathcal{X} = \{x_1, \ldots, x_n : \|x_i\|_2 \leq R\}, \mathbf{a}, \mathbf{b}, \eta, \tau, \varepsilon, k$
> $Z \leftarrow 1 + 2(\tau + \eta)$ or $2 + \eta + \frac{2\tau}{\eta}$     (RSOT or ROT)
> $\varepsilon' \leftarrow \min(1, \frac{\varepsilon}{Z})$
> $(\widetilde{K}, r) \leftarrow$ ADAPTIVENYSTRÖM$(\mathcal{X}, \eta, \frac{\varepsilon'}{2}e^{-4\eta^{-1}R^2})$
> $\widetilde{C} \leftarrow -\eta \log \widetilde{K}$
> $\widehat{X} \leftarrow$ ROBUST-(SEMI)SINKHORN$(\widetilde{C}, \mathbf{a}, \mathbf{b}, \eta, \tau, k)$
> **Output:** $\widehat{X}$

---

**Theorem 4.** *We denote by $f_C$ the objective function of RSOT (5) and ROT (9) problems regarding some cost matrix $C$. Furthermore, let $\widehat{X}_C$ be the corresponding optimal solution, and $X_{\widetilde{C}}^k$ be the output of Algorithm 5 for $k$ Sinkhorn iterations. Then, for $0 < \varepsilon < 1$, Algorithm 5 achieves an $\varepsilon$-approximation $X_{\widetilde{C}}^k$ of $\widehat{X}_C$, i.e., $f_C(X_{\widetilde{C}}^k) - f_C(\widehat{X}_C) \leq \varepsilon$, in $\widetilde{O}(nr^2 + \frac{nr}{\varepsilon})$ calculations.*

Theorem 4 indicates that using Nyström approximation reduces the original complexity of the robust algorithms by a factor $n/r^2$. As a side note, [2] provides a probabilistic bound on $r$ (for more detail see Appendix E). Furthermore, in terms of space complexity, Algorithm 5 uses $O(n(r + d))$ space, where $d$ is the dimension of data constructing the cost matrix $C$.

Subsequently, we derive the complexity of Sinkhorn-based algorithms using Nyström approximation in both RSOT and ROT problems. As the proof for both problems share many similarities, we abuse the notation by using the same notations for both cases. In particular, we denote $f_C$ to be the objective functions of RSOT and ROT as in (5) and (9), respectively, with $C$ is the cost matrix. Similarly we denote $g_C$ to be the objective functions with entropic regularization of RSOT and ROT as in (6) and (52), respectively. We recall and define some other quantities as follow:

$$\widehat{X}_C = \arg\min f_C(X),$$
$$X_C^* = \arg\min g_C(X),$$
$$X_{\widetilde{C}}^* = \arg\min g_{\widetilde{C}}(X);$$

where $\widetilde{C}$ is the matrix produced by the Nyström method. For other notations, we remove the index rsot and rot in quantities i.e. $u_{\text{rsot}}^k$ in order to keep them simple.

*Proof of Theorem 4.* Assume that we have following bounds

$$\|X_{\widetilde{C}}^k\|_1 \le S_x, \tag{62}$$

$$g_{\widetilde{C}}(X_{\widetilde{C}}^k) - g_{\widetilde{C}}(X_{\widetilde{C}}^*) \le \eta S_g, \tag{63}$$

$$H(X_{\widetilde{C}}^k) - H(\widehat{X}_C) \le S_H, H(X_{\widetilde{C}}^k) - H(\widehat{X}_{\widetilde{C}}) \le S_H, \tag{64}$$

$$\left| g_{\widetilde{C}}(X_{\widetilde{C}}^*) - g_C(X_C^*) \right| \le S_C \|C - \widetilde{C}\|_\infty, \tag{65}$$

where $S_x, S_g, S_H, S_C$ are constants that may contain $\alpha, \beta, \eta, \tau$ or $C$, varying between cases.

By definitions of $\widehat{X}_C$ and $X_{\widetilde{C}}^*$, we have

$$f_C(\widehat{X}_C) = g_C(\widehat{X}_C) + \eta H(\widehat{X}_C) \ge g_C(X_C^*) + \eta H(\widehat{X}_C),$$

and

$$f_C(X_{\widetilde{C}}^k) \le \left| f_C(X_{\widetilde{C}}^k) - f_{\widetilde{C}}(X_{\widetilde{C}}^k) \right| + f_{\widetilde{C}}(X_{\widetilde{C}}^k)$$
$$= \left| \langle C - \widetilde{C}, X_{\widetilde{C}}^k \rangle \right| + \eta H(X_{\widetilde{C}}^k) + g_{\widetilde{C}}(X_{\widetilde{C}}^k).$$

For the first term, using Holder's inequality and (62) we get $\left| \langle C - \widetilde{C}, X_{\widetilde{C}}^k \rangle \right| \le \|C - \widetilde{C}\|_\infty \|X_{\widetilde{C}}^k\|_1 \le \|C - \widetilde{C}\|_\infty S_x$. Combining with (63), we have $f_C(X_{\widetilde{C}}^k)$ is bounded by

$$\|C - \widetilde{C}\|_\infty S_x + \eta H(X_{\widetilde{C}}^k) + \eta S_g + g_{\widetilde{C}}(X_{\widetilde{C}}^*).$$

We thus obtain

$$f_C(X_{\widetilde{C}}^k) - f_C(\widehat{X}_C) \le \|C - \widetilde{C}\|_\infty S_x + \eta \underbrace{\left( H(X_{\widetilde{C}}^k) - H(\widehat{X}_C) \right)}_{\le S_H} + \eta S_g + \underbrace{\left( g_{\widetilde{C}}(X_{\widetilde{C}}^*) - g_C(X_C^*) \right)}_{\le S_C \|C - \widetilde{C}\|_\infty}$$

$$\le \|C - \widetilde{C}\|_\infty S_x + \eta S_H + \eta S_g + S_C \|C - \widetilde{C}\|_\infty$$

$$= \underbrace{(\eta S_H + \eta S_g)}_{\le \varepsilon'} + (S_x + S_C) \underbrace{\|C - \widetilde{C}\|_\infty}_{= \eta \|\log(K) - \log(\widetilde{K})\|_\infty}$$

$$\le \varepsilon' + (S_x + S_C)\eta \|\log(K) - \log(\widetilde{K})\|_\infty$$

$$\le \varepsilon' + (S_x + S_C)\eta \varepsilon'$$

$$= \varepsilon'(1 + \eta S_x + \eta S_C)$$

$$= \varepsilon,$$

where the third inequality $\eta S_H + \eta S_g \le \varepsilon'$ comes from using ROBUST-(SEMI)SINKHORN algorithm on the approximated cost $\widetilde{C}$ with the error $\varepsilon'$, and the fourth inequality $\|\log(K) - \log(\widetilde{K})\|_\infty \le \varepsilon'$ is a result of the ADAPTIVENYSTRÖM procedure (see Lemma L, [2]).

**Time complexity.** Since $S_x = \widetilde{O}(1)$ and $S_C = \widetilde{O}(1)$, we get $\widetilde{O}(\frac{1}{\varepsilon'}) = \widetilde{O}(\frac{1 + \eta S_x + \eta S_C}{\varepsilon}) = \widetilde{O}(\frac{1}{\varepsilon})$. The ADAPTIVENYSTRÖM routine takes $O(nr^2)$ time, while the ROBUST-(SEMI)SINKHORN routine runs through $\widetilde{O}(\frac{1}{\varepsilon'})$ iterations. Each iteration then takes $O(n + nr) = O(nr)$ time, in which $O(n)$ for vector additions, and $O(nr)$ for low-rank matrix vector multiplications. In total, the time complexity is $\widetilde{O}(nr^2 + \frac{nr}{\varepsilon'})$.

**Space complexity.** As we only need to save the implicit form of $\widetilde{K}$ via two matrices $KS \in \mathbb{R}^{n \times r}$ and $(S^T K S)^+ \in \mathbb{R}^{r \times r}$ (where $S$ is the column selection matrix, i.e. $KS$ comprises $r$ columns of $K$), $n$ data points of dimension $d$ as well as other $n$-dimensional vectors, the total space required is $O(nr + r^2 + nd) = O(nr + nd)$. $\qquad \square$

Now we take a look at the cases of RSOT and ROT. In particular, we derive the upper bounds for $S_x$, $S_g$, $S_H$ and $S_C$.

## E.1 Robust Unbalanced Optimal Transport

In this case, the constants are

$$S_x = 1, S_g = \frac{3(\tau + 2)}{4(\tau + 1)}, S_H = 2\log(n), S_C = \frac{2\tau + \eta}{\eta^2}.$$

*Proofs of Inequalities.* The inequalities for $S_x, S_g$ and $S_H$ comes from the fact that the $X_{\widetilde{C}}^k$ was normalized, inequality (61) and inequality (56) respectively in the section D of ROT's proofs. Regarding to $S_C$, we have

$$g_C(X_C^*) = \tau(\alpha + \beta - 2) - \eta - (2\tau + \eta)\log(x_C^*),$$
$$g_{\widetilde{C}}(X_{\widetilde{C}}^*) = \tau(\alpha + \beta - 2) - \eta - (2\tau + \eta)\log(x_{\widetilde{C}}^*).$$

Consequently, $\left|g_{\widetilde{C}}(X_{\widetilde{C}}^*) - g_C(X_C^*)\right| = (2\tau + \eta)\left|\log\left(\frac{x_{\widetilde{C}}^*}{x_C^*}\right)\right|.$

**Upper bound for** $\left|\log\left(\frac{x_{\widetilde{C}}^*}{x_C^*}\right)\right|$. For any $u, v \in \mathbb{R}^n$ and $C \in \mathbb{R}^{n \times n}$, defining $B(u, v; C)$ is a matrix with entries $B(u, v; C)_{ij} = \exp\left(\frac{u_i + v_j - C_{ij}}{\eta}\right)$, we have the following lemma

**Lemma 17.** *For $\tau > 0$ and $a \in \mathbb{R}^n$, if $\frac{u}{\tau} = \log a - B(u, v; C)\mathbf{1}_n$ and $\frac{u'}{\tau} = \log a - B(u', v'; C')\mathbf{1}_n$, then*

$$\left(\frac{1}{\tau} + \frac{1}{\eta}\right)\|u' - u\|_\infty \leq \frac{1}{\eta}\|v' - v\|_\infty + \frac{1}{\eta}\|C' - C\|_\infty.$$

*Proof of Lemma 17.* Taking the difference between $u/\tau$ and $u'/\tau$, for $i \in [n]$,

$$\frac{u_i' - u_i}{\tau} = \log\left(\frac{B(u, v; C)_i}{B(u', v'; C')_i}\right) = -\frac{u_i' - u_i}{\eta} + \log\left(\frac{\sum_j \exp\left(\frac{v_j' - C_{ij}'}{\eta}\right)}{\sum_j \exp\left(\frac{v_j - C_{ij}}{\eta}\right)}\right)$$

$$\leq -\frac{u_i' - u_i}{\eta} + \frac{\|v' - v\|_\infty}{\eta} + \frac{\|C' - C\|_\infty}{\eta},$$

which results in the final statement. $\square$

From the fixed-point equations for $(u_C^*, v_C^*)$ and $(u_{\widetilde{C}}^*, v_{\widetilde{C}}^*)$ and Lemma 17, we have

$$\left(\frac{1}{\tau} + \frac{1}{\eta}\right)\|u_{\widetilde{C}}^* - u_C^*\|_\infty \leq \frac{1}{\eta}\|v_{\widetilde{C}}^* - v_C^*\|_\infty + \frac{1}{\eta}\|\widetilde{C} - C\|_\infty$$

$$\left(\frac{1}{\tau} + \frac{1}{\eta}\right)\|v_{\widetilde{C}}^* - v_C^*\|_\infty \leq \frac{1}{\eta}\|u_{\widetilde{C}}^* - u_C^*\|_\infty + \frac{1}{\eta}\|\widetilde{C} - C\|_\infty,$$

leading to $\|u_{\widetilde{C}}^* - u_C^*\|_\infty + \|v_{\widetilde{C}}^* - v_C^*\|_\infty \leq \frac{2\tau}{\eta}\|\widetilde{C} - C\|_\infty$.

Hence, we find that

$$\left|\log\left(\frac{x_{\widetilde{C}}^*}{x_C^*}\right)\right| = \left|\log\left(\frac{\sum_{i,j=1}^n \exp\left(\frac{(u_{\widetilde{C}}^*)_i + (v_{\widetilde{C}}^*)_j - \widetilde{C}_{ij}}{\eta}\right)}{\sum_{i,j=1}^n \exp\left(\frac{(u_C^*)_i + (v_C^*)_j - C_{ij}}{\eta}\right)}\right)\right|$$

$$\leq \frac{1}{\eta}\|u_{\widetilde{C}}^* - u_C^*\|_\infty + \frac{1}{\eta}\|v_{\widetilde{C}}^* - v_C^*\|_\infty + \frac{1}{\eta}\|\widetilde{C} - C\|_\infty.$$

$$\leq \frac{2\tau + \eta}{\eta^2}\|\widetilde{C} - C\|_\infty.$$

$\square$

### E.2 Robust Semi-Optimal Transport

In this case, the constants are

$$S_x = 1, S_g = \log(n), S_H = 2\log(n), S_C = \frac{2\tau + \eta}{\eta}.$$

*Proofs of Inequalities.* The inequalities regarding $S_x, S_g$ and $S_H$ comes from the fact that $\|X_{\widetilde{C}}^k\|_1 = 1$, inequality (34) and inequality (28) of Section B, respectively. In terms of $S_C$, from equation (29) we have

$$g_C(X_C^*) = -\eta - \tau(1-\alpha) + \langle v_C^*, b^* \rangle, \qquad g_{\widetilde{C}}(X_{\widetilde{C}}^*) = -\eta - \tau(1-\alpha) + \langle v_{\widetilde{C}}^*, b^* \rangle.$$

Recall that it is the RSOT problem, thus $b^* = (X_{\text{rsot}}^*)^\top \mathbf{1}_n = \mathbf{b}$, thus

$$\left| g_{\widetilde{C}}(X_{\widetilde{C}}^*) - g_C(X_C^*) \right| = \left| \langle v_{\widetilde{C}}^* - v_C^*, b^* \rangle \right| \leq \|v_{\widetilde{C}}^* - v_C^*\|_\infty \|b^*\|_1 = \|v_{\widetilde{C}}^* - v_C^*\|_\infty.$$

**Upper bound for** $\|v_{\widetilde{C}}^* - v_C^*\|_\infty$**.** Defining $B(u, v; C)$ is a matrix with entries $B(u, v; C)_{ij} = \exp\left(\frac{u_i + v_j - C_{ij}}{\eta}\right)$. The fixed-points $u_C^*$ and $u_{\widetilde{C}}^*$ satisfy the following equations

$$\frac{u_C^*}{\tau} = \log a - \log B(u, v; C), \qquad \frac{u_{\widetilde{C}}^*}{\tau} = \log a - \log B(u', v'; C').$$

By Lemma 17,

$$\left(\frac{1}{\tau} + \frac{1}{\eta}\right)\|u_{\widetilde{C}}^* - u_C^*\|_\infty \leq \frac{1}{\eta}\|v_{\widetilde{C}}^* - v_C^*\|_\infty + \frac{1}{\eta}\|\widetilde{C} - C\|_\infty \tag{66}$$

By the fixed-point theorem, $B(u_C^*, v_C^*; C)^T \mathbf{1}_n = b$ and $B(u_{\widetilde{C}}^*, v_{\widetilde{C}}^*; \widetilde{C})^T \mathbf{1}_n = b$, and similarly we obtain

$$\frac{1}{\eta}\|v_{\widetilde{C}}^* - v_C^*\|_\infty \leq \frac{1}{\eta}\|u_{\widetilde{C}}^* - u_C^*\|_\infty + \frac{1}{\eta}\|\widetilde{C} - C\|_\infty. \tag{67}$$

Combining (66) and (67), we have $\|u_{\widetilde{C}}^* - u_C^*\|_\infty \leq \frac{2\tau}{\eta}\|\widetilde{C} - C\|_\infty$, and consequently $\|v_{\widetilde{C}}^* - v_C^*\|_\infty \leq \frac{2\tau+\eta}{\eta}\|\widetilde{C} - C\|_\infty$, completing the proof. $\qquad\square$

## F    Additional Experiments

### F.1    The Complexity of ROBUST-SINKHORN Algorithms on Synthetic Data

First, we investigate the runtime of Algorithm 3 (ROBUSTSINKHORN) for solving ROT, with the same synthetic setting of RSOT described in the main text (which will be repeated here for the sake of completion).

***Synthetic Data.*** We let $n = 100, \tau = 1$, generate entries of $C$ uniformly from the interval $[1, 50]$ and draw entries $a, b$ uniformly from $[0.1, 1]$ then normalizing them to form probability vectors. $\eta$ is set according to Theorem 1. For each $\varepsilon$ varying from $5 \times 10^{-2}$ to $5 \times 10^{-5}$, we calculate the number of theoretical and empirical iterations described above, as well as their ratio.

This experiment is run 10 times and we report their mean and standard deviation values in Figure 4, which shows that ROT lines experience a similar trend to those of RSOT in Section 5, with the ratio decreasing in the direction of $\varepsilon$ toward zero.

### F.2    The Complexity of ROBUST-SEMISINKHORN and ROBUST-SINKHORN Algorithms on Realistic Data

***MNIST Data.*** We consider each $28 \times 28$ MNIST image as a discrete distribution by flattening it into a 784-dimensional vector then performing normalization. For any pair of this MNIST distribution, the distance between their support equals to the Manhattan distance between corresponding pixel locations. Here, we let $\tau = 1$ and vary $\varepsilon$ from $10^{-2}$ to $10^{-5}$ (which is relatively small compared

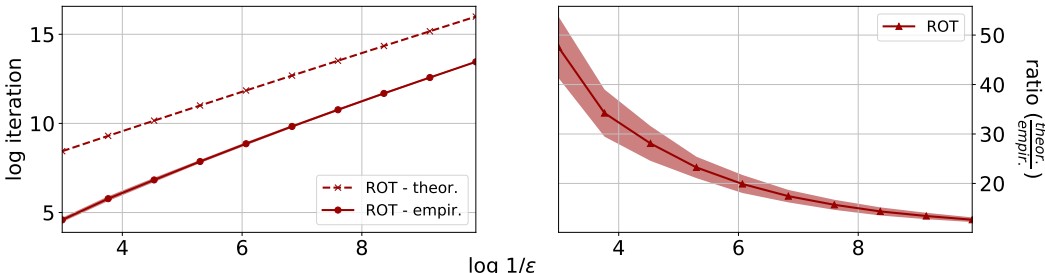

Figure 4: Complexity demonstration for ROBUSTSINKHORN on synthetic data. All the plots presented in this figure are set up similarly to those in Figure 3.

to $f_{\text{rsot}}(X^*_{\text{rsot}}) = 1.86 \pm 0.59$ and $f_{\text{rot}}(X^*_{\text{rot}}) = 1.15 \pm 0.33$ in this setting). For each value of $\varepsilon$, the regularized parameter $\eta$ is set accordingly as presented in Theorem 1. The theoretical and empirical values for the number of necessary iterations, as well as their ratio, are computed similar to the synthetic case, and their mean and standard variation values over 5 random MNIST pairs are reported in Figure 5. It can be seen from Figure 5 (compared to Figure 3 and 4) that the theory-practice relation

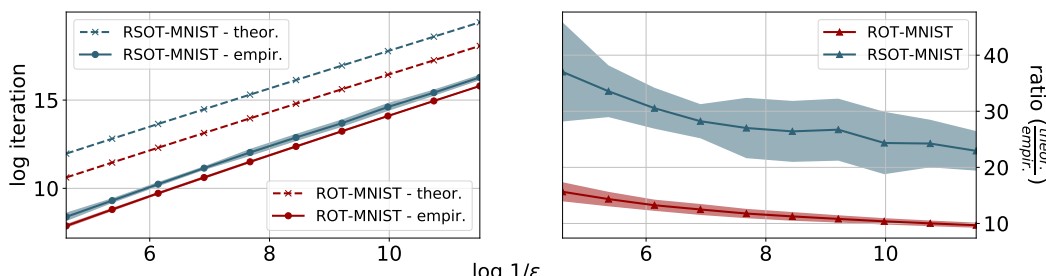

Figure 5: Complexity demonstration for ROBUST-SEMISINKHORN (blue) and ROBUST-SINKHORN (red) algorithms used to compute Robust Optimal Transport between MNIST images. All the plots presented in this figure are set up similarly to those in Figure 3.

of the two discussed algorithms (regarding the total iterations needed to reach an $\varepsilon$-approximation) behave quite similarly in both real and synthetic settings: two theoretical and empirical lines in the left plot run almost linearly while coming close to each other as $\varepsilon$ goes toward zero.

### F.3 Robust Comparison between Different Formulations

In this section, we compare the marginals induced by using different variants of optimal transport in the presence of corrupted measures. With the setting described in Figure 1, four following formulations are considered:

- Robust optimal transport with KL divergence (see Problem (9))

$$\min_X \quad \langle C, X \rangle$$
$$\text{s.t.} \quad X \geq 0, \|X\|_1 = 1, \mathbf{KL}(X\mathbf{1}_n\|\mathbf{a}) \leq \tau, \mathbf{KL}(X^\top\mathbf{1}_n\|\mathbf{b}) \leq \tau,$$

- Partial optimal transport [14]

$$\min_X \quad \langle C, X \rangle$$
$$\text{s.t.} \quad X \geq 0, \|X\|_1 = s, X\mathbf{1}_n \leq \mathbf{a}, X^\top\mathbf{1}_n \leq \mathbf{b},$$

- Robust optimal transport with total variation distance [23]

$$\min_X \quad \langle C, X \rangle$$
$$\text{s.t.} \quad X \geq 0, \|X\|_1 = 1, \mathbf{TV}(X\mathbf{1}_n, \mathbf{a}) \leq \tau, \mathbf{TV}(X^\top\mathbf{1}_n, \mathbf{b}) \leq \tau,$$

- Robust optimal transport with $\chi^2$ divergence [5]

$$\min_{X} \quad \langle C, X \rangle$$

$$\text{s.t.} \quad X \geq 0, \|X\|_1 = 1, \chi^2(X\mathbf{1}_n, \mathbf{a}) \leq \tau, \chi^2(X^\top \mathbf{1}_n, \mathbf{b}) \leq \tau.$$

The results are plotted in Figure 6. It is apparent that all the variants approximate the corrupted measures well with a proper choice of hyperparameter $\tau$ or $s$, and those with $f$-divergence relaxation have different behaviors when $\tau$ goes to infinity.

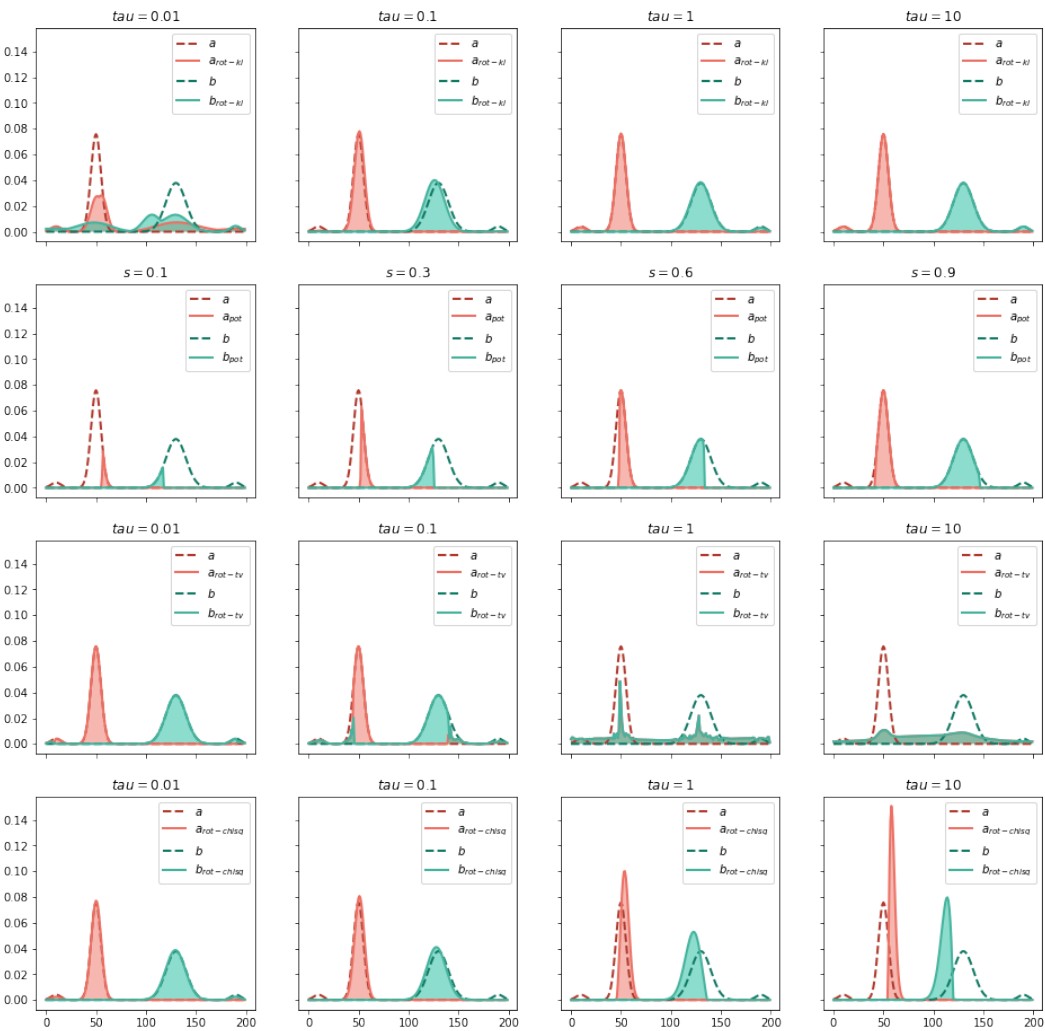

Figure 6: Comparison between robust optimal transport (ours, using KL divergence), partial optimal transport and robust formulations in [23] (using total variation distance) and [5] (using $\chi^2$-divergence), in that order from the first row to the fourth row, with different hyperparameter settings. The experiment setup is similar to the one in Figure 1.

## F.4 Some Applications of Robust Optimal Transport

In this section we demonstrate the robustness of two discussed versions of Robust Optimal Transport in two applications: color transfer and generative modeling.

### F.4.1 Color Transfer

Here, the optimal transport problem is conducted between the histograms of two images. Considering a source RGB image of size $h_s \times w_s \times 3$, and the a target RGB image of size $h_t \times w_t \times 3$, we can

present all the pixels in these images as point clouds in 3-dimensional RGB space (see the second row in Figure 7). To transfer the color from the target image into the source image, we compute the optimal transportation plan between the two corresponding point clouds and and use it to perform mapping from the source cloud to another point cloud that resembles the target cloud (i.e., transferring from the histogram in the first column to the third and fourth columns in Figure 7). As the total number of pixels in source/target image is large, it is a common practice to just sample a subset of pixels from each image, namely $\mathcal{I}_{src} = \{x_1, \ldots, x_n\}$ and $\mathcal{I}_{tar} = \{y_1, \ldots, y_m\}$. We consider two discrete measure formed by these two point clouds, $\alpha = \sum_i a_i x_i$ and $\beta = \sum_j b_j y_i$ and let $\mathbf{a} = [a_1, \ldots, a_n], \mathbf{b} = [b_1, \ldots, b_m]$. To compute the optimal transportation plan, we solve

(for standard optimal transport) $\qquad X^* = \underset{\substack{X\mathbf{1}_n=\mathbf{a}, \\ X^T\mathbf{1}_n=\mathbf{b}}}{\arg\min} \quad \langle C, X \rangle,$

(for robust optimal transport) $\qquad X^* = \underset{\substack{X\in\mathbb{R}_+^{n\times n}, \\ \|X\|_1=1}}{\arg\min} \quad \langle C, X \rangle + \tau\mathbf{KL}(X\mathbf{1}_n\|\mathbf{a}) + \tau\mathbf{KL}(X^T\mathbf{1}_n\|\mathbf{b}),$

where $C$ is the cost matrix with each entry $C_{ij} := \|x_i - y_j\|_2^2$. This optimal plan $X^*$ is then extended to cover all possible pixels using mapping estimation in [25]. In the experiment, we let $m = n = 1000, \tau = 1$ and $\mathbf{a}, \mathbf{b}$ being uniform mass vectors. Additionally, we approximate solutions of two optimal transport problems above using Sinkhorn algorithms on their entropic formulations with $\eta = 0.001$. To demonstrate the robustness when dealing with outliers in support points, we corrupt both source and target image by randomly changing their pixel intensities to other values (see Figure 7). It can be seen from the figure that the transferred color histogram induced by the OT solution still contains noisy values (see the bottom-right of the histogram visualization on the third column), while the transferred histogram resulted from ROT is clean as expected. As a consequence, the twilight scene corresponding to OT contains red noises at corners and is not as visually appealing as its ROT counterpart.

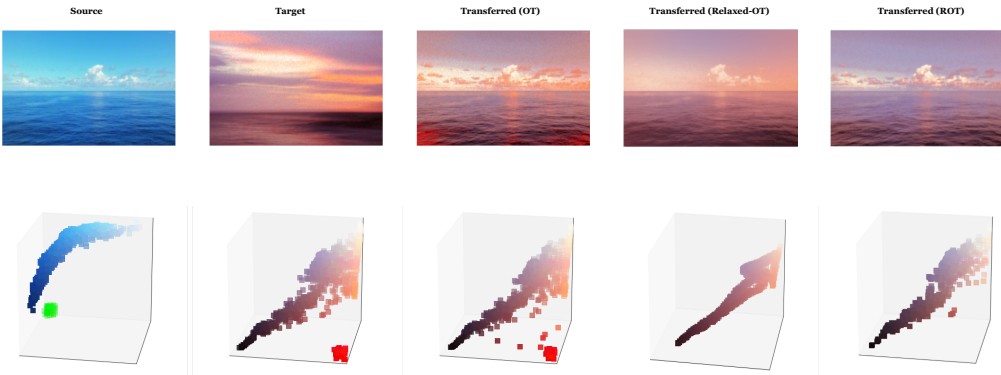

Figure 7: Demonstration for robust color transfer. The first row, from left to right, consists of source image, target image, and three last ones that are source images with each pixel replaced by its mapped value via standard optimal transport, relaxed optimal transport [28] and robust optimal transport (ours) respectively. The second row comprises corresponding (RGB) histograms of images on the first row. Note that the source (or target) image is corrupted by replacing pixels at random positions by green (or red) pixels, resulting in two green and red point clouds in corners in the first two histograms.

#### F.4.2 Generative Modeling

Next, we utilize the robust formulation of optimal transport in the problem of generative modeling. Assume that we have finite samples from a data distribution, which are $x_1, \ldots, x_n \sim p_{\text{data}}(x)$, the goal is to find a parametric mapping from a latent space $\mathcal{Z}$ to the data space $\mathcal{X}$, namely $g_\theta : \mathcal{Z} \to \mathcal{X}$, so that the pushforward measure $g_{\theta\#p_{\mathcal{Z}}}$ is close to the data distribution $p_{\text{data}}$ as much as possible. This problem can be formulated as to find $\theta^* = \arg\min_\theta \mathcal{D}(p_{\text{data}}, g_{\theta\#p_z})$, where $\mathcal{D}$ is a divergence between probability measures. Usually, $p_{\mathcal{Z}}$ is taken to be a simple distribution that we can easily sample from, such as an isotropic Gaussian distribution $\mathcal{N}(\mathbf{0}, \mathbf{I})$, and the divergence $\mathcal{D}(p_{\text{data}}, g_{\theta\#p_z})$ is approximated via samples from two distributions, i.e. by $\mathcal{D}(\alpha, \beta)$ where $\alpha$ and $\beta$ are two discrete

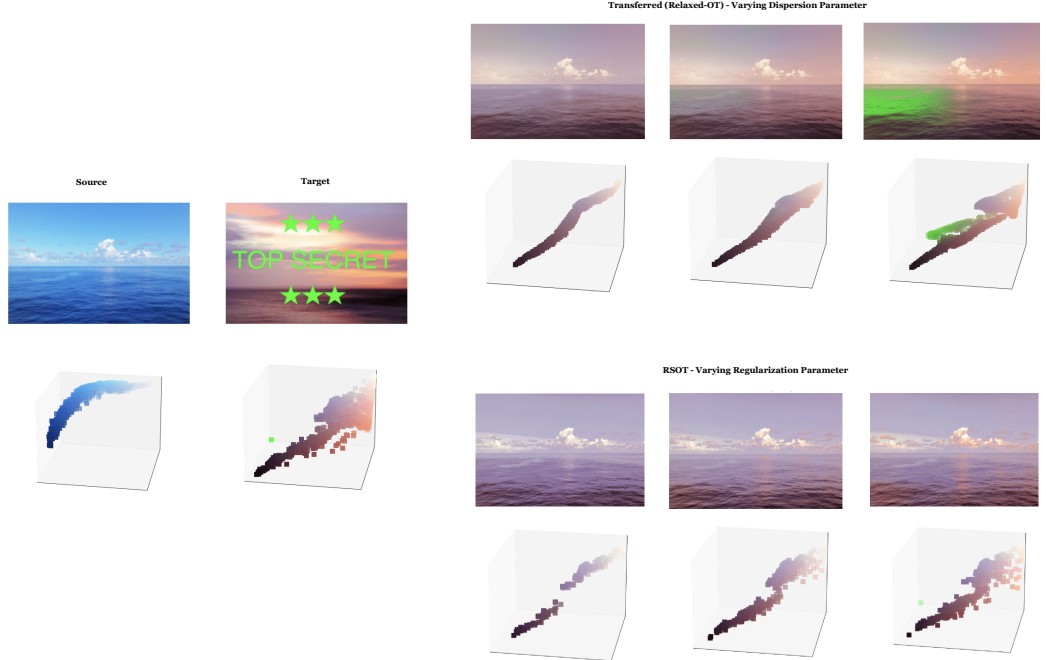

Figure 8: Comparison between relaxed optimal transport [28] and robust optimal transport in the color transfer problem. The setting is the same as in Figure 7, but here the "robust" parameters of both methods are varied (from left to right, the dispersion parameter of relaxed OT is set to $0.003, 0.03, 0.3$ respectively, and the parameter $\tau$ of robust OT is set to $0.1, 1, 10$ respectively). It is noticeable that the histogram of transferred image induced by the relaxed OT is not as diverse and exact as the one produced by our robust OT, resulting in a less visually appealing output.

measures supported on $n$ data samples $\{x_i\}_{i=1}^n$ and $m$ generated samples $\{g(z_i) : z_i \sim p_{\mathcal{Z}}(z)\}_{i=1}^m$, with probability histograms $\mathbf{a}$ and $\mathbf{b}$ respectively. We consider three versions of $\mathcal{D}$, which are

- Sinkhorn divergence in [15], which reads

$$SD_\eta(\alpha, \beta) = \mathcal{W}_\eta(\alpha, \beta) - \frac{1}{2}\mathcal{W}_\eta(\alpha, \alpha) - \frac{1}{2}\mathcal{W}_\eta(\beta, \beta),$$

where $\mathcal{W}_\eta(\alpha, \beta)$ is the Wasserstein distance, a special case of optimal transport where the cost comes from a metric,

- Entropic robust unconstrained optimal transport in Section 3.2, i.e.

$$ROT_\eta(\alpha, \beta) = \min_{\substack{X \in \mathbb{R}^{n \times n}, \\ \|X\|_1 = 1}} \langle C, X \rangle + \tau \mathbf{KL}(X\mathbf{1}_n \| \mathbf{a}) + \tau \mathbf{KL}(X^\top \mathbf{1}_n \| \mathbf{b}) - \eta H(X),$$

- Robust Sinkhorn divergence inspired from the above Sinkhorn divergence, which has the form

$$RSD_\eta(\alpha, \beta) = ROT_\eta(\alpha, \beta) - \frac{1}{2}ROT_\eta(\alpha, \alpha) - \frac{1}{2}ROT_\eta(\beta, \beta).$$

We train different generators corresponding to three different objectives, which are based on three variants of $\mathcal{D}$ listed above. Consider that data comes from a mixture of isotropic, two-dimensional Gaussians with four modes located at $(10, 0), (0, 10), (-10, 0)$ and $(0, -10)$. To demonstrate robustness, we corrupt the data by letting $10\%$ of them come from the uniform distribution on $[20, 25]$. We parameterize $g_\theta$ by a fully-connected neural network $(2 \rightarrow 64 \rightarrow \text{LeakyReLU} \rightarrow 128 \rightarrow \text{LeakyReLU} \rightarrow 2)$, and minimize the objective via stochastic gradient descent, where $D(\alpha, \beta)$ at each iteration is computed by sampling a batch of data and generated samples then running $k$ Sinkhorn updates. We set $\eta = 100, \tau = 1, k = 10, \mathcal{Z} \equiv \mathbb{R}^2$ and use Adam optimizer [18] with a learning rate of $0.001$. The generated distributions during the training process in three cases of interest are reported in Figure 9.

As shown in this figure, the objective derived from robust optimal transport can help the generator learn to ignore outliers in data distribution (see the third row), while the model based on standard optimal transport still generates noises (see the first row).

In addition to the simple Gaussian setting, we also demonstrate the generative capacity of robust optimal transport on the contaminated set of real MNIST images. Particularly, the dataset is $10\%$-corrupted by random image noises uniformly drawn from $[0, 1]^{28 \times 28}$. The generator is a fully-connected neural network mapping from 16-d Gaussian to $[0, 1]^{784}$ (the full architecture is $16 \rightarrow 500 \rightarrow$ Softplus $\rightarrow 500 \rightarrow$ Softplus $\rightarrow 784 \rightarrow$ Sigmoid). We train this network with the same procedure described in the previous paragraph, using the normal and the robust formulation of Sinkhorn divergence as the objective. The generated images are shown in Figure 10. As expected, while the network trained with the standard Sinkhorn divergence still generates noises (appearing as a mixed version of a MNIST image and a noise image), the network learned with the robust optimal transport ignores the noise and only produce clean digit pictures.

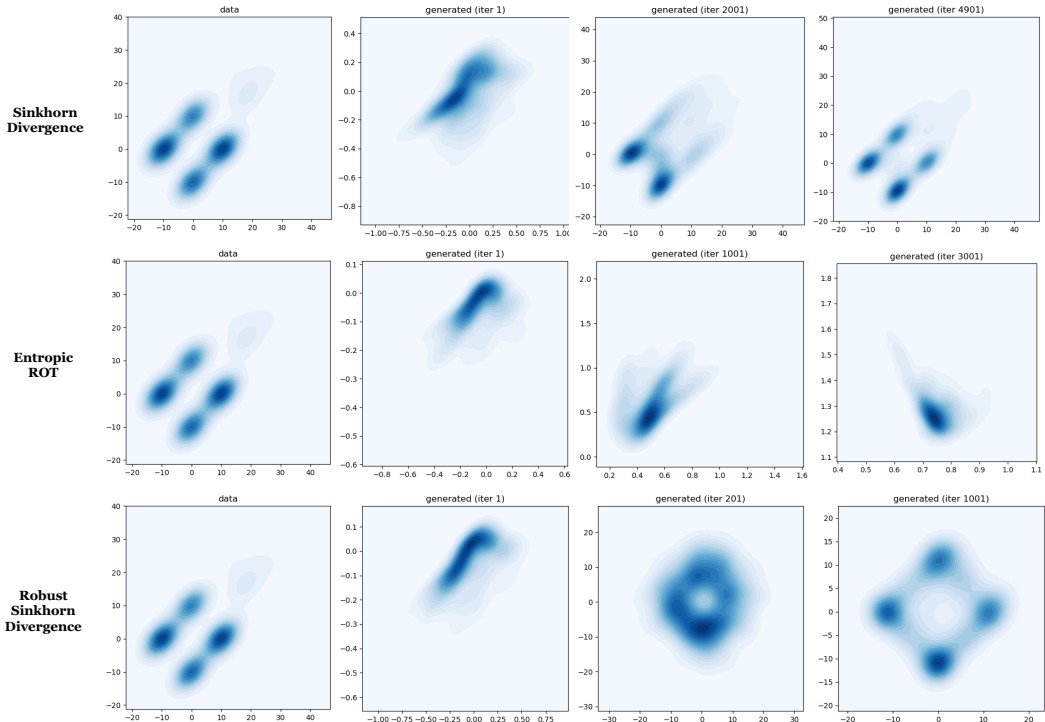

Figure 9: Generative modeling with three different objectives: Sinkhorn divergence (first row), entropic ROT (second row) and Robust Sinkhorn divergence (last row). In each image, we show 1000 points created by first sampling $z \sim \mathcal{N}(\mathbf{0}_2, \mathbf{I}_2)$ then generating $x_{gen} = g_\theta(z)$. At each row, from left to right, we present generated distributions at several iterations in the chronological order.

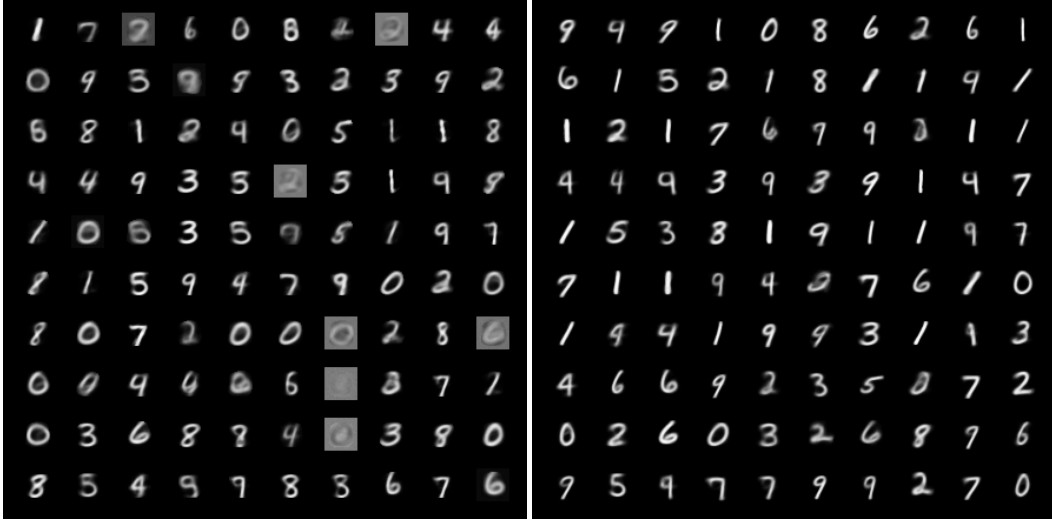

Figure 10: Generating contaminated MNIST data. The left and the right figures are the outputs of the generator trained with Sinkhorn divergence and with robust Sinkhorn divergence respectively.