# OpenReview forum: "On Robust Optimal Transport: Computational Complexity and Barycenter Computation"
_NeurIPS.cc/2021/Conference — NeurIPS 2021 Poster_

### Official Review · Reviewer_8LKP · 2021-07-13

**Rating:** 6
**Confidence:** 4

**Summary:**

This paper focuses on the computational complexity of robust optimal transport between discrete probability measures. The authors propose two Sinkhorn-based algorithms with near-optimal complexity to approximate the optimal cost. Furthermore, for solving a robust barycenter problem, they propose an algorithm based on iterative Bregman projections (IBP) and show that it also has near-optimal complexity between two discrete probability distributions.
Experiments verify the presented complexities for the above algorithms. The main limitations and future work directions are also discussed.

**Limitations And Societal Impact:**

- The complexity of ROBUSTIBP algorithm is near-optimal only when the number of discrete probability distributions, $m$, equals 2. This case is quite special and restrictive.

- In the experimental part, the approximation error of proposed algorithms is not discussed. Also, there is a lack of comparison with other methods.

- Optimal transport methods usually suffer from the curse-of-dimensionality.
For example, the convergence rate of the Wasserstein distance may be arbitrarily slow as the number of dimension increases.
I am wondering how does the size of the dimension affect the performance of the proposed method? Some discussion and empirical results are needed.

**Main Review:**

- For robust optimal transport and its corresponding barycenter problem, the authors develop Sinkhorn-based and IBP-based algorithms for solving them, and provide a comprehensive study of the computational complexity. The proposed algorithms can achieve near-optimal complexity, which outperform state-of-the-art competitors.

- The organization of this paper is satisfactory, and the motivation is very clear.

- The experiments are implemented on various synthetic and real datasets; the experimental settings and results are well explained.

- Some existing work considers the problem of robust optimal transport with similar  idea, e.g., [1] and [2]. More discussion is needed to compare them with the proposed method.

[1] Mukherjee, Debarghya, et al. "Outlier-Robust Optimal Transport." arXiv preprint arXiv:2012.07363 (2020).

[2] Jawanpuria, Pratik, N. T. V. Dev, and Bamdev Mishra. "Efficient robust optimal transport: formulations and algorithms." arXiv preprint arXiv:2010.11852 (2020).


**Time Spent Reviewing:**

2

---

> ### Author Response · Authors · 2021-08-10
> **Responses to Reviewer 8LKP**
>
> We would like to thank the reviewer for the constructive feedback and comments.
>
> ---Q1: *"Some existing work considers the problem of robust optimal transport with similar idea, e.g., [1] and [2]. More discussion is needed to compare them with the proposed method."*
>
> **Response**: We will include these references in the revision of our manuscript, as well as the discussions and empirical robustness comparisons. Briefly speaking, while [1] indeed considers the same corruption model as ours, [2] is essentially different to our work since they consider the worst possible cost function (i.e., robust to metric) while we aim at the robustness to measure corruption. Moreover, [1] mainly targets Minimum Kantorovich Estimation problems, and only deals with one marginal being possibly corrupted via the relaxation under total variation norm, while our ROT can tackle the outliers in both marginals.
>
> ---Q2: *"The complexity of ROBUSTIBP algorithm is near-optimal only when the number of discrete probability distributions, equals 2. This case is quite special and restrictive."*
>
> **Response**: Please refer to our general response (C.2).
>
> ---Q3: *"In the experimental part, the approximation error of proposed algorithms is not discussed. Also, there is a lack of comparison with other methods."*
>
> **Response**: In the experiment, we have plotted the number of iteration to reach a certain approximation error w.r.t. the approximation error itself.
>
> Though it is true that we don’t put our robust formulations in comparison to other robust methods, it is noteworthy that the main merits of our work lie in the new insight into the connection between UOT and ROT, as well as a new algorithm built on that insight with a near-optimal complexity for the robust barycenter problem. Our formulations, following the standard OT formulation, may find their own adaptations in future works on many machine learning problems where robustness is needed since we have shown that our robust formulations can be solved scalably and efficiently by Sinkhorn-based algorithms like the standard OT.
>
> However, we agree that it is worth comparing the robust effects induced by different formulations, and we will include the comparison with [1] and [3] (though their formulation is similar to ours except for the relaxing divergence) in the future revision.
>
> [3] Y. Balaji, R. Chellappa, and S. Feizi. Robust optimal transport with applications in generative modeling and domain adaptation. In NeurIPS, 2020.
>
> ---Q4: *"Optimal transport methods usually suffer from the curse-of-dimensionality. For example, the convergence rate of the Wasserstein distance may be arbitrarily slow as the number of dimensions increases. I am wondering how does the size of the dimension affect the performance of the proposed method? Some discussion and empirical results are needed."*
>
> **Response**: Generally speaking, we do not suffer more from the influence of the dimensionality on the sample and computational complexities compared to standard optimal transport. Specifically,
>
> - **Sample Complexity**: One important feature of our work is that we approximate the optimal values of robust optimal transport via its entropic-regularized version. Note that in Equation (3), if we take $P_1 = P, Q_1 = Q$, we get the standard OT.  As a result, the entropic-regularized ROT can be upper bounded by the entropic-regularized OT. Since the sample complexity of the latter had been shown to be of order $C(d) n^{- 1/ 2}$ (c.f. [4], [5]), it indicates that the same order can be attained for the former. Thus, our approaches based on the entropic regularized ROT do not suffer from the curse of dimensionality.
>
> - **Computational Complexity**: As in the standard discrete OT formulation where the cost matrix C is considered to be a given parameter, our algorithms and computational complexities are dimension-free regarding the ambient space. If the cost computation is taken into account, its complexity is dependent on which metric is taken (Euclidean distance, shortest distance on the nearest-neighbour graph, etc.)
>
> [4] Aude Genevay, Lénaïc Chizat, Francis Bach, Marco Cuturi, Gabriel Peyré. Sample complexity of Sinkhorn divergences. In AISTATS 2019
>
> [5] Gonzalo Mena and Jonathan Weed. Statistical bounds for entropic optimal transport: sample complexity and the central limit theorem. In NeurIPS, 2019.

---

> > ### Author Response · Authors · 2021-08-26
> > **Feedback before the discussion period ends**
> >
> > Dear Reviewer,
> >
> > We would like to thank you again for spending your time evaluating our paper.
> >
> > As the discussion period is expected to conclude next week, we look forward to hearing your feedback about whether we have addressed your concerns in the rebuttals. We would be happy to discuss if you still have any other concerns.
> >
> > Best,
> >
> > Authors

---

### Official Review · Reviewer_aBfM · 2021-07-14

**Rating:** 4
**Confidence:** 4

**Summary:**

The paper proposes the Robust Optimal Transport(ROT), Robust Semi-constrained Optimal Transport (RSOT), Robust Unconstrained Optimal Transport (RUOT) and Robust Semi-constrained Barycenter Problem (RSBP). In these problems, the equality constraints in the original OT problem is replaced by the soft KL divergence. Then the Sinkhorn-like algorithms are proposed to solve the corresponding dual problems. Finally, the detailed convergence analysis is given.

**Main Review:**

- It seems unfair to compare the proposed methods with the Sinkhorn algorithm, since the solution of the proposed method is even not in $\Pi(P,Q)$. Also, there is no comparison between the proposed methods and Sinkhorn, in terms of both accuracy and efficiency.
- By further relaxing the equality constraints, the problem should be simpler and easier to compute. But what's the purpose of such a relaxing?
- In Fig. 2, what's the meaning of different colors? Is there any way to enlarge the difference of different lines like plotting them in the log-domain? Currently they just mix together.
- Similar relaxation about OT has been proposed in [Fast Unbalanced Optimal Transport on a Tree] and its cited works. Please see the first formula of this paper.
- No comparison between the proposed methods and other similar ones, like the standard OT and UOT. Without such comparisons, it is hard to see the merits of the proposed ones.
- Many missing references about partial optimal transport:
  - Alessio Figalli.  The optimal partial transport problem. Archive for rational mechanics and analysis, 195(2):533–560, 2010
  - Luis A Caffarelli and Robert J McCann. Free boundaries in optimal transport and monge-ampere obstacle problems. Annals of mathematics, pages 673–730, 2010.

**Time Spent Reviewing:**

7

---

> ### Author Response · Authors · 2021-08-10
> **Response to Reviewer aBfM**
>
> We would like to thank the reviewer for the constructive feedback and comments.
>
> ---Q1: "*It seems unfair to compare the proposed methods with the Sinkhorn algorithm, since the solution of the proposed method is even not in $\Pi(P, Q)$.*"
>
> **Response**: For the robust OT, we generally cannot solve it directly by using Sinkhorn algorithm because its dual form is  more complicated than the dual forms of UOT and OT. Therefore, we need to develop new algorithms for solving the dual form of robust OT and we do not compare our algorithms with the Sinkhorn algorithm. We prove ( in Lemma 13 in Appendix D) that the optimal solution of robust OT can be obtained by normalizing the optimal solution of UOT. It is a new result that connects robust OT to the UOT. Furthermore, that connection makes it natural to develop simple algorithms for solving robust OT based on the Sinkhorn algorithm. At a high level, these algorithms are based on normalizing the Sinkhorn algorithm. The challenge of analyzing these Sinkhorn-based algorithms comes from the normalizing constants, and we need to develop techniques to deal with them. Finally, the normalization idea also provides an important insight into developing a near-optimal RobustIBP algorithm, which is a normalized version of the Iterative Bregman Projection (IBP) algorithm, for solving the robust barycenter problem.
>
> ---Q2: "*Also, there is no comparison between the proposed methods and Sinkhorn, in terms of both accuracy and efficiency.*"
>
> **Response**: As in our response to your question Q1, we cannot use the Sinkhorn algorithm directly for solving robust OT due to the complicated dual forms of robust OT. Furthermore, our robust algorithms are Sinkhorn-based algorithms for solving the robust OT. Hence, we do not compare the Sinkhorn-based algorithms to the Sinkhorn algorithm in the framework of robust OT.
>
> ---Q3: "By further relaxing the equality constraints, the problem should be simpler and easier to compute. But what's the purpose of such a relaxing?"
>
> **Response**: The purpose of relaxing the equality constraints is that we now consider a class of distributions that contains the true distribution or a good approximation to the true distribution, that class of distributions are controlled by the hyperparameter $\tau$ in the penalty function (KL divergence). The work [1] has demonstrated that these relaxations lead to favorable results under robust deep generative model and robust domain adaptation.
>
> Furthermore, we would like to clarify that by further relaxing the equality constraints, the problem is not necessarily simpler, because it depends on which one is more difficult when searching for a solution in a small space  and in a larger space. For example, hard marginal constraints are used as the destination to tell the Sinkhorn algorithm when to stop, while the UOT does not have that advantage.
>
> [1] Y. Balaji, R. Chellappa, and S. Feizi. Robust optimal transport with applications in generative
> 305 modeling and domain adaptation. In NeurIPS, 2020.
>
> ---Q4: "*In Fig. 2, what's the meaning of different colors? Is there any way to enlarge the difference of different lines like plotting them in the log-domain? Currently they just mix together.*"
>
> **Response**: As stated in the caption, different colors correspond to different trials (i.e., with different random seeds generating different data, while the parameters like $\eta$ and $\tau$ are fixed). Its sole purpose is to present the hardness of the case $m\geq 3$, in which our current technique is inapplicable (see our discussion in line 250). In particular, our result on the complexity of the barycenter problems roots in the convergence rate of dual variables (cf. Equation 15), which in turn relies on two bounds on $R_{uu}$ and $R_{uv}$ (defined on lines 250-251, the bound is on lines 243-244). Figure 2 empirically points out that while these bounds are tight in the case $m = 2$, only one of them is tight when $m\geq 3$. For visualization on the log-domain, see this https://imgur.com/5iHN54T.
>
> ---Q5: "*Similar relaxation about OT has been proposed in [Fast Unbalanced Optimal Transport on a Tree] and its cited works. Please see the first formula of this paper.*"
>
> **Response**: Thanks for your comment. We will include the reference [Fast Unbalanced Optimal Transport on a Tree] and its related works in our paper.
>
> ---Q6: "No comparison between the proposed methods and other similar ones, like the standard OT and UOT. Without such comparisons, it is hard to see the merits of the proposed ones."
>
> **Response**: We would like to divide our response to your comment into two parts:
>
> (i) The main focus of the paper is the computational complexity of the robust formulations and the barycenter problem based on them. The closest to our work is [1] with a similar formulation (please refer to our replies to Reviewer uvAE), but their work focuses on the computation method and applications. While the proposed methods for solving robust optimal transport in [1] are complicated and dependent on applications, our work provides theoretical studies on different computation methods, i.e., Sinkhorn-based algorithms, whose scalability and efficiency have been well-demonstrated in the OT literature (see [2]).
>
> (ii) Though our work doesn’t claim the novelty of the proposed formulations, we still provide empirical comparisons to other optimal transport forms. For instance, regarding UOT, we have Figure 1 that provides insights on the marginals induced by the optimal plan of UOT and ROT (and also Lemma 13 that connects the solution of those two, which is briefly described in Section 3.2). In terms of OT, we have experiments in the Appendix showing the superiority of ROT over OT-based approaches in real applications where objects of interest are subject to corruption.
>
> [2] M. Cuturi. Sinkhorn distances: Lightspeed computation of optimal transport. In NeurIPS, 2013.
>
> ---Q7: "*Many missing references about partial optimal transport:*
> *- a. Alessio Figalli. The optimal partial transport problem. Archive for rational mechanics and analysis, 195(2):533–560, 2010*
>
> *- b. Luis A Caffarelli and Robert J McCann. Free boundaries in optimal transport and monge-ampere obstacle problems. Annals of mathematics, pages 673–730, 2010.*"
>
> **Response**: Thanks for your comment. We will include these references in our paper. However, we would like to remark that while the partial optimal transport can possibly reduce the effect of outliers on the optimal transportation plan, it only transfers a fraction of the total mass between marginals, thus having a different behavior (see this experiment https://imgur.com/mBOlFpM, where $s$ is the transferred mass).

---

> ### Author Response · Authors · 2021-08-30
> **Feedback before the discussion ends**
>
> Dear Reviewer aBfM,
>
> We would like to thank you again for spending your time evaluating our paper.
>
> As the discussion period is expected to conclude early this week, we look forward to hearing your feedback about whether we have addressed your concerns in the rebuttals. We would be happy to discuss if you still have any other concerns.
>
> Best,
>
> Authors

---

### Official Review · Reviewer_uvAE · 2021-07-16

**Rating:** 6
**Confidence:** 3

**Summary:**

This paper proposes two robust formulation of the optimal transport problem (namely RSOT and ROT) based on the relaxation of the marginal constraints of the optimal transport plan using Kullback-Leibler divergence. The authors introduces algorithms based on Sinkhorn iterations in order to solve RSOT and ROT problems, and analyse their computational complexities. The fixed-support Wasserstein barycenter problem is also tackled using the proposed RSOT as a divergence, and the algorithm shows a better complexity than the one based on entropy regularised OT. Numerical experiments are also provided to support the theoretical results and applied task are performed (color transfer and generative modeling).

**Limitations And Societal Impact:**

Limitations :
- When dealing with color transfer applications, it is known that OT behaves poorly. To be fair, the authors should compare their results with more adapted OT-liked method, such as [Adaptive Color Transfer With Relaxed Optimal Transport, Julien Rabin, Sira Ferradans, Nicolas Papadakis] for example.
- The complexity of the RobustIBP algorithm, that aims at computing the Wasserstein barycenter, is only available when computing the barycenter of two measures.

Questions :
- I find the toy example in Figure 1 interesting in order to differentiate the proposed method and UOT. However, is the poor solutions coming from UOT are simply due to the choice of the regularisation parameter? Since UOT does not impose any constraints on the optimal plan being a probability measure, I imagine that larger parameters for the KL penalties on the marginals should give better results.

**Main Review:**

Originality : The proposed robust optimal transport is very similar the unbalanced optimal transport. Moreover the algorithms proposed are largely inspired from existing method (UOT [9]). For example, the ROT algorithm follows the generalised Sinkhorn algorithm with a extra step that is the normalisation of the transport plan.

Quality : The complexity analysis of the proposed algorithms (solving ROT, RSOT and RobustIBP) are theoretically strong, as they in particular provide an $\varepsilon$-approximation of the solutions. Additionally, a low-rank approximation improves the complexity of the algorithm solving ROT. On the other hand, ROT is presented as a robust method but there are no theoretical or experimental results that truly support this claim (except for Figure 1, see Limitations).

Clarity : This paper is well written and the different problems are clearly stated.

Significance : In my opinion, the improvements of the methods proposed in the article compared to the UOT is not clear. Indeed, both problem have the same complexity. Moreover, the methods are compared with UOT in terms of complexity analysis but not in terms of behaviour: how does an optimal transport plan move the mass under ROT or RSOT? Does the normalising step significantly impact the results? What are the key differences?

**Time Spent Reviewing:**

4

---

> ### Author Response · Authors · 2021-08-10
> **Responses to Reviewer uvAE**
>
> We would like to thank the reviewer for the constructive feedback and comments.
>
> ---Q1: "*The proposed robust optimal transport is very similar the unbalanced optimal transport. Moreover, the algorithms proposed are largely inspired from existing method (UOT [9]).*"
>
> **Response**: Please refer to our answers in General Comments (C.1).
>
> ---Q2: "*On the other hand, ROT is presented as a robust method but there are no theoretical or experimental results that truly support this claim (except for Figure 1, see Limitations).*"
>
> **Response**: Our formulation is motivated by the robust optimal transport in [1], in which the robustness has been already demonstrated experimentally (Figure 1 and experiments) and theoretically (Theorem 2). Note that while [1] uses a different form of optimization (i.e., inequality constraints instead of Lagrange multipliers as ours) and also different divergences, the spirit is the same. Furthermore, we provide the connection between ROT and UOT, and the theoretical result on the robustness of UOT can be found in [Lemma 1, [2]].
>
> [1] Y. Balaji, R. Chellappa, and S. Feizi. Robust optimal transport with applications in generative modeling and domain adaptation. In NeurIPS, 2020.
>
> [2] K. Fatras, T. Séjourné, N. Courty, and R. Flamary. Unbalanced minibatch optimal transport;
> applications to domain adaptation. In ICML, 2021.
>
> ---Q3: "*In my opinion, the improvements of the methods proposed in the article compared to the UOT is not clear. Indeed, both problem have the same complexity. Moreover, the methods are compared with UOT in terms of complexity analysis but not in terms of behaviour: how does an optimal transport plan move the mass under ROT or RSOT? Does the normalising step significantly impact the results? What are the key differences?*"
>
> **Response**: We would like to clarify the difference between the UOT problem and the ROT problem. The UOT formulation is to deal with two measures of different total masses, meanwhile, the ROT is to deal with the corrupted distributions, which have the same mass.  Figure 1 in our paper only shows the difference between the solutions of ROT and UOT. We by no means consider the ROT's solution as an improvement over the UOT's solution since they are different solutions to two different problems. We comment on their  transport plans below:
>
> - **(i)** Transportation plan of ROT/ RSOT: The transportation plans of ROT/RSOT move mass as that of OT usually does. The key difference here is that the marginals of the ROT/RSOT are not the same as the corrupted $P$ and $Q$; they are better approximations to the true distributions of $P$ and $Q$ than the corrupted distributions. That explains the effect of the regularizer to eliminate the corruption of the distributions $P$ and $Q$.
>
> - **(ii)** "Transportation plan" of UOT: The solution of UOT does not have to satisfy marginal constraints, so its form is dependent on the hyperparameter (see this experiment https://imgur.com/paM8Myg) and its interpretation is dependent on the application. For instance, in a cell-gene application, the UOT has a different name which is Waddington-OT. They solve a sequence of UOT problems to obtain a final solution for the  UOT. Each transportation plan entry for example, $\pi(x,y) = 1.2$ is interpreted as "cell $x$ will have on average 1.2 descendants with expression profile similar to $y$" (see [4]). We refer to [3] and [4] for more detailed explanations.
>
> Although the difference mentioned above between the UOT and the ROT, our new result linking the solutions of UOT and ROT gives some insights into the UOT optimal plan, showing that the mass moving induced by the optimal ROT plan is between two “uncorrupted” versions of two given marginals. Note that the normalization is crucial (and not at all trivial), since it makes the transportation plan a valid mass-moving between distributions, and since it is performed universally on the UOT plan, it gives that plan a new meaning.
>
> [3] G. Schiebinger, J. Shu, M. Tabaka, B. Cleary, V. Subramanian, A. Solomon, S. Liu, S. Lin, P. Berube, L. Lee, et al. Reconstruction of developmental landscapes by optimal-transport analysis of single-cell gene expression sheds light on cellular reprogramming, BioRxiv.
>
> [4] https://broadinstitute.github.io/wot/tutorial/
>
> ---Q4: "*When dealing with color transfer applications, it is known that OT behaves poorly. To be fair, the authors should compare their results with more adapted OT-liked method, such as [Adaptive Color Transfer With Relaxed Optimal Transport, Julien Rabin, Sira Ferradans, Nicolas Papadakis] for example.*"
>
> **Response**: It is noteworthy that our formulation is simple with only one tuning parameter $\tau$ regarding the robustness. It also can be solved efficiently by Sinkhorn algorithm (which is scalable and GPU-friendly) with provable guarantees. Furthermore, it can be used in many applications where OT has been already presented. While the RelaxedOT [5] can indeed provide a robust approach for color transfer (see these experiments https://imgur.com/Cv3URrb), it has some limitations: first, it is quite complicated and tailored specifically to that application with many tuning hyperparameters; second, as seen in the experiment, the histogram of transferred image induced by the RelaxedOT is not as diverse and exact as the one produced by our ROT, resulting in a less visually appealing output.
>
> [5] Julien Rabin, Sira Ferradans, and Nicolas Papadakis.
> Adaptive color transfer with relaxed optimal transport. In Proc. of International Conference on Image
> Processing, pages 4852–4856, 2014f
>
> ---Q5: "*The complexity of the RobustIBP algorithm, which aims at computing the Wasserstein barycenter, is only available when computing the barycenter of two measures.*"
>
> **Response**: Please refer to our answers in General Comments (C.2).
>
> ---Q6: "*I find the toy example in Figure 1 interesting in order to differentiate the proposed method and UOT. However, is the poor solutions coming from UOT are simply due to the choice of the regularisation parameter? Since UOT does not impose any constraints on the optimal plan being a probability measure, I imagine that larger parameters for the KL penalties on the marginals should give better results.*"
>
> **Response**:  In Figure 1, the KL penalty parameter $\tau$ is set to 0.1 for all cases. Note that $\tau$ is used to eliminate the corruption from the sampled data. Though it is true that a larger $\tau$ for UOT will bring the marginals of the transportation plan close to the given marginals, the other side of the coin is that these approximated marginals will be subject to corruption in our noisy setting (see this https://imgur.com/paM8Myg). Moreover,  a larger $\tau$ also increases the complexity of the algorithm.

---

> > ### Author Response · Authors · 2021-08-26
> > **Feedback before the discussion period ends**
> >
> > Dear Reviewer,
> >
> > We would like to thank you again for spending your time evaluating our paper.
> >
> > As the discussion period is expected to conclude next week, we look forward to hearing your feedback about whether we have addressed your concerns in the rebuttals. We would be happy to discuss if you still have any other concerns.
> >
> > Best,
> >
> > Authors

---

> > > ### Comment · Reviewer_uvAE · 2021-08-28
> > > **Post-rebuttal**
> > >
> > > Many thanks for your detailed responses to my concerns, and for the additional figures. I agree that UOT and ROT are similar enough to expect ROT to enjoy the robustness properties of UOT, nevertheless I think this robustness claim is still vague. However, I believe the main issues have been addressed, and I will change my score accordingly.

---

### Official Review · Reviewer_7VwK · 2021-07-17

**Rating:** 6
**Confidence:** 4

**Summary:**

The present paper is on optimal transport problems. The authors derived computational complexity results for robust optimal transport and robust barycenter in the special case with two marginals.

**Limitations And Societal Impact:**

The limitations have been discussed. It is stated that there is not societal impact.

**Main Review:**

Originality:
The theoretical results derived in this work is new. The methods and proof techniques are essentially the same as [1].

Quality:
The theoretical results and the associated proof seem to be solid. The result on barycenter is restricted to the setting with only two marginals.
Though this is a pure theoretical work, the authors provided numerical examples to support their claims, which is a plus.

Clarity:
The paper is well-written. The presentation is well-structured and clear.

Significance:
The theoretical results in this paper are elegant, but I feel the contribution is incremental in view of [1]. Also, the barycenter result is a bit restricted.

[1] K. Pham et al, On unbalanced optimal transport: an analysis of Sinkhorn algorithm

**Time Spent Reviewing:**

4

---

> ### Author Response · Authors · 2021-08-10
> **Responses to Reviewer 7VwK**
>
> We would like to thank the reviewer for the constructive feedback and comments.
>
> ---Q1: "*The result on barycenter is restricted to the setting with only two marginals.*"
>
> **Response**: Please refer to our answers in General Responses (C.2).
>
> ---Q2: "*The contribution is incremental in view of [1].*"
>
> **Response**: Please refer to our answers in General Responses (C.1).

---

> > ### Comment · Reviewer_7VwK · 2021-09-16
> > **acknowledge**
> >
> > Thank you for the response. I will keep my score unchanged.

---

> ### Author Response · Authors · 2021-08-30
> **Feedback before the discussion ends**
>
> Dear Reviewer 7VwK,
>
> We would like to thank you again for spending your time evaluating our paper.
>
> As the discussion period is expected to conclude early this week, we look forward to hearing your feedback about whether we have addressed your concerns in the rebuttals. We would be happy to discuss if you still have any other concerns.
>
> Best,
>
> Authors

---

### Official Review · Reviewer_yb1x · 2021-07-19

**Rating:** 5
**Confidence:** 2

**Summary:**

This paper presents the complexity analysis of two problems: robust optimal transport and robust barycenter problem. For both these problems, the paper analyzes a sinkhorn-based algorithm and obtains a complexity of O(n^2 / \epsilon). Finally, the authors show some numerical analysis to support the theoretical claims.



**Limitations And Societal Impact:**

The similarity with [26] seem to be the biggest concern for me.

**Main Review:**

My main concern is that the paper is very similar to the analysis in  [26] where the convergence proof of unbalanced Sinkhorn is provided. This paper analyses a modification of unbalanced optimal transport in which the variable X is forced to be a transportation plan. This modification also translates to a normalization step in the sinkhorn algorithm. In Figure 1, the authors show why this modification is important. The convergence proof of Theorem 1, however, looks very similar to [26]. There is a slight difference due to the normalization step.

The authors also analyze the robust barycenter problem, which is an extension of the robust OT problem. This convergence analysis, which looks like an extension of the robust OT analysis, provides a O(n^2/\epsilon) solution for the barycenter problem.

I am not a theory person, so I did not go over the proofs in detail. However, from a quick look, I am concerned about the similarity of the problem/analysis compared to [26] and not sure of the novelty. I would like if the authors help clarify this.


**Time Spent Reviewing:**

2.5 hours

---

> ### Author Response · Authors · 2021-08-10
> **Responses to Reviewer yb1x**
>
> We would like to thank the reviewer for the constructive feedback and comments.
>
> ---Q1: "*The similarity with [26] seems to be the biggest concern for me.*"
>
> **Response**: We refer to our answers in General Responses, which address the reviewer's concern about the comparison between ROT and UOT, as well as emphasize the significance of our main contribution regarding the robust barycenter problem.

---

> > ### Author Response · Authors · 2021-08-30
> > **Feedback before the discussion period ends**
> >
> > Dear Reviewer yb1x,
> >
> > We would like to thank you again for spending your time evaluating our paper.
> >
> > As the discussion period is expected to conclude early this week, we look forward to hearing your feedback about whether we have addressed your concerns in the rebuttals. We would be happy to discuss if you still have any other concerns.
> >
> > Best,
> >
> > Authors

---

### Author Response · Authors · 2021-08-09
**General Responses to Main Concerns**

We would like to thank reviewers for spending their time reviewing our paper. Below, we would like to provide a response to reviewers' main concerns:

(C.1) Some reviewers raised an issue regarding the similarity between the analyses of RSOT/ROT and those of UOT [1]. We would like to take this opportunity to highlight the crucial differences:

- (R.1.1) In terms of RSOT, the main difference (and difficulty) lies in the dual form (7) in the paper, especially the last term $\langle - v, b \rangle$. This is a term appearing in the dual form of standard entropic OT, i.e. $\eta ||B(u, v)||_{1} + \langle -u, \mathbf{a}\rangle + \langle -v, \mathbf{b}\rangle$.

     According to [1], the dual form of UOT is $\eta||B(u, v)||_{1} +\tau\left\langle e^{-u / \tau}, \mathbf{a}\right\rangle +\tau\left\langle e^{-v / \tau}, \mathbf{b}\right\rangle $.

     The dual form of RSOT is $\eta ||B(u,v)||_1 + \tau \langle e^{-u/\tau},\mathbf{a} \rangle - \langle v,\mathbf{b}\rangle $.
Thus, looking at the dual form, RSOT can be viewed as a mixture of OT and UOT, and the difficulty in analyzing RSOT (compared to UOT) can be seen from the fact that the best-known complexity of the Sinkhorn algorithm for solving OT is $\mathcal{O}(n^2/ \varepsilon^2)$.

- (R.1.2) Regarding ROT, there are two challenges that prevent us from directly using the result of UOT in [1] to study ROT. First, if we directly analyze the dual form of the entropic ROT (see Equation (53), Appendix D) for Sinkhorn-like algorithmic development, we will not get closed-form derivations for the updates (see our discussion after Lemma 12 in Appendix D). Thus, our novel discovery, which is a bridge between the optimal solution of ROT and that of UOT, not only enables us to develop a Sinkhorn-based algorithm for ROT in Algorithm 3 but also helps provide the computational complexity for that algorithm. Second, even when we figure out the connection between UOT and ROT (Lemma 13, Appendix D), a lower bound for $||X^{\star}||_1$, where $X^{\star}$ is the UOT solution, is still needed. However, it is non-trivial to establish such a lower bound.

[1] K. Pham, K. Le, N. Ho, T. Pham, and H. Bui. On unbalanced optimal transport: An analysis of Sinkhorn algorithm. In ICML, 2020.

(C.2) Some reviewers mentioned that the complexity for solving the robust barycenter problem is only established for two marginals, which might be too special and restrictive.

- (R.2.1) To the best of our knowledge, the computational complexity of the robust barycenter problem has remained an open problem in the literature, and no similar results appear before our work. The challenges of solving the robust barycenter problem for even the case $m=2$ lie in
 its complicated dual form (cf. equation (12)) in which an efficient algorithm is non-trivial to develop. Our main discovery is in Lemma 1, that bridges the optimal solution of the entropic regularized robust barycenter problem to its entropic barycenter counterpart. This connection allows us to develop RobustIBP algorithm in Algorithm 2.

- (R.2.2) For the case $m \geq 3$, we note that the geometric convergence is difficult to prove, since the updates on $\{v_k\}$ in Algorithm 2 are intertwined, which could not be separated as the case $m=2$. We would like to emphasize that to the best of our knowledge, there has been no work on the case $m \geq 3$ that obtains the near-optimal complexity for the UOT. That highlights the challenge for the multi-marginal case.

- (R.2.3) Finally, as mentioned in the paper (lines 247-248), it is noteworthy that if we have the geometric convergences for the dual variables $\mathbf{u^k} $ and $\mathbf{v^k}$ for the case $m \ge 3$, which we conjecture to be true from the experiment https://imgur.com/z0fj87q, the overall complexity still holds.

---

### Decision · Program_Chairs · 2021-09-27

**Decision:**

Accept (Poster)

**Comment:**

Most of the reviewers and the AC agree that the submission makes a worthwhile theoretical contribution in designing and analyzing new algorithms for RSOT and RIBP.

The reviewers highlight the following main concerns:
- The algorithm appears to be a simple normalized version of the Sinkhorn algorithm for UOT.
- the technical novelty appears to be somewhat incremental, as the necessary techniques already appear in the OT and UOT literature.
In the rebuttal, the authors explain that the obstacle to overcome in the analysis of the new algorithms. They also stress the fact that no simple, very efficient algorithms were known for RSOT and RIBP before this work. The AC believes that the simplicity of the algorithm and the lack of other algorithms besides standard convex solvers makes this paper close to the acceptance threshold for NeurIPS.

A number of issues spotted by the reviewers and the AC could be addressed by the authors in a final version:
- the discussion of the significance and novelty of the convergence analysis of RSOT should appear in the paper,
- the experimental section would ideally include more varied experiments. In particular, larger examples would be nice to test the scalability of the method, even if it may not be possible to run cvxpy on them to obtain the ground-truth optimum.